# Prompts to Proxies: Emulating Human Preferences via a Compact LLM Ensemble

## Abstract

Large language models (LLMs) often collapse toward average responses, obscuring the diversity needed to model different population-level preferences. While prompting can steer models toward diverse responses, it remains a non-trivial challenge on how it can be used to efficiently align with the preference of a target population. We propose a new theoretical lens, preference reconstruction theory, which formalizes population preference alignment as the construction of a functional basis of proxy agents. We implement this via Prompts-to-Proxies (`P2P`), a framework for preference reconstruction that formulates alignment as a two-stage problem. First, we use structured prompting with entropy-based adaptive sampling to construct a diverse set of endowed agents, each representing a vector in the latent preference space. Second, we reconstruct the population preference by estimating sparse weights over these agents via L1-regularized regression, aligning resulting aggregate response distribution with observed data. This yields a compact proxy population that captures both scope and distribution of preferences without demographic conditioning. `P2P` offers a cost-effective alternative to large-scale personalization and a principled testbed for studying pluralistic alignment. We validate the approach through an empirical evaluation on 14 waves of the American Trends Panel, demonstrating high-fidelity reconstruction, substantial diversity, and cross-domain generalization.

## 1 Introduction

Large language models (LLMs) have demonstrated remarkable success in emulating human-like responses across a wide range of tasks, coinciding with a shift away from monolithic alignment methods such as supervised fine-tuning (SFT) (Tan et al., 2024) and reinforcement learning from human feedback (RLHF) (Bai et al., 2022; Christiano et al., 2017). These conventional approaches, while effective for general-purpose assistants, produce a single dominant behavioral profile and are thus ill-suited for modeling the plurality of values, beliefs, and preferences found in real-world populations. LLMs, by contrast, have been lauded as a more flexible and scalable alternative due to their ability to generate diverse outputs through prompting alone. However, they are themselves trained to minimize loss over broad and heterogeneous datasets, which may lead to an averaging over multiple modes of response—ultimately limiting their capacity to reflect the full spectrum of human perspectives (Feng et al., 2024; Kirk et al., 2024; Lee et al., 2024; Slocum et al., 2025).

To overcome this, researchers have increasingly turned to pluralistic alignment (Chen et al., 2025; Feng et al., 2024; Sorensen et al., 2024b), a growing body of work that aims to align models not to a single gold standard, but to the diversity of plausible human responses. Within this literature, methods differ both in how pluralism is formalized and in how much data they require[1]. A particularly promising strand in this literature is post-training conditioning, where models are guided at inference time using structured prompts called *endowments* which encode sociodemographic traits or behavioral dispositions that condition an LLM agent to emulate the responses of a particular type of profile. These endowments are also known as personas (Castricato et al., 2025) or Silicon Samples (Argyle et al., 2023; Horton, 2023). They offer a lightweight alternative to post-training approaches such as fine-tuning (Feng et al., 2024; Yuan et al., 2024) and in-context learning (Adams et al., 2025; Chen et al., 2024). Cheap and fast inference-time conditioning gives way to mixture or

---

[1]See Appendix E

vector formulations of pluralistic alignment, which combine preferences via multi-objective rewards to avoid averaging away minority preferences (Chen et al., 2025; Feng et al., 2025), and a key open challenge lies in using them to efficiently align with the preferences of a target population.

In this work, we introduce `P2P`, a new theoretical and algorithmic framework for preference reconstruction, inspired by revealed preference theory in economics. Rather than condition on fixed identities, we instead learn a latent representation of population-level preferences and construct a compact set of diverse endowments that span this preference space using entropy-based adaptive sampling. These endowments act as proxy decision-makers whose responses expose the varied latent preferences of the target population. We then select a representative and compact subset of endowments and optimize their aggregation weights to match the observed response distribution of a target population. This two-stage process yields a distributionally pluralistic model that captures variation by recovering the functional structure of preferences in the population.

`P2P` is validated using real-world survey data from OpinionsQA (Santurkar et al., 2023) and the American Trends Panel (Kennedy, 2015), demonstrating that `P2P` can approximate the ground-truth distribution of responses with fewer agents and lower cost than traditional methods. We view `P2P` as a step toward addressing a broader challenge in the social sciences: the declining public willingness to participate in surveys (Kennedy, 2019; U.S. Bureau of Labor Statistics, 2025). As response rates drop across regions (Jabkowski & Cichocki, 2025) and demographics (Clinton et al., 2022; Parshall, 2024), synthetic agents that reflect pluralistic distributions offer a promising tool for simulating public opinion and correcting for sample selection bias.

Our contributions are as follows:

- **Theoretical foundation.** We formalize population preference alignment as a problem of representation learning, and establish the preference reconstruction theory, which states that learnable population preference can be represented using a functional basis of proxy agents and with appropriately determined aggregation weights.
- **Modular alignment system.** We implement the preference reconstruction theory for preference alignment via a modular system, Prompts to Proxies (`P2P`), that combines active endowment generation (to promote response diversity) with regression-based aggregation (to estimate preferences without demographic alignment).
- **Empirical validation.** We validate `P2P` on 14 waves of the American Trends Panel (ATP), showing that it recovers population preference patterns with high fidelity, while detailed baseline and ablation studies highlight its cost efficiency and design insights.

## 2 MODELING PREFERENCES VIA FUNCTIONAL BASES

**Alignment as representation learning** Mathematically, human-AI preference alignment can be formalized as a representation learning problem, inspired by the revealed preference theory (Mas-Colell et al., 1995; Samuelson, 1938). We denote a latent individual human preference $p$ as a vector in the abstract space $\mathcal{P}$, i.e., $p \in \mathcal{P}$, which manifests via a decision-making mechanism $h$ into observed responses, denoted by vector $r \in \mathcal{R} \subset \mathbb{R}^n$. Similarly, we denote the latent model[2] preference $\hat{p}$ as a vector in the abstract space $\widehat{\mathcal{P}}$[3], which manifests through the inference mechanism $f$ into observed responses $\hat{r} \in \widehat{\mathcal{R}} \subset \mathbb{R}^n$. The representation learning process can thus be abstracted as:

$$p \xrightarrow{h} r \xrightarrow{f^{-1}} \hat{p}. \tag{1}$$

where $f^{-1}$ is implemented via a learning algorithm.

While the above describes the learning process of an individual preference, alignment of large language models (LLMs) is usually done using responses from a human population. However, this target population is typically ill-defined for general-purpose LLMs, which have been through distinct

---

[2]In this paper, we use model and LLM interchangeably, as we contextualize alignment in the NLP context.

[3]We intentionally define the model preference space $\widehat{\mathcal{P}}$ as an abstract space to accommodate the fact that both training and prompting can lead to different model preferences.

stages of pretraining, supervised finetuning (SFT) and reinforcement learning from human feedback (RLHF), in which data sources are the responses from different user groups. In this respect, we introduce learnability of individual human preference in a revealed sense.

**Definition 2.1.** Individual human preference is *learnable in the revealed sense* if $h(\mathcal{P}) \subset f(\hat{\mathcal{P}})$.

**Learning the population preference via survey data** In this paper, we focus on preference alignment with a specific human population, motivated by recent calls on pluralistic alignment (Sorensen et al., 2024b) and the use of LLMs in social science research (Argyle et al., 2023; Horton, 2023). In particular, we use aggregate responses in general social surveys as the alignment yardstick, because these surveys are designed to represent the views of the target population. Formally, aggregate responses in a survey are defined as weighted averages of individual responses:

$$r_{\text{pop}} = \sum_{i=1}^{I} w_i h(p_i). \tag{2}$$

where $w_i$ is the weight of individuals with preference $p_i$ in the population, $I$. Accordingly, we define learnability of human population preference in a revealed sense.

**Definition 2.2.** Human population preference is *learnable in the revealed sense* if conv $(h(\mathcal{P})) \subset f(\hat{\mathcal{P}})$.

The convex hull of the revealed human preferences comprises all possible interpolated signals from humans. Human revealed preference is learnable if all possible interpolated signals from humans are fully contained inside the set of predictions generated by the learned model. This requires the model to cover the span of human preferences.

A direct consequence is that if human population preference is learnable in the revealed sense, so is individual human preference—we defer the proof to Appendix A.

Methodologically, the population preference, revealed through aggregate responses $r_{\text{pop}}$, can be learned using two different paradigms. The first relies on a single model, called a *representative LLM agent*, which has a preference $\hat{p}_{\text{pop}} \in \hat{\mathcal{P}}$ such that $r_{\text{pop}} = f(\hat{p}_{\text{pop}})$[4]. The second, which relates to our work, relies on an ensemble of $J$ models, whose preferences $\{\hat{p}_j\}_{j=1}^{J}$ satisfy $r_{\text{pop}} = \sum_{j=1}^{J} \tilde{w}_j f(\hat{p}_j)$, with $\{\tilde{w}_j\}_{j=1}^{J}$ convex weights. Here we provide two theorems with proofs deferred to Appendix A.

**Theorem 1** (Representative revealed preference learning). If human population preference is learnable in the revealed sense, it can be learned by a representative LLM agent with model preference $\hat{p}_{\text{pop}} \in \hat{\mathcal{P}}$.

**Theorem 2** (Ensemble revealed preference learning). If human population preference is learnable in the revealed sense, it can be learned by an ensemble of LLM agents with model preferences $\{\hat{p}_i\}_{i=1}^{I}$.

**Efficient learning via preference reconstruction** Based on Theorem 2, we can establish an important theorem (Appendix A) which we call the *preference reconstruction theory*.

**Theorem 3** (Preference reconstruction theory). If human population preference is learnable in the revealed sense, the ensemble of LLM agents that can learn this is not unique.

The direct implication of Theorem 3 is that instead of trying to find a one-to-one map between $p_i$ and $\hat{p}_i$, we can construct a *functional basis*—a set of proxy LLM agents with distinct preferences $\{\hat{p}_j\}_{j=1}^{J}$—that represents the human population preference. This distinguishes our focus from the demographic conditioning approach (Argyle et al., 2023; Castricato et al., 2025), which is a post-training alignment method that aims to identify the one-to-one map between $p_i$ and $\hat{p}_i$ through restrictive access to sensitive personal information and labor-heavy profile matching. Figure 1 provides a functional perspective of the preference reconstruction theory.

**Alignment as a two-stage problem** Preference reconstruction via a functional basis can be formulated as a concrete learning problem:

$$\min_{\{\hat{p}_j, \tilde{w}_j\}_{j=1}^{J}} L(r_{\text{pop}}, \sum_{j=1}^{J} \tilde{w}_j f(x_0; \hat{p}_j)). \tag{3}$$

---

[4]We provide in Appendix A a necessary and sufficient condition for its existence.

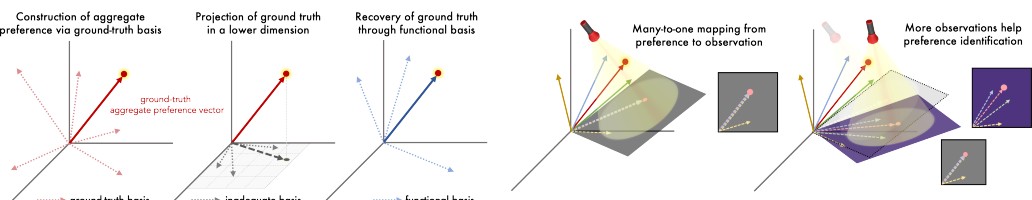

Figure 1: **Left**: Finding a functional basis to recover latent ground-truth preference. **Right**: Identification of latent preference through multiple observations.

where $L$ is a loss function.

Directly optimizing (3) is challenging, especially as the dimension $J$ is an optimizer per se.

As a solution strategy, we treat alignment as a two-stage problem with distinct stage goals. The first stage prioritizes the construction of a diverse set of proxy agents, measured in a revealed sense by their response variability. This goal aligns with the notion of *pluralistic alignment* (Sorensen et al., 2024b), which stresses the importance of representing a spectrum of human values rather than converging on a single ideal. The second stage concentrates on selecting a parsimonious ensemble from the set of candidate agents, based on the fit between ensemble aggregate responses and human ground truth. To avoid overfitting and test whether the ensemble has indeed learned the population preference, we partition survey questions into training and test sets, using the training set for selection and test set for validation. This solution strategy allows us to implement preference reconstruction using a modular system, which we describe in the next section.

## 3 PROMPTS TO PROXIES (P2P)

To implement our preference reconstruction strategy in the survey context, we engineer a modular alignment system called **Prompts to Proxies (P2P)** (Figure 2). The system consists of two core components: (1) active endowment generation, powered by a dynamic attribute bank, and (2) regression-based aggregation, where constrained variable section algorithms form the backbone. They concretize the two stages of alignment introduced in the previous section and enable the generation of a compact set of opinionated LLM agents to emulate population-level survey results.

### 3.1 SURVEY SPLITTING AND EVALUATION PROTOCOL

To support the alignment procedure, the full set of survey questions denoted $\mathcal{Q}$ are partitioned into training, validation, and test subsets $\{\mathcal{Q}_i\}_{i=\text{train,valid,test}}$. $\mathcal{Q}_{\text{train}}$ is used in Stage 1 while $\mathcal{Q}_{\text{train}}$ and $\mathcal{Q}_{\text{valid}}$ are used in Stage 2. $\mathcal{Q}_{\text{test}}$ serves as the validation to test preference generalizability.

### 3.2 ATTRIBUTES AS CONTROL HANDLES

Language models, as next-token predictors, lack intrinsic preferences like humans; their outputs are governed by statistical associations learned from text corpora and shaped by the user prompts. The prompt space is far too vast to explore in an unstructured manner. To address this, we adopt a structured prompting strategy based on *attributes*—interpretable factors, such as ideology, dispositions, or values, whose variation is likely to influence preference expression. P2P treats attributes as *control handles* for steering model behavior.[5]

To mitigate the sparsity of survey data, we steer models through post-training conditioning on a preset *attribute bank* that draws inspiration from theories across diverse social science disciplines, such as economics, political science, management science, and psychology. The bank is organized hierarchically: individual attributes are grouped into **modes**, each representing a coherent subject

---

[5]This contrasts from the human case, where such attributes are typically viewed as reflections of latent preferences. P2P's active endowment generation deliberately leverages this inversion: rather than inferring attributes from preferences, it uses attributes to shape the model's expressed preferences.

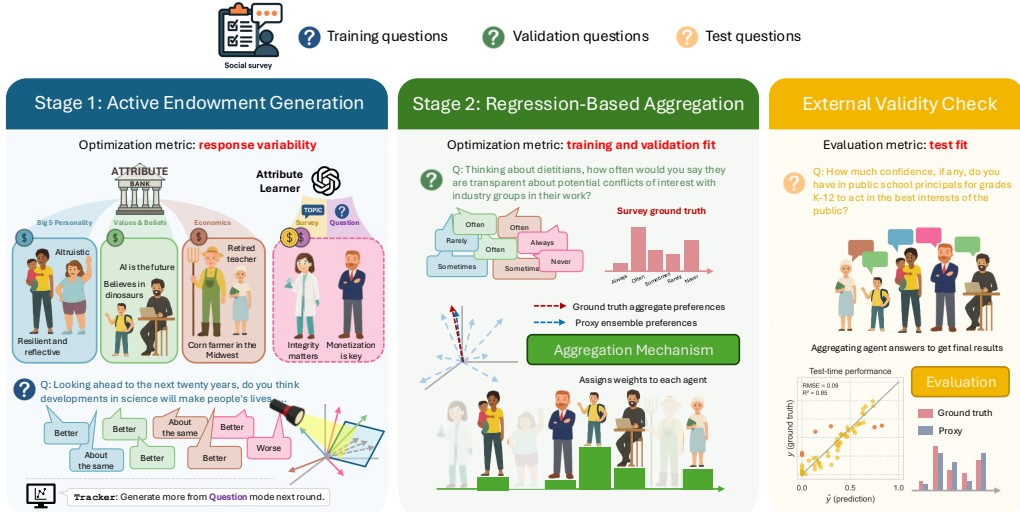

Figure 2: **System Overview of the Prompts to Proxies (`P2P`) Framework.** `P2P` operationalizes preference alignment as a two-stage process inspired by revealed preference theory. In **Stage 1**, active endowment generation uses a structured attribute bank and entropy-guided adaptive sampling strategies to construct diverse agent personas, called endowments. In **Stage 2**, regression-based aggregation assigns weights to reconstruct population-level preference patterns. An external validity check accesses how well the weighted ensemble predicts aggregate responses on held-out questions, assessing out-of-sample generalization.

area (e.g., cognitive biases, political ideology). These modes are further clustered into higher-level **templates**, which reflect different design purposes or theoretical orientations.

We define three primary templates. The *core* template contains general demographic descriptors and serves as the default baseline. The *thematic* template includes topic-specific modes such as economics, politics, culture and history. The *theoretical* template groups attributes derived from canonical frameworks such as Maslow's hierarchy of needs (Maslow, 1943) and the Big Five personality traits (John & Srivastava, 1999). A condensed summary of these templates, representative modes, and attributes is shown in Table 6 (Appendix D.1). These templates and modes are not fixed: they serve as a structured starting point but are fully extensible. Users can define new templates, introduce domain-specific modes, or modify existing attribute sets to suit different experimental contexts. By sampling and varying attributes from this bank, we create prompts that instantiate proxy agents at different locations of the latent model preference space.

While the attribute bank ensures structured coverage, we also enable dynamic discovery through two *freeform templates* that are specific to the survey under analysis. The **survey** template prompts an LLM *attribute learner* to extract decision-making factors from the overall survey topic. The **question** template derives attributes for a specific survey question. These attributes are added to the bank only for the survey run. This hybrid approach balances theory- and data-driven methods, enabling human-AI co-discovery of key preference descriptors.

With the attribute bank in place, we build an endowment generator using a dedicated LLM termed *endowment model* that takes a set of attributes as input and produces diverse agent profiles by varying the instantiation of each selected attribute.

### 3.3 STAGE 1: ACTIVE ENDOWMENT GENERATION

To construct a diverse preference basis without exhaustively enumerating attribute combinations, we create an active endowment generation procedure to operate under resource constraints. It proceeds in iterative steps. At each step, new endowments are sampled, instantiated as agents to elicit responses to survey questions, and evaluated based on their contribution to response diversity. The

remainder of this section describes the key components of this procedure, including our variability scoring strategy, adaptive sampling steps, and entropy-based patching mechanisms.

**Tracking response diversity with variability score**     To measure preference diversity of agents in a revealed sense, we use response variability as an evaluation metric. Specifically, we introduce the notion of **question entropy**:

$$H_i(\mathcal{A}) = \frac{-\sum_{k=1}^{K_i} p_{ik} \log_2 p_{ik}}{\log_2 K_i} \tag{4}$$

where $H_i(\mathcal{A})$ denotes the normalized entropy of question $i$ based on responses from a group of agents $\mathcal{A}$, $K_i$ is the number of unique response options $\{1, \ldots, K_i\}$ for question $i$, and $p_{ik}$ is the empirical proportion of responses selecting option $k$. The denominator normalizes for scale, ensuring that entropy is comparable across questions with different numbers of choices. A high question entropy indicates greater response diversity for that specific question. By restricting the group $\mathcal{A}$ to agents sampled from a particular mode (subject area), we can compute question entropy at the mode level, termed **mode-question entropy**, and compare the response variability across modes.

To turn question entropy into a useful metric for adaptive sampling, we introduce **variability score** for each mode. Formally, it is defined as

$$V(\mathcal{A}_{\mathrm{mode}}) = \sum_{i \in \mathcal{Q}_{\mathrm{train}}} \frac{H_i(\mathcal{A}_{\mathrm{mode}})}{H_i(\mathcal{A}_{\mathrm{total}})}. \tag{5}$$

where $H_i(\mathcal{A}_{\mathrm{mode}})$ denotes the entropy of responses to question $i$ based on the cumulative set of agents from a given mode up to the current generation step, and $H_i(\mathcal{A}_{\mathrm{total}})$ is the corresponding question entropy across all agents generated so far. Computed this way, the variability score favors modes that are capable of generating endowments which elicit diverse responses to questions that other modes typically fail to diversify. In P2P, we task a dedicated class object, `Tracker`, to compute the variability score for each mode after each generation round. The scores are then passed through a softmax function to produce a probabilty distribution over modes, which is used to guide endowment sampling in the next generation step.

**Adaptive sampling and update steps**     In P2P, endowment generation takes several steps (see Algorithm 1 in Appendix D.3). At the outset of the generation, an attribute learner is invoked to infer attributes from the survey based on the training questions, yielding the survey template on top of the existing templates in the attribute bank. The system then enters an initial sampling stage, drawing an equal number of endowments from each mode. The endowments are then used to instantiate agents to elicit responses to training questions, $\mathcal{Q}_{\mathrm{train}}$. The collective responses are then passed to `Tracker` for the computation of the variability scores and the sampling probabilities for the next update step (see Algorithm 2 in Appendix D.3). At the next update step, endowment budget allocation follows the multinomial distribution. A higher probability exploits the top-performing mode, while randomness allows exploration of its under-performing peers.

**Patching low-entropy questions**     In addition to the existing trade-off between exploitation and exploration, we adopt an additional patching strategy, which we call *question patching*. At the end of each update step, as the `Tracker` computes the variability scores for each existing mode, it also keeps track of the questions with the lowest entropy values. When question patching is activated, the attribute learner will be called in the next generation step to exclusively infer attributes on each of the lowest-$k$ entropy questions, where $k$ is user-defined. The split of the endowment budget between the active generation and patching generation is governed by a user-specified ratio.

To avoid wasting resources on persistently low-entropy and to promote cross-pollination between modes, we additionally introduce a *mixed mode* strategy. It is triggered after a question appears $t$ times in the lowest-$k$ entropy list, where $t$ is a user-defined threshold. When in effect, the strategy relocates the assigned endowment budge to a new mixed mode, which combines attributes from the question with those of the current top-performing mode. This encourages diversity injection from high-performing modes while maintaining a question-specific focus.

The adaptive sampling cycle continues until the generation budget $N_A$ (an upper bound on the maximum number of agents allowed) is exhausted. By iteratively targeting low-diversity questions,

expanding the attribute space and promoting cross-pollination between modes, active endowment generation progressively increases response diversity while preserving coverage over previously explored regions of the preference space (cf. Figure 12 in Appendix C.1).

### 3.4 STAGE 2: REGRESSION-BASED AGGREGATION

With the proxy basis assembled, P2P enters the second stage: regression-based aggregation. The goal at this stage is to find a functional basis from the proxy basis to reconstruct the observed population preference. Depending on the endowment budget $N_A$ and the total number of trainable questions $\mathcal{Q}_{\text{train}} \cup Q_{\text{valid}}$, we could end up in a situation where there are more variables (proxy agents) than observations (trainval questions). Consequently, a variable selection algorithm is vital. Besides, even with $|\mathcal{Q}_{\text{train}} \cup Q_{\text{valid}}| >> N_A$, variable selection allows us to retain a parsimonious agent ensemble, saving inference costs on further simulations. We consider regression methods with an added L1 penalization term and use cross-validation (CV) for hyperparameter tuning. The objective of the learning problem can be stated as

$$\min_{\mathbf{w}} \sum_{i \in \mathcal{Q}_{\text{train}}} \sum_{k=1}^{K_i} \left( r_{ik} - \sum_{j=1}^{N_A} w_j d_{j,ik} \right)^2 + \alpha \Omega(\mathbf{w}), \tag{6}$$

$$\text{s.t.} \quad \sum_{k=1}^{K_i} \sum_{j=1}^{N_A} w_j d_{j,ik} = 1 \quad \forall i \in \mathcal{Q}_{\text{train}}, \tag{7}$$

where $\mathbf{w} \in \mathbb{R}^{N_A}$ is the weight vector for the agents, $r_{ik}$ denotes the ground-truth proportion of responses selecting option $k$ for question $i$, $d_{j,ik}$ is a dummy response variable equal to 1 if agent $j$ selects option $k$ and 0 otherwise, and $\alpha$ is the hyperparameter for the penalization. When $\Omega$ consists of only a L1 norm, the formulation amounts to constrained lasso. When $\Omega$ subsumes an additional L2 norm, it becomes constrained elastic net. Our simulation studies on synthetic ground truth data indicate that constrained lasso faithfully recovers the ground-truth agents from a large candidate pool under an adequate number of observations (Appendix B.1), and it is potent in selecting a proxy ensemble to recover ground truth even when the number of proxy agents is small (Appendix B.2). Both results corroborate using it as a selection backbone. Our simulation study also investigates the relationship between entropy and prediction fit—we leave the details to Appendix B.3.

## 4 TAKING P2P TO THE WILD

To empirically test the preference reconstruction theory implemented via P2P, we use datasets from the American Trends Panel (ATP)(Pew Research Center), a nationally representative U.S. survey conducted by Pew Research Center in discrete waves covering a wide range of topics. We obtain the digitalized versions of these datasets from the OpinionsQA benchmark (Santurkar et al., 2023), which repurposed ATP responses to study value pluralism and misalignment in language models. Without further specification, the results shown are conducted on Wave 42, which probes trust in science by asking respondents about scientific competence, integrity, and public accountability. The wave includes responses from approximately 4464 U.S. adults (Pew Research Center, 2019) and the cleaned dataset contains 128 questions. When results use additional waves, we explicitly specify the corresponding dataset configuration.

### 4.1 BASELINES

To benchmark P2P's performance, we created two baselines with different endowment generation logic, which are then fed through the same regression-based aggregation stage as P2P for proxy selection and test set evaluation. The prompts for P2P and the Vanilla baseline can be found in Appendix D.2: **Vanilla baseline** We adopt a simple endowment generator prompted directly to generate diverse agent profiles conditioned only on the given survey topic (trust in science), without the additional attribute bank or entropy-based sampling strategy. In total, 300 endowments are generated. **PERSONA** We use the demographically conditioned PERSONA dataset (Castricato et al., 2025), which contains 1000 persona profiles. We randomly sample a subset of 300 persona profiles derived from U.S. census data. to use as endowments for the agents.

## 4.2 EXPERIMENTAL SETTING

For the experiment, we split the questions according to a $7 : 1.5 : 1.5$ ratio into the training, validation, and test sets. For the endowment generation stage, we set the endowment budget to be 300, with 10 endowments generated for each mode during initial sampling. Active endowment generation is run in 10 update steps with the preset attribute bank, question patching (lowest 3; 75% endowment budget), and 10 attributes drawn for each endowment generation. At the end of the generation, 5 questions are patched, with an additional 5 mixed modes created. Unless otherwise noted, all models use Gemini-2.0-flash as the backend.

## 4.3 MAIN RESULTS

Table 1: Comparison of baseline (Vanilla), PERSONA, and P2P (AEG) models. Reported are average question entropy, test MSE (Lasso and ElasticNet), and generation cost. For each model, we fix the endowment budget at 300. All values are mean $\pm$ std over 3 repeated runs.

| Metric | Vanilla | PERSONA | P2P (AEG) |
|---|---|---|---|
| Avg. Entropy $\uparrow$ | $0.317 \pm 0.0069$ | $0.287 \pm 0.00026$ | $\mathbf{0.445} \pm 0.0256$ |
| Test MSE (Lasso) $\downarrow$ | $0.0285 \pm 0.00490$ | $0.0362 \pm 0.00116$ | $\mathbf{0.00946} \pm 0.00241$ |
| Test MSE (Enet) $\downarrow$ | $0.0286 \pm 0.00476$ | $0.0339 \pm 0.00124$ | $\mathbf{0.00792} \pm 0.00261$ |
| Cost (USD) $\downarrow$ | $\mathbf{0.865} \pm 0.0070$ | $1.713 \pm 0.000016^{*}$ | $0.971 \pm 0.0036$ |

[*] PERSONA cost only covers the response elicitation stage—higher if persona generation tokens are included.

Table 1 shows baseline comparisons. P2P significantly outperforms Vanilla and PERSONA, with mean test MSEs near 0.01 for both Lasso and ElasticNet. Its cost is similar to Vanilla (under 1 USD per run). The PERSONA pipeline (Castricato et al., 2025) requires resampling and re-inference, adding further API usage costs. Further qualitative analyses of entropy changes across updates and regression visualizations are provided in Appendix C.1.

After validating the effectiveness of P2P against the baselines, we present a comprehensive 14-wave ATP to demonstrate its generalizability. Because the data volume increases with 14 waves, we cap active endowment generation at 5 update steps to reduce runtime and cost, while keeping all other hyperparameters unchanged. As shown in Table 2, 12/14 waves achieve MSE below 0.02, with an average MSE of 0.0142 across all 14 waves, confirming generalizability. The reduced number of updates does not materially affect performance, as evidenced by the W42 cell results for P2P in Table 1 versus Table 2. A more detailed analysis of all 14 waves is provided in Appendix G.8.

## 4.4 ABLATION

To understand how different parts of P2P contribute to the overall performance, we conduct three ablation studies, focusing on endowment budget, regression selection and model backends.

**Endowment budget** We run P2P for 9 different endowment budgets, scaled linearly from 130 to 450. We increase the update steps of active endowment generation accordingly to ensure that at each step the number of new endowments generated is fixed at 20. We also include a budget of 110 with no updates as an ablation of adaptive sampling. Table 3 in Appendix C.2 shows that on W42, a higher budget generally improve test performance but gains plateau after about three updates, indicating a moderate budget suffices.

**Regression** To gauge the contribution of the regression-based aggregation module, we ablate it by replacing regression with simple averaging of agent responses, which increases the test MSE from 0.0104 to 0.0254. In essence, regression not only distills the agent collection into a more compact ensemble, but it also plays a crucial role in ensuring the population preference reflected in the survey data are actually learned by the ensemble.

**Model backends** Finally, we test P2P's performance on different model backends, including GPT-4.1-mini, GPT-4.1-nano, Gemini-2.0-flash, Gemini-2.5-flash and a locally hosted Qwen. Perfor-

mance varies substantially, with the best average question entropy (0.471) and test MSE (0.009) achieved by GPT-4.1-mini. Gemini-2.0-flash ranks second on these metrics but is markedly more cost-efficient, requiring only one quarter of the cost of GPT-4.1-mini. Smaller models, like GPT-4.1-nano and Gemini-2.5-flash-lite, perform worse. This aligns with our theoretical intuition: Smaller models may struggle to satisfy Definition 2.2, as their limited expressive capacity constrains $f(\hat{\mathcal{P}})$, reducing the likelihood that it fully covers $\text{conv}(h(\mathcal{P}))$.

Table 2: Performance of `P2P` (5 update steps) on 14 ATP waves, measured by test MSEs for Lasso and ElasticNet. Values are mean over 3 repeated runs, with SD reported in the parentheses below.

| **Wave** *Topic* | W26 *Guns* | W27 *Auto* | W29 *Gender* | W32 *Comm.* | W34 *Biomed* | W36 *Lead.* | W41 *2050* | |
|---|---|---|---|---|---|---|---|---|
| **Lasso MSE** | 0.0098 (.0002) | 0.0218 (.0010) | 0.0156 (.0049) | 0.0160 (.0019) | 0.0111 (.0011) | 0.0191 (.0042) | 0.0126 (.0009) | |
| **Enet MSE** | 0.0098 (.0002) | 0.0238 (.0027) | 0.0156 (.0049) | 0.0160 (.0019) | 0.0111 (.0011) | 0.0187 (.0037) | 0.0126 (.0009) | |
| **Wave** *Topic* | W42 *Science* | W45 *Misinfo* | W49 *Privacy* | W50 *Family* | W54 *Econ* | W82 *Global* | W92 *Poli.* | Avg |
| **Lasso MSE** | 0.0110 (.0003) | 0.0079 (.0023) | 0.0211 (.0037) | 0.0133 (.0039) | 0.0144 (.0018) | 0.0083 (.0016) | 0.0168 (.0029) | **0.0142** (.0044) |
| **Enet MSE** | 0.0101 (.0006) | 0.0079 (.0023) | 0.0211 (.0037) | 0.0133 (.0039) | 0.0146 (.0018) | 0.0083 (.0016) | 0.0173 (.0023) | **0.0143** (.0048) |

**`P2P` as a means to an important end** At a broader level, our framework operationalizes pluralistic alignment (Sorensen et al., 2024b). Its structured use of attribute-based control enables the representation of diverse value systems, while its regression-based weighting mechanism supports distributional pluralism without enforcing demographic fidelity. Crucially, `P2P` demonstrates that steerability and pluralism are not separate challenges but deeply intertwined: the ability to steer model behavior through promptable attributes is what enables meaningful diversity in aggregate preference reconstruction.

## 5 CONCLUSION

This work establishes a theoretical foundation for viewing preference alignment as a problem of preference reconstruction. Inspired by revealed preference theory, our approach formalizes alignment as a two-stage process: first, constructing a functional basis of proxy agents using attribute-guided prompting; and then, recovering aggregation weights through supervised learning. In doing so, our work offers the first tractable implementation of pluralistic alignment, along with quantitative metrics to assess both diversity and fidelity.

Beyond its theoretical contributions, this framework offers a practical platform for interdisciplinary research. In the social sciences, it has the potential to support survey design, question testing, and nonresponse mitigation, serving as a cost-effective complement to traditional data collection. For researchers in LLM alignment, the modularity of our system provides a controlled setting to evaluate prompt engineering strategies and their influence on agent diversity and downstream performance. More broadly, it lays the groundwork for agentic systems that enable value-sensitive modeling across disciplines. Notably, `P2P` repositions social survey data not as training material, but as a guiding signal for culturally grounded model alignment.

## ETHICS STATEMENT

Our work utilizes and cites survey datasets in the public domain that have been anonymized in advance, with sensitive personally identifiable information removed. Although the large language models (LLMs) used as backends in the study have been trained to align with common human values, we recognize that they may still propagate social biases and stereotypes, especially under adversarial or intentional prompting. The endowment profiles generated in the first stage of P2P are diverse in nature, while some generated profiles may contain opinionated and potentially offensive content.

The goal of our framework is methodological: to study preference reconstruction at the aggregate level rather than through individual personalization, thereby reducing the risk of profiling. We see potential for positive impact on improving pluralistic alignment and enabling the principled use of AI in social science research. Meanwhile, we acknowledge that any downstream deployment of P2P must strictly adhere to fairness and ethical safeguards. Finally, we stress that the agent ensemble is designed as an auxiliary tool to understand human choice patterns and should not replace actual humans in critical policy-making scenarios.

## REPRODUCIBILITY STATEMENT

We provide an anonymized, dated version of our codebase (prepared for a prior submission) that includes the full pipeline, configuration files, and instructions to reproduce all core experiments. This ensures replication of the main tables and figures.Additional experiments were run on an updated internal codebase supporting different model backends; for fairness and anonymity, these engineering improvements are not included here but will be released in the latest version of the code upon acceptance.

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

## A PROOFS

**Theorem 0** (Existence of a representative agent). Let $\widetilde{\mathcal{P}}$ be a latent preference space (regardless of human or model), with an individual preference denoted by $\tilde{p} \in \widetilde{\mathcal{P}}$. Let $g$ be the map from $\widetilde{\mathcal{P}}$ to the observed response space $\widetilde{\mathcal{R}} \subset \mathbb{R}^n$. Then the following two statements are equivalent:

1. The image $g(\widetilde{\mathcal{P}}) \subset \widetilde{\mathcal{R}}$ is convex.

2. For any aggregate response $\tilde{r}_{\text{pop}} = \sum_{i=1}^{I} w_i \tilde{r}_i$ with $\{w_i\}_{i=1}^{I}$ convex weights and $\tilde{r}_i = g(\tilde{p}_i)$, there exists $\tilde{p}_{\text{pop}} \in \widetilde{\mathcal{P}}$ such that $g(\tilde{p}_{\text{pop}}) = \tilde{r}_{\text{pop}}$.

*Proof.* First, we prove $(1) \implies (2)$.

By definition, $\tilde{r}_i = g(\tilde{p}_i) \in g(\widetilde{\mathcal{P}})$ for all $i \in \{1, \ldots, I\}$. Since $\{w_i\}_{i=1}^{I}$ are convex weights ($\sum_{i=1}^{I} w_i = 1$ and $w_i \geq 0$ for all $i \in \{1, \ldots, I\}$), and $g(\widetilde{\mathcal{P}})$ is convex, it follows that

$$\sum_{i=1}^{I} w_i \tilde{r}_i \in g(\widetilde{\mathcal{P}}).$$

By the definition of image, it follows that there exists $\tilde{p}_{\text{pop}} \in \widetilde{\mathcal{P}}$ such that

$$g(\tilde{p}_{\text{pop}}) = \sum_{i}^{I} w_i \tilde{r}_i \triangleq \tilde{r}_{\text{pop}}.$$

Claim: $(2) \implies (1)$.

Let $\{\tilde{r}_i\}_{i=1}^{I}$ denote any finite collection of observed responses with $\tilde{r}_i \in g(\widetilde{\mathcal{P}})$, and $\{\lambda_i\}_{i=1}^{I}$ be any convex weights. Consider $\tilde{r}_{\text{agg}} = \sum_{i=1}^{I} \lambda_i \tilde{r}_i$. By (2), there exists $\tilde{p}_{\text{agg}} \in \widetilde{\mathcal{P}}$ such that

$$\tilde{r}_{\text{agg}} = g(\tilde{p}_{\text{agg}}) \in g(\widetilde{\mathcal{P}}). \tag{8}$$

Therefore, $g(\widetilde{\mathcal{P}})$ is convex by definition. $\square$

**Lemma 1** (Sufficient condition for representative agent). If the decision-making mechanism $g$ maps a preference into a convex subset of a probability simplex $\Delta^{n-1}$, then any aggregate of individual responses is realizable by a representative agent.

*Proof.* A convex subset of a probability simplex is by definition convex. The result follows from the theorem. $\square$

The implications of this lemma are two-fold:

1. In human decision-making, the condition that $g(\widetilde{\mathcal{P}})$ is convex is less likely to be true unless we assume stochastic choice models with stringent functional forms (cf. Kirman, 1992; Jackson & Yariv, 2019).

2. In LLM decision-making, the condition that $g(\widetilde{\mathcal{P}})$ is convex is more likely to hold is more likely to be hold assuming that the model has considerable steerability. (E.g., through finetuning or prompting, we can effectively change the latent model preference embeddings and influence the output probability distributions.)

**Definition A.1.** Human preference is *learnable in the revealed sense* if $h(\mathcal{P}) \subset f(\hat{\mathcal{P}})$.

**Definition A.2.** Human population preference is *learnable in the revealed sense* if $\text{conv}(h(\mathcal{P})) \subset f(\hat{\mathcal{P}})$.

**Proposition 1.** If human population preference is learnable in the revealed sense, then human preference is learnable in the revealed sense.

*Proof.* The proof follows from the fact that $h(\mathcal{P}) \subset \mathrm{conv}\,(h(\mathcal{P}))$. $\square$

**Theorem 1** (Representative revealed preference learning). If human population preference is learnable in the revealed sense, it can be learned by a representative LLM agent with model preference $\hat{p}_{\mathrm{pop}} \in \widehat{\mathcal{P}}$.

*Proof.* Since human population preference is learnable in the revealed sense,

$$\mathrm{conv}\,(h(\mathcal{P})) \subset f(\widehat{\mathcal{P}}).$$

This means there is a convex subset of the image $f(\widehat{\mathcal{P}})$.

Denote the preimage of $\mathrm{conv}\,(h(\mathcal{P}))$ as $\widehat{\mathcal{P}}_s$, i.e., $\widehat{\mathcal{P}}_s := f^{-1}\,(\mathrm{conv}\,(h(\mathcal{P})))$, with $\widehat{\mathcal{P}}_s \subset \widehat{\mathcal{P}}$. Then it follows from Theorem 0 that for any $r_{\mathrm{pop}} \in \mathrm{conv}\,(h(\mathcal{P})) \subset f(\widehat{\mathcal{P}})$, there exists $\hat{p}_{\mathrm{pop}} \in \widehat{\mathcal{P}}_s \subset \widehat{\mathcal{P}}$ such that $f(\hat{p}_{\mathrm{pop}}) = r_{\mathrm{pop}}$. $\square$

**Theorem 2** (Ensemble revealed preference learning). If human population preference is learnable in the revealed sense, it can be learned by an ensemble of LLM agents with model preferences $\{\hat{p}_i\}_{i=1}^I$.

*Proof.* Since human population preference is learnable in the revealed sense,

$$\mathrm{conv}\,(h(\mathcal{P})) \subset f(\widehat{\mathcal{P}}).$$

For any $r_{\mathrm{pop}} \in \mathrm{conv}\,(h(\mathcal{P}))$, it follows from the definition of convex hull that there exists a finite set of $\{r_i\}_{i=1}^I$ with $r_i \in h(\mathcal{P})$ and convex weights $\{w_i\}_{i=1}^I$ such that

$$r_{\mathrm{pop}} = \sum_{i=1}^I w_i r_i.$$

Since $r_i \in h(\mathcal{P}) \subset \mathrm{conv}\,(h(\mathcal{P})) \subset f(\widehat{\mathcal{P}})$, there exists $\hat{p}_i$ such that $f(\hat{p}_i) = r_i$.

It follows that

$$r_{\mathrm{pop}} = \sum_{i=1}^I w_i f(\hat{p}_i) \in \mathrm{span}\,\{f(\hat{p}_i)\}_{i=1}^I.$$

$\square$

**Theorem 3** (Preference reconstruction theory). If human population preference is learnable in the revealed sense, the ensemble of LLM agents that can learn this is not unique.

*Proof.* If human population preference is learnable in the revealed sense, we know from Theorem 2 that there exist $\{\hat{p}_i\}_{i=1}^I$ such that

$$r_{\mathrm{pop}} = \sum_{i=1}^I w_i r_i = \sum_{i=1}^I w_i f(\hat{p}_i). \tag{9}$$

However, since $r_1 \in h(\mathcal{P}) \subset \mathrm{conv}(h(\mathcal{P}))$, we know there exists a finite set of $\{r_k^*\}_{k=1}^K$ with $r_k^* \in h(\mathcal{P})$ and convex weights $\{\lambda_k\}_{k=1}^K$ such that

$$r_1 = \sum_{k=1}^K \lambda_k r_k^*.$$

Plug this into (9) and we get

$$r_{\mathrm{pop}} = \sum_{k=1}^K w_1 \lambda_k r_k^* + \sum_{i=2}^I w_i r_i.$$

Since $r_k^* \in h(\mathcal{P}) \subset \mathrm{conv}\,(h(\mathcal{P})) \subset f(\widehat{\mathcal{P}})$, there exists $\hat{p}_k^*$ such that $f(\hat{p}_k^*) = r_k^*$.

In conclusion,

$$r_{\mathrm{pop}} \in \mathrm{span}\{f(\hat{p}_1^*), \ldots, f(\hat{p}_K^*), f(\hat{p}_2), \ldots, f(\hat{p}_I)\}$$

Note that in the proof, we have used the expansion technique. A contraction technique also works. In the extreme case, we get the representative agent, which is an ensemble of one agent. $\square$

# B  SIMULATION STUDIES

In order to gauge the validity of the proposed preference model and the factors affecting alignment performance, we conduct three simulation studies where we have the privilege of ground-truth knowledge.

**Simulation Design**  For the simulation studies, we use the American Trends Panel (ATP) Wave 42 (W42) as the survey source, covering the topic of public trust in science and scientists. We run the active endowment generation on the survey to get 300 persona endowments, and partition the survey questions into training, validation and test sets by the 0.7:0.15:0.15 ratio. We subsequently simulate the responses to the survey questions using endowed GPT-4o agents as the synthetic respondents. The ATP W42 dataset has in total 129 questions, which after binarization leads to 517 observations—85% of these observations form the train-and-validation (trainval) set and the rest 15% is used to test the predictive accuracy of the selected agent ensemble.

**Construction of the Ground Truth Data**  To construct the ground truth data, we sample a fraction of the endowments as the ground truth and randomly assign weights to these endowments, which represent the proportions of the associated personas in the ground truth population. The ground truth data—percentages of individuals choosing difference options for a given question—are computed using weighted averages of responses provided by the ground-truth agents. The unselected endowments are used to construct the pool of proxy agents.

**Regression Method**  For the simulation studies, we choose constrained lasso as our main regression alignment method for investigation. As a secondary choice, constrained elastic net is an extension of constrained lasso that trades off between the L1 and L2 penalty terms. It is expected to have more stable performance when the features (agent responses) are highly correlated. Our initial simulations indicate that constrained elastic net demonstrates near identical performance to constrained Lasso, partly because the latter is a special case of the former and we are using cross-validation to select the best penalization parameters. Therefore, for the sake of building intuitions, we focus on constrained lasso in the simulations and expect the results to be largely indicative of constrained elastic net's performance.

## B.1  RECOVERING GROUND TRUTH AGENTS WITH CONSTRAINED LASSO

As a first test of constrained lasso as a variable selection algorithm, we aim to evaluate its ability to recover ground-truth agents from a broader candidate pool. To render the selection task non-trivial, we set the prevalence of the ground-truth agents to 0.3—out of 10 selectable agents, only 3 contribute to the construction of the ground-truth signal. As the proxy agents dominate the agent pool, an adequate number of observations becomes essential for Lasso to reliably identify the true contributors. As a key metric, we define **observation-to-agent ratio (OAR)** to be the ratio of the number of training and validation observations to the number of selectable agents. By default, the OAR for our simulation baseline is 1.203.

To develop a panoramic view of constrained lasso's ability in ground truth recovery, we conduct multiple simulation rounds under varying OARs. We generate different OAR variations using two complementary subsampling strategies:

- **Subsampling endowments**. To increase the OAR, we adopt a subsampling strategy on endowments, where we subsample from the pool of selectable endowments, varying the total number of candidate agents while keeping the proportion of ground-truth agents fixed at 30%. The ground-truth weights are renormalized to sum to 1. We vary the subsample fraction from 0.5 to 1 in 20 equally spaced steps.
- **Subsampling questions**. To decrease the OAR, we subsample survey questions while keeping the original train-validation-test split and ratio unchanged. This mimics a setting with reduced behavioral signal for inference. As in the endowment-based approach, we use 20 subsample fractions ranging from 0.5 to 1.

We repeat each subsampling procedure 10 times to construct the final dataset used in our simulation study. Figure 3 shows the trends of the mean squared error of the test set and the precision—that is,

the proportion of selected agents who possess ground-truth endowments—across different OARs. To further illustrate the dynamics of selection, Figure 4 presents three representative snapshots of simulation runs, corresponding to low, medium, and high OAR settings.

As the figures indicate, constrained lasso's ability to recover ground-truth agents from the candidate pool improves with a higher observation-to-agent ratio. This is intuitive as more observations offer a richer signal of both individual-level idiosyncrasies and aggregate preferences. With abundant behavioral cues, the selection algorithm is better equipped to distinguish true contributors from correlated siblings. In contrast, when observations are sparse, Lasso tends to relinquish its sharpness as a selection tool and instead defaults to identifying a functional basis that best explains the limited data. The mild decline in predictive accuracy under low OAR regimes suggests that this functional basis still approximates the aggregate preference reasonably well, albeit imperfectly.

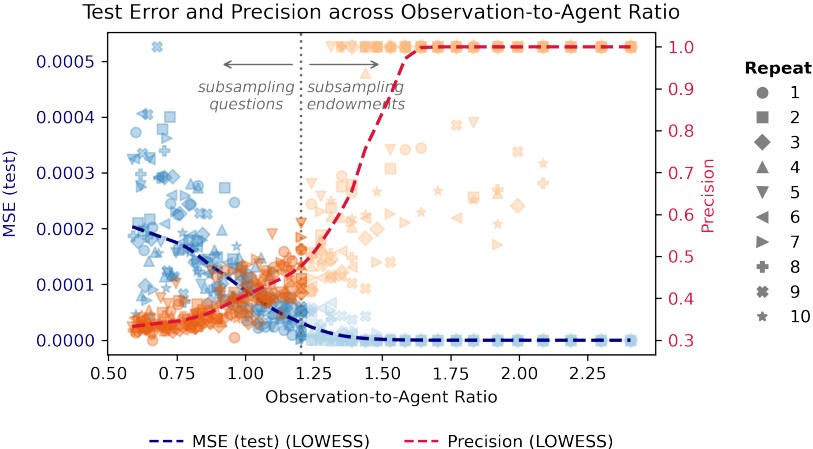

Figure 3: Mean Squared Error (Test Set) and Precision by Different Observation-to-Agent Ratio (OAR) for Constrained Lasso. Each simulation is repeated 10 times with different random seeds. Subsampling of endowments is done with fixed ground-truth-to-proxy ratio. Subsampling of responses is done with fixed train-valid-test split and ratio.

## B.2 EMULATING GROUND TRUTH RESPONSES WITH PROXY AGENTS

While the first simulation exercise has readily allured to constrained lasso's ability to find a functional basis to explain the observed data, in this second simulation study we extend the investigation by removing the ground-truth agents from the selectable agent pool. Under this setting, constrained lasso needs to fully rely on proxy agents to recover the ground-truth patterns. We analyze two distinct scenarios:

- **Subsampling proxies.** We fix the total number of observations and reduce the proxy pool by retaining only a fraction as selectable agents. We sweep across the fractions from 0.05 to 1 in 20 equal spaced steps.

- **Subsampling training and validation questions.** We fix the proxy pool but retain a fraction of training and validation observations. Similar to subsampling proxies, we construct 20 simulation rounds gradually raising the retained fraction from 0.05 to 1.

Like in Simulation Study 1, we repeat each simulation 10 times using different random seeds to form the final results. In Figure 5, the left panel displays the mean squared errors (MSE) and coefficients of determination ($R^2$) for the subsampling proxies scenario, while the right panel showcases the associated metrics for the subsampling training and validation questions scenario.

Notably, constrained lasso exhibits robust performance across all simulated conditions. Even with only 10 selectable proxy agents, it achieves a mean test MSE of $0.0067$, yielding an $R^2$ of $0.93$. Predictive accuracy improves as we increase the number of selectable proxy agents, plateauing at

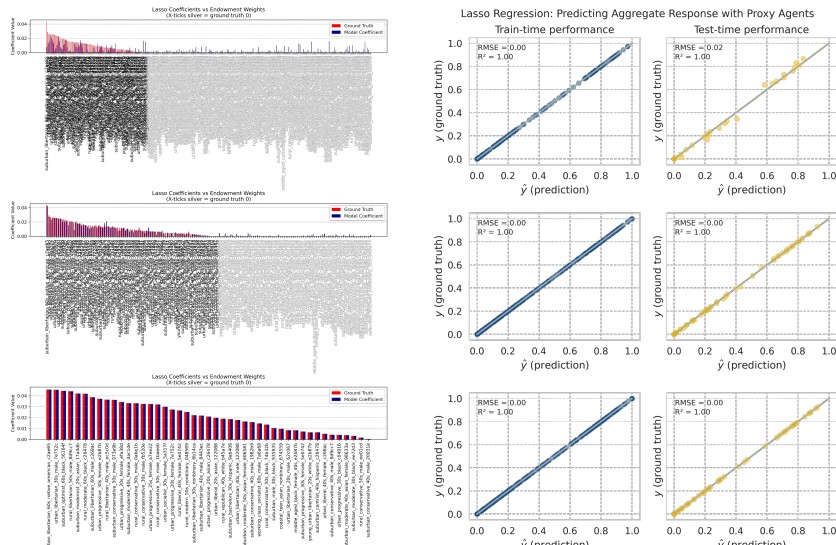

Figure 4: Snapshots of Weight-vs-Coefficient Comparison and Prediction Performance. Each row represents a simulation run. The red bars on the left panel represent ground truth agent weights used to generate the aggregate data and the blue bars Lasso coefficients. The observation-to-agent ratios for the top, middle and bottom panels are: 0.617, 1.203, 2.407. Lasso's ability to recover the ground truth agents from the agent pool degrades with lower observation-to-agent ratio, but its predictive accuracy only drops mildly.

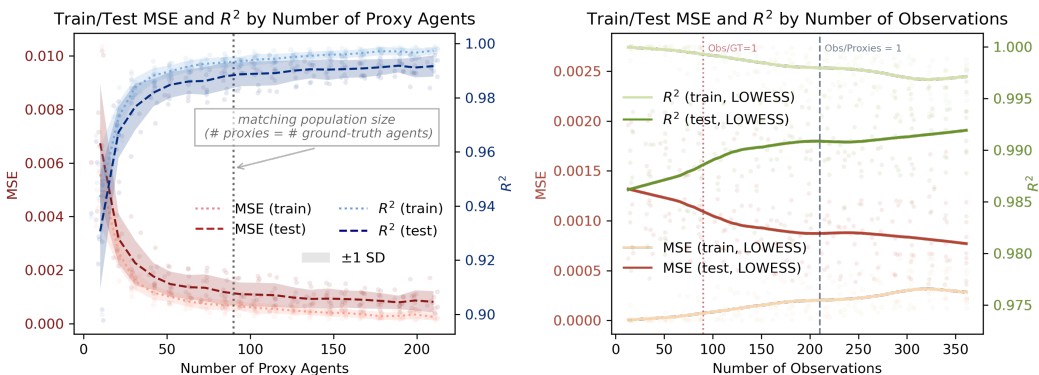

Figure 5: Left: Mean Squared Errors and Coefficients of Determination ($R^2$) by Number of Proxy Agents. Right: Mean Squared Errors and Coefficients of Determination ($R^2$) by Number of Observation. Each simulation is repeated 10 times with different random seeds.

0.99. In comparison, decreasing the number of observations has a lesser effect on predictive performance: in the recorded run with the lowest number of observations at 13, constrained lasso attains a test MSE of 0.0019 and an $R^2$ of 0.98. Figure 6 displays representative snapshots of the simulation runs.

The high predictive accuracy of the simulation runs indicates that constrained lasso can efficiently find a functional basis from a select few agents and determine the agent weights to emulate the ground-truth data patterns based on limited observations. This is in fact not surprising because in the initial assignment of the endowments into the ground truth and proxy groups we have used a random partition. As informed by our proposed behavioral preference model, this causes the ground truth vectors—used in the simulation study to form the aggregate preference—and proxy vectors to span the same preference space. As the preference space is not directly accessible, we can derive a partial gauge through the observed agent responses. If the responses of a group of agents have higher

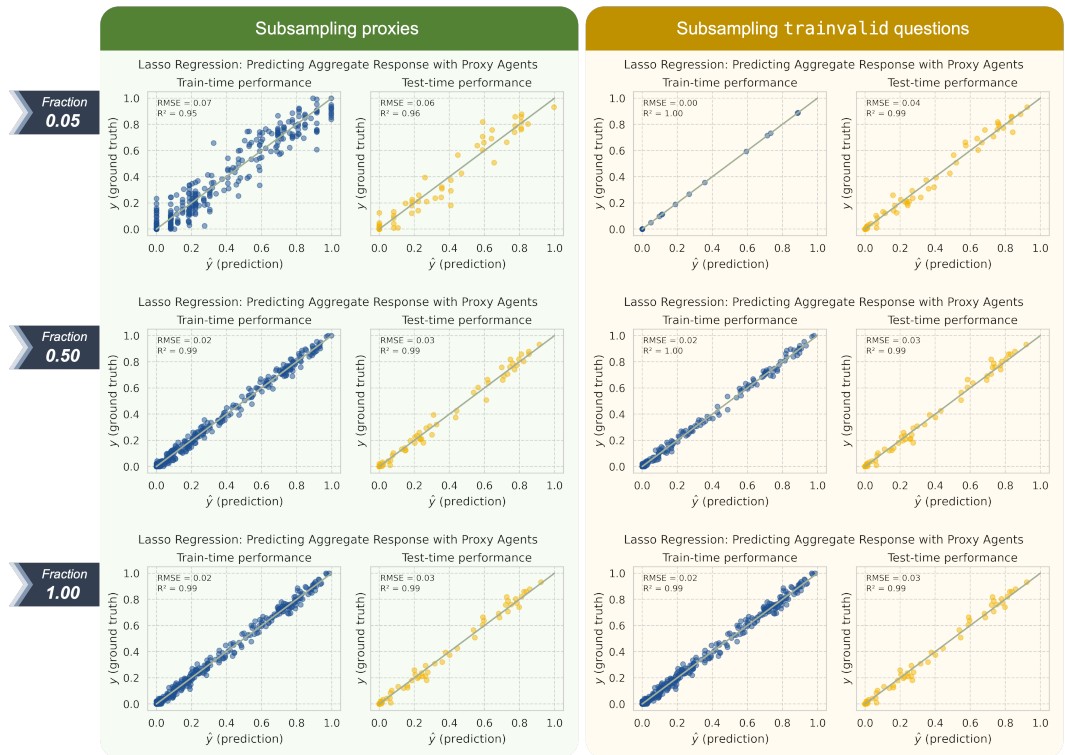

Figure 6: Snapshots of Prediction Performance for Simulation Study 2.

variability, it is an indication that the preference space they span is large. Conversely, if agents tend to agree on their responses to different questions, it is a signal that their associated vectors in the preference space might overlap in sub-dimensions, leading to a pool generalization ability.

To effectively measure response variability, in the paper we introduce the notion of **question entropy**:

$$H_i(\mathcal{A}) = \frac{-\sum_{k=1}^{K_i} p_{ik} \log_2 p_{ik}}{\log_2 K_i} \tag{10}$$

where $H_i(\mathcal{A})$ is the normalized entropy of question $i$ based on responses from a group of agents $\mathcal{A}$, $K_i$ denotes the number of unique response options $\{1, \ldots, K_i\}$ for question $i$, and $p_{ik}$ represents the empirical proportion of responses selecting option $k$. The normalization factor ensures comparison across questions. A high question entropy indicates that the responses are highly varied for this question.

For the simulation studies, we define **group entropy** as the average question entropy computed using responses from that group:

$$H(\mathcal{A}) = \frac{1}{N} \sum_{i=1}^{N} H_i(\mathcal{A}). \tag{11}$$

The group entropy is calculated using responses to all survey questions, regardless of the split. Based on it, we introduce the notion of **Entropy Coverage Ratio (ECR)** defined as the ratio between the group entropy of proxy agents and that of the ground-truth agents:

$$ECR = \frac{H(\mathcal{A}_{\text{proxy}})}{H(\mathcal{A}_{\text{gt}})}. \tag{12}$$

An ECR lower than 1 indicates that the proxy agents have less response variability than the ground truth, while an ECR greater than 1 indicates the opposite.

The left panel of Figure 7 shows the distribution of entropy coverage ratios (ECRs) for the two analyzed scenarios. The mass is concentrated around $1.0$, corroborating the earlier observation

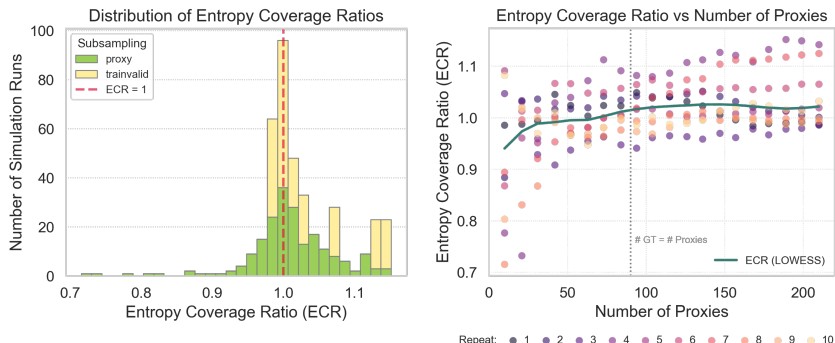

Figure 7: Entropy Coverage Ratio Diagnostics for Simulation Study 2.

that random partitioning tends to preserve entropy structure. The right panel plots ECR against the number of proxies for the proxy-subsampling scenario. The relationship between ECR and the number of proxies is subtly positive: when the number of proxies is small, the likelihood of drawing an unrepresentative sample increases. As the sample size grows, group-level entropy converges toward the population-level entropy—an empirical manifestation of the law of large numbers.

## B.3 RESPONSE VARIABILITY MATTERS

In this final simulation study, we analyze the effect of the entropy coverage ratio on predictive accuracy for constrained lasso.

Figure 8 displays the group entropies for THE W42 endowments organized by modes—we call them **mode entropies** in the endowment generation logic. We subsume them further into three different tiers according to the group entropy value: in total, we have 71 endowments in the low entropy tier, 92 in the mid entropy tier, and 137 in the high entropy tier.

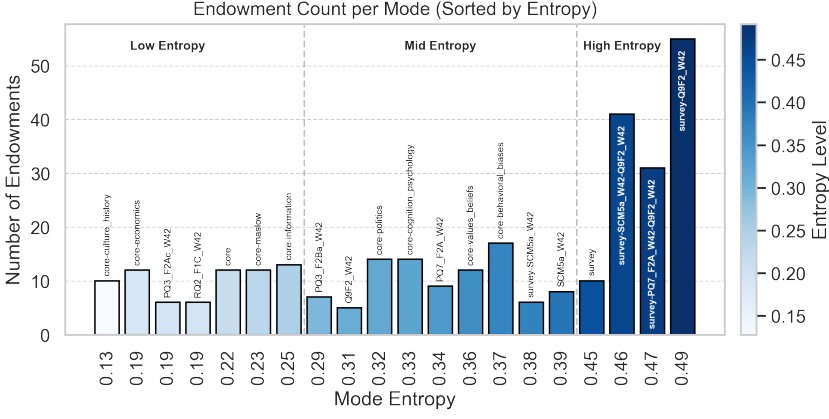

Figure 8: Endowment Count per Mode (Sorted by Entropy).

To cover a wide range of ECR ratios, we design the following simulation strategy:

1. **Ground Truth Construction.** For each simulation round, pick one of the three tiers as the ground-truth tier and sample 30 endowments from the tier to construct the ground-truth agents and assign ground-truth weights. The unselected endowments from the tier are joined with the endowments from the other two tiers to form the pool of proxy candidates.

2. **Proxy Agents Construction.** For each simulation run, we select 30 proxy agents to form the proxy pool. We begin by selecting agents from the lowest entropy modes (ensuring that the cumulative candidate endowments surpass 30), and in each consecutive run, include the

next mode with higher entropy into the selectable modes. We sweep across the modes by increasing mode entropy.

3. **Proxy Agents Selection and Response Emulation.** In each simulation run, after the proxy pool has been constructed, we use constrained lasso to select proxy agents to emulate the observed aggregate response data.

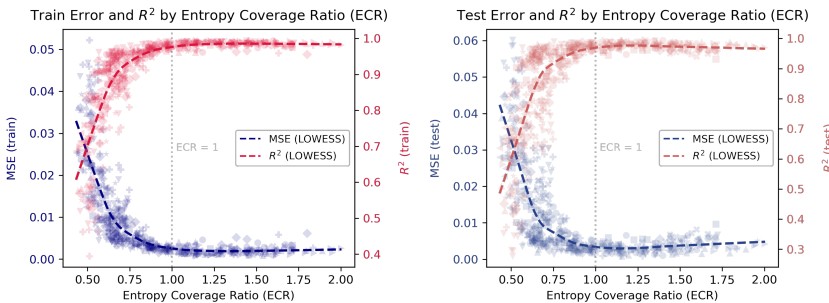

Figure 9: Mean Squared Error and Coefficient of Determination ($R^2$) by Entropy Coverage Ratio (ECR) for the trainval set (left) and test test (right). Entropy Coverage Ratio is defined as the ratio of the average response entropies of the proxies and ground-truth agents. LOWESS curves are fit using 10 repetitions.

We repeat each simulation 10 times to form the final results. Figure 9 visualizes the train and test performance of the constrained lasso by the entropy coverage ratio. As the ECR increases, the predictive accuracy improves at both the training and the test times, lending further credence to the analysis based on the preference model in the second simulation study.

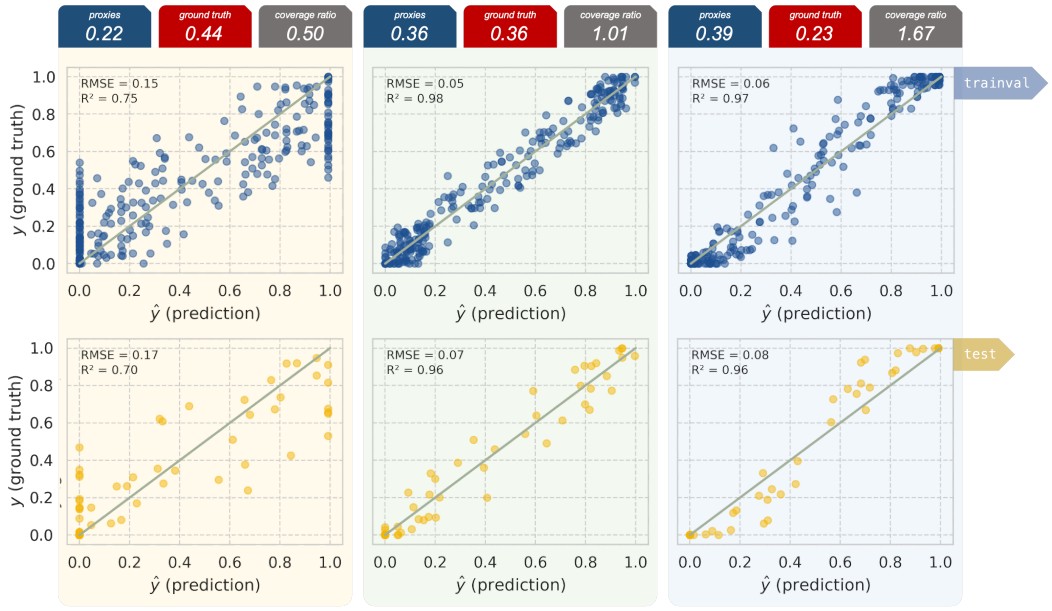

Figure 10: Snapshots of Prediction Performance for Simulation Study 3.

Figure 10 offers three snapshots of the simulation runs corresponding to ECRs equaling 0.50, 1.01 and 1.67. When proxy entropy is low compared with ground truth, the proxy agents fail to provide a sufficiently expressive basis to span the aggregate ground-truth preference. As a result, constrained Lasso can only approximate a lower-dimensional projection of the true preference signal. This results in poor prediction performance, as illustrated in the leftmost snapshot. Conversely, when proxy entropy exceeds ground-truth entropy, prediction error begins to increase—albeit at an infinitesimal

scale. The subtle deterioration of performance when ECR surpasses 1 could be due to various reasons. A plausible cause is from the data perspective: when the proxy pool becomes too diverse, it may introduce spurious variability that does not align well with the true underlying structure. Put differently, an observed response might amalgamate signals from different latent preference factors, some of which are irrelevant to the aggregate preference in focus. While greater diversity expands the representational capacity of the proxy pool, it may also dilute the signal and induce overfitting, thereby reducing generalization accuracy.

The key takeaway from the simulation study is that absent the knowledge of ground truth, it is crucial to ensure an adequate group entropy among the proxy agents to form a functional basis for ground-truth emulation. However, one should also refrain from solely relying on entropy as a measure of preference diversity lest spurious correlations beguile us under limited data regimes. While our alignment method is intentionally designed to function with minimal data—requiring only aggregate response data—in practice, if individual human responses are available, a practitioner may use the group entropy computed from these responses as an anchor for endowment generation.

## C    TAKING P2P TO THE WILD: SUPPLEMENTARY EXPERIMENT RESULTS

### C.1    WAVE 42

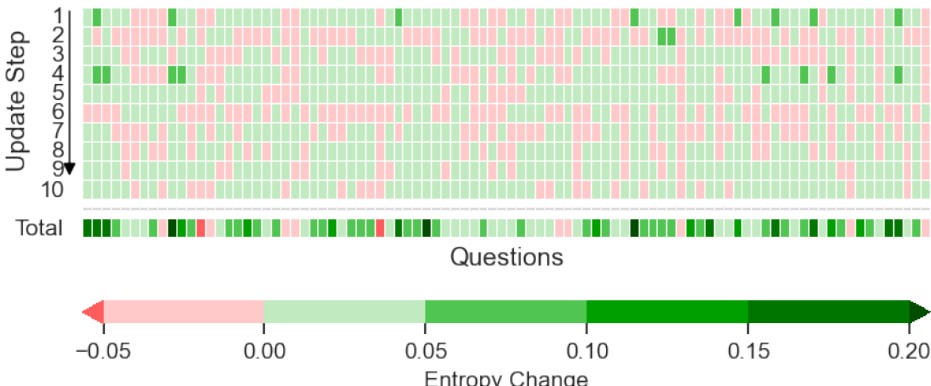

Figure 11: Entropy change across update steps for each question during active endowment generation for ATP W42. Each column represents a question. Green indicates an increase in entropy from the previous step, while red indicates a decline.

**Endowment generation improves entropy**    Figure 11 displays the entropy change for each question across the update steps. Evidently, a significant number of questions experience a noticeable rise in entropy at the end of the first update step. From the second step onward, mixed mode enters the generation loop. Marginal gains in entropy are observed for the majority of questions, whereas some experience mild entropy drops.

**Tracking entropy during active endowment generation**    Figure 12 shows the entropy trajectories by question during active endowment generation. Of the tracked questions, 53 experience a clear rising trend in entropy, while 3 see a noticeable fall in entropy at the end of the generation, the other questions have stable entropy values across the update steps. Figure 13 displays the question entropies at the end of active endowment generation.

**Lasso Selection**    Figure 14 plots the cross-validated mean squared error (MSE) for each value of $\alpha$ examined during hyperparameter tuning. Among the 30 candidate values, cross-validation identifies the optimal $\alpha$ as $1.62 \times 10^{-3}$.

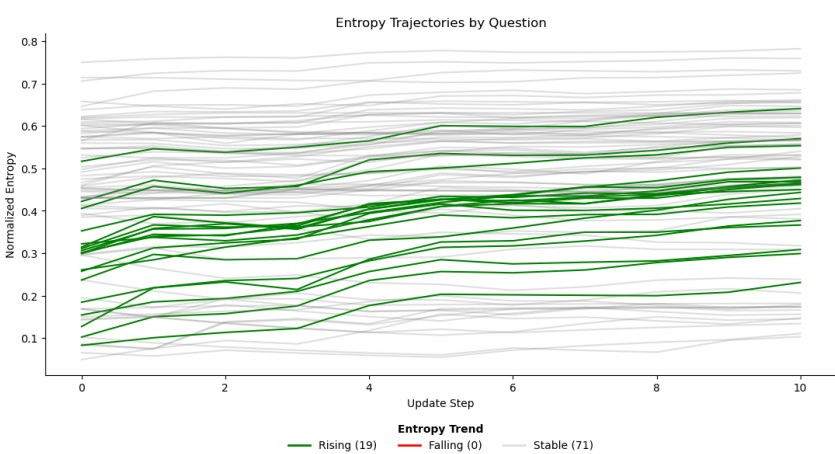

Figure 12: Entropy trajectories by question during active endowment generation. Trends are classified based on slope and volatility of the entropy trajectory: "rising" if the slope exceeds 0.01 and standard deviation is above 0.02; "falling" if the slope is below –0.01 with sufficient volatility; otherwise labeled "stable".

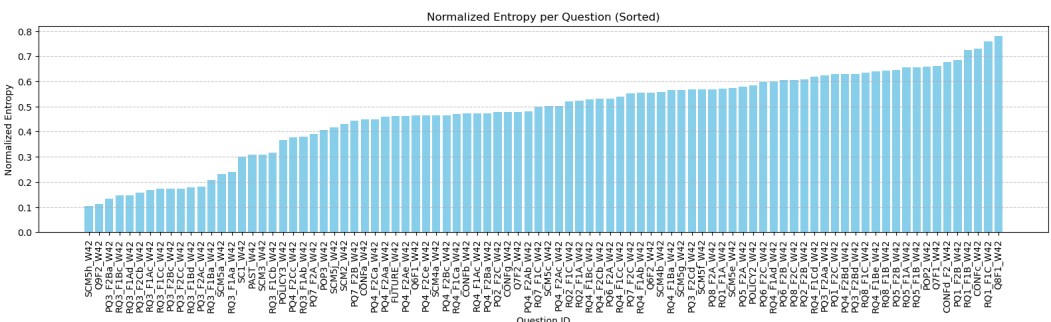

Figure 13: Question entropies at the end of active endowment generation.

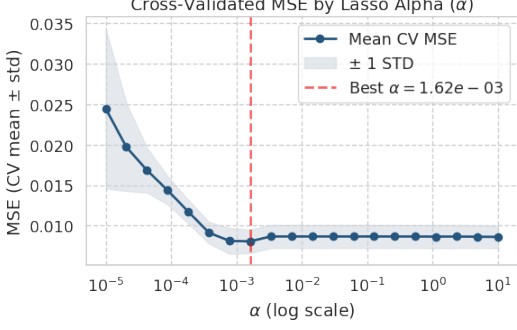

Figure 14: Cross-validated MSE by alpha ($\alpha$), the penalization parameter, for constrained lasso. The best alpha selected by CV is $1.17 \times 10^{-3}$.

**Regression Visualization** For regression analysis, categorical questions are binarized into question-option pairs. For the selection of the agent ensemble, we fit both a constrained lasso and a constrained elastic net, using CV to select the penalization parameters. Consequently, lasso selects 58 out of the 300 synthetic agents to form the agent ensemble, while elastic net selects 107. The training and testing performances are similar between the two methods.

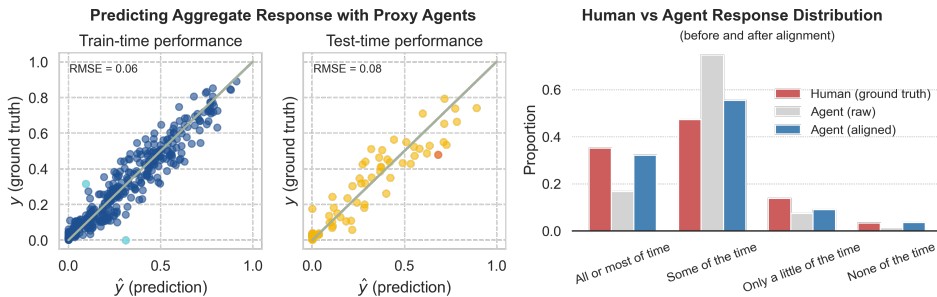

Figure 15: Prompts to Proxies: Emulation Results for ATP W42. **Left**: Train (left) and test (right) prediction performances for the binarized responses. **Right**: Human and agent (pre- and post- alignment) aggregated responses for the sample test question `RQ4_F1Ae_W42`.

Figure 15 shows the prediction performances of the agent ensemble selected with constrained lasso. The train-time root mean squared error (RMSE) is 0.06, with test RMSE slightly higher at 0.08, indicating adequate generalization ability of the functional basis. The bottom panel of Figure 15 presents a snapshot for the aggregated responses for a test set question, comparing the human ground truth with agentic emulations before and after regression-based aggregation. Overall, the results are encouraging, especially considering that `P2P` constructs the endowments based solely on preset and learned attributes and is agnostic on the ground-truth demographic data. Additionally, results of constrained elastic net are in Figures 16 and 17. Overall, the prediction results are similar to those using constrained lasso.

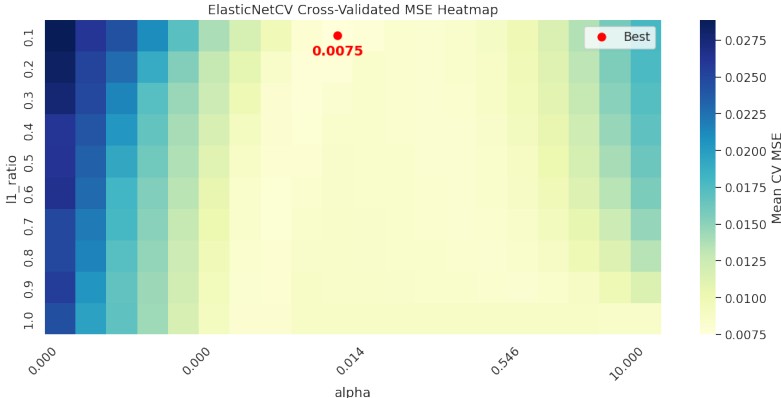

Figure 16: Cross-validated MSE by alpha, the overall penalization parameter, and L1 ratio, the penalization weight for the L1 term, for constrained elastic net. The lowest cross-validated MSE is achieved at $\alpha = 6.95 \times 10^{-3}$, with l1 ratio set at $0.10$.

## C.2 Ablation

### C.2.1 Endowment Budget

Results for the performance of `P2P` with varying endowment budget can be found in Table 3.

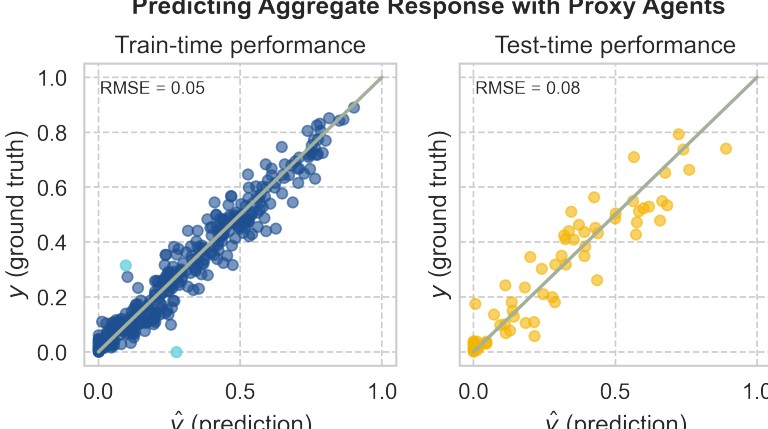

Figure 17: Prediction performance of the agent ensemble selected by constrained elastic net. Results are similar to those using constrained lasso.

Table 3: Performance of `P2P` with varying endowment budget. The average question entropy, test MSE (Lasso and ElasticNet), and generation cost are repeated. All values are mean $\pm$ std over 3 repeated runs.

(a) Budgets 110–210

| Endowment Budget | 110 | 130 | 170 | 210 |
|---|---|---|---|---|
| Avg. Entropy ↑ | $0.3777 \pm 0.0191$ | $0.3697 \pm 0.0057$ | $0.4180 \pm 0.0080$ | $0.4341 \pm 0.0043$ |
| Test MSE (Lasso) ↓ | $0.0144 \pm 0.0018$ | $0.0124 \pm 0.0043$ | $0.0167 \pm 0.0041$ | $0.0100 \pm 0.0025$ |
| Test MSE (Enet) ↓ | $0.0149 \pm 0.0014$ | $0.0123 \pm 0.0045$ | $0.0156 \pm 0.0028$ | $0.0103 \pm 0.0026$ |
| Cost (USD) ↓ | $0.3487 \pm 0.0035$ | $0.8043 \pm 0.0032$ | $\mathbf{0.4108} \pm 0.0089$ | $0.5453 \pm 0.0063$ |

(b) Budgets 250–330

| Endowment Budget | 250 | 290 | 330 |
|---|---|---|---|
| Avg. Entropy ↑ | $0.4179 \pm 0.0165$ | $0.4369 \pm 0.0168$ | $0.4308 \pm 0.0208$ |
| Test MSE (Lasso) ↓ | $0.0111 \pm 0.0006$ | $0.0090 \pm 0.0013$ | $0.0133 \pm 0.0030$ |
| Test MSE (Enet) ↓ | $0.0104 \pm 0.0006$ | $0.0097 \pm 0.0017$ | $0.0121 \pm 0.0031$ |
| Cost (USD) ↓ | $0.6715 \pm 0.0129$ | $0.9365 \pm 0.0027$ | $1.0688 \pm 0.0119$ |

(c) Budgets 370–450

| Endowment Budget | 370 | 410 | 450 |
|---|---|---|---|
| Avg. Entropy ↑ | $0.4216 \pm 0.0179$ | $\mathbf{0.4540} \pm 0.0142$ | $0.4433 \pm 0.0092$ |
| Test MSE (Lasso) ↓ | $0.0140 \pm 0.0037$ | $\mathbf{0.0080} \pm 0.0006$ | $0.0099 \pm 0.0020$ |
| Test MSE (Enet) ↓ | $0.0136 \pm 0.0055$ | $\mathbf{0.0084} \pm 0.0004$ | $0.0097 \pm 0.0020$ |
| Cost (USD) ↓ | $1.2050 \pm 0.0179$ | $1.3295 \pm 0.0142$ | $1.4619 \pm 0.0092$ |

### C.2.2 REGRESSION

Results for the comparison of `P2P` with and without the regression-based aggregation stage on ATP W42 can be found in Table 4.

### C.2.3 MODEL BACKEND

Results for the comparison of `P2P` with varying model backends can be found in Table 5.

Table 4: Comparison of `P2P` performance with and without the regression-based aggregation stage on ATP W42. In the ablation, agent responses from active endowment generation are combined by simple averaging. For the regression setting, we use test MSE for Lasso.

| Metric | Without regression | With regression |
|---|---|---|
| Test MSE (Lasso) ↓ | $0.0254 \pm 0.00041$ | $\mathbf{0.0104} \pm 0.00383$ |

Table 5: Comparison of `P2P` performance on W42 using different model backends. Reported are average question entropy, test MSE (Lasso and ElasticNet), and generation cost. For each model, we fix the endowment budget at 300. All values are mean $\pm$ std over 3 repeated runs.

| Model | Avg. Entropy | Test MSE (Lasso) | Test MSE (Enet) | Cost (USD) |
|---|---|---|---|---|
| GPT-4.1-mini | **0.471** | **0.0090** | **0.0088** | 3.83 |
| | (0.017) | (0.0014) | (0.0014) | (0.045) |
| GPT-4.1-nano | 0.3996 | 0.0226 | 0.0213 | 0.9265 |
| | (0.0154) | (0.0042) | (0.0048) | (0.0043) |
| Qwen | 0.193 | 0.0204 | 0.0196 | – |
| | (0.040) | (0.0014) | (0.0027) | |
| Gemini-2.0-flash | 0.445 | 0.0104 | 0.0091 | **0.971** |
| | (0.026) | (0.0038) | (0.0046) | (0.004) |
| Gemini-2.5-flash-lite | 0.360 | 0.0149 | 0.0151 | 1.023 |
| | (0.0008) | (0.0009) | (0.0013) | (0.0010) |

*Note.* Qwen was run on a local server, thereby incurring no API cost.

## C.3 PANEL STUDY ACROSS ATP WAVES

To obtain a panoramic view of `P2P`'s performance empirically, we conduct a preliminary panel study applying `P2P` to a collection of 14 waves from the American Trends Panel (ATP). Each wave is repeated three times to assess the consistency and robustness of our method under varying random seeds and endowment draws. This subsection provides an extended empirical perspective on the real-world applicability of our approach and sheds light on variability patterns that may not arise in controlled simulation environments. The ATP waves used in this panel study are: W26, W27, W29, W32, W34, W36, W41, W42, W45, W49, W50, W54, W82, and W92.

Figure 18 illustrates the test MSEs for the 42 experiments organized by the number of survey questions per wave. As shown, within-wave performance is generally stable, with most waves exhibiting low variance in test MSE across repeats. However, test MSE varies more significantly across waves. The performance distribution is summarized below:

- 6/14 waves achieve test MSE consistently below 0.015.
- 5/14 waves fluctuate around 0.015.
- 3/14 waves have higher error ranges (0.020–0.025).

These preliminary results reveal no clear relationship between test MSE and the number of survey questions, suggesting that wave-specific factors, such as survey content, attribute coverage, or question formulation, may play a larger role in shaping performance.

To illustrate the interaction between question-level entropy dynamics and predictive performance, we include three snapshots of the panel study in Figure 19, showcasing the differences in question entropy trajectories during active endowment generation and prediction performance.

The patterns observed in the panel study warrant deeper analysis, which is likely to inform refinements that make `P2P` more stable and effective across settings. One especially promising direction is to extend the entropy-based active endowment generation strategy to support multi-objective criteria, allowing the system to balance diversity, coverage, and alignment more systematically.

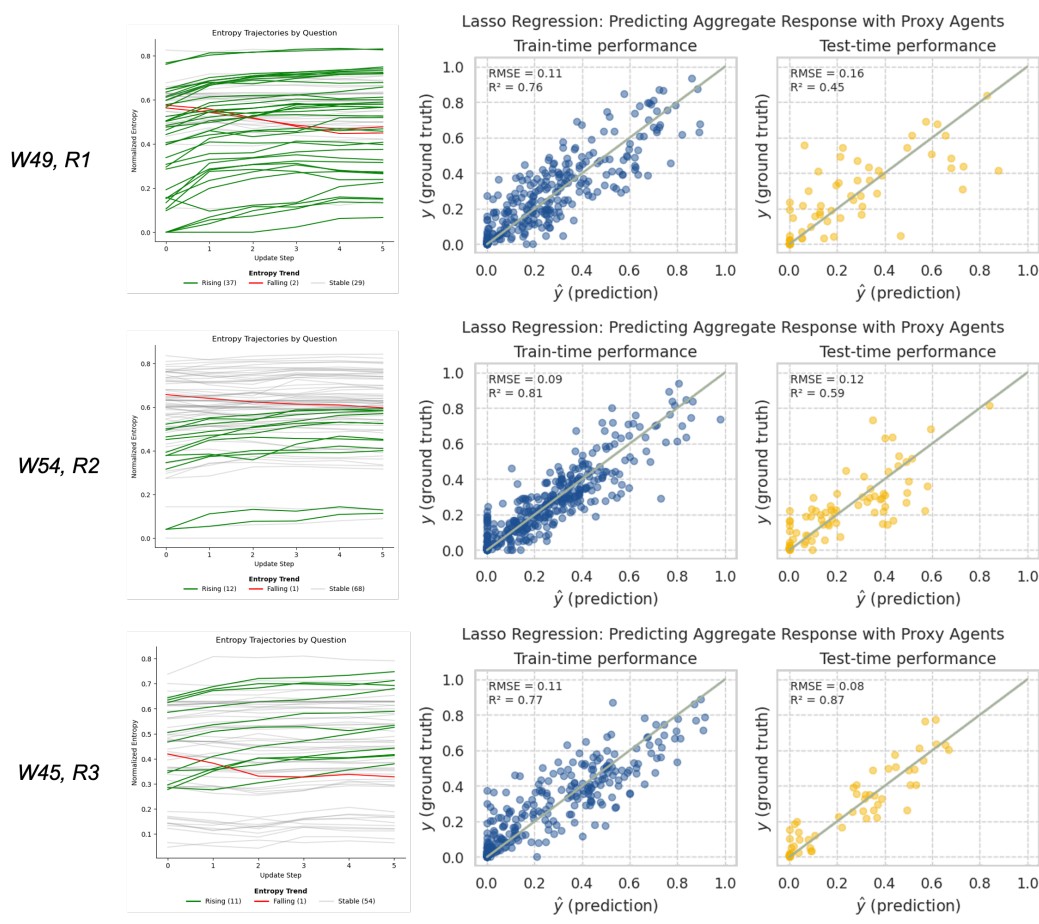

Figure 18: Test performance across ATP waves. Each point represents a wave-specific run, colored by wave identity. The horizontal dashed lines mark two threshold levels: 0.015 and 0.020, used to classify performance regimes.

Figure 19: Representative snapshots for three ATP waves with different test-time MSE. **Top**: the worst-performing wave. **Middle**: a moderately performing wave. **Bottom**: the best-performing wave. **Left**: entropy trajectories classified as rising (green), falling (red), or stable (gray). **Center/right**: Lasso regression fits for aggregate response prediction. While rising entropy often coincides with improved fit, some waves achieve strong performance with limited entropy change, suggesting additional contributing factors beyond diversity expansion.

Another promising extension is to include a module that directly measures question semantics. Based on our preference reconstruction theory, survey questions define the response space $\mathcal{R}$. Question entropy serves as a key signal for underlying diversity when conditioning on a fixed question. However, an equally important source of variation lies in the diversity across questions themselves. This includes differences in topic coverage, difficulty, and framing—factors that shape the expressiveness and completeness of the response space. Ideally, the question set should span a semantically broad topic space so that variations in response entropy meaningfully reflect differences in latent preferences. A dedicated semantic module would enable content-aware data-splitting strategies and support more rigorous generalization analysis across question types. This is an important direction for future development of P2P.

## D METHODOLOGICAL AND EXPERIMENTAL DETAILS

### D.1 ATTRIBUTE BANK

Table 6: Structured view of the attribute bank, with representative modes and attributes grouped by template. Users can extend the bank with extra templates, modes, or attributes.

| Template | Representative **Modes** and Attributes |
|---|---|
| **Core** | core: gender, race, ethnicity, education, income level, ... |
| **Thematic** | economics: occupation, price sensitivity, wealth/debt, ...
politics: policy preferences, civic engagement, ideological orientation, ...
behavioral_biases: status quo bias, loss aversion, overconfidence bias, ... |
| **Theoretical** | maslow: safety needs, self-actualization, love and belonging needs, ...
big_five: conscientiousness, neuroticism, extraversion, ... |

### D.2 PROMPTS

### D.2.1 ATTRIBUTE LEARNER (ATTRIBUTELEARNER)

To generate attributes from a single question:

```
You are an intelligent research assistant trained to analyze
individual survey questions and infer which human attributes might
influence how different people respond.

You are given a single survey question.  Your task is to propose a
list of relevant human attributes|such as demographics, beliefs,
values, personality traits, or ideological leanings|that are
likely to shape responses to this question.

Focus on underlying factors that would cause meaningful variation
in answers across different types of people.  Avoid generic or
overly broad attributes.

Respond **only** with a Python-style list of double-quoted
strings.  Do not include any explanation, headers, or prose before
or after the list.

**Example output format:** ["religious affiliation", "political
ideology", "trust in government"]
```

To generate attributes from a set of questions:

```
You are an intelligent research assistant trained to analyze
survey questions and infer which human attributes might influence
how individuals respond.

You are given a set of training-only survey questions.  Your
task is to propose a list of relevant human attributes|such as
demographics, beliefs, values, personality traits, or ideological
leanings|that are likely to affect responses to these questions.

Carefully analyze the content and framing of the questions.
Identify underlying factors that might shape how different people
respond.  Focus on attributes that are salient, discriminative,
and potentially variable across respondents.

Respond **only** with a Python-style list of double-quoted
strings.  Do not include any explanation, headers, or prose before
or after the list.

**Example output format:** ["attribute1", "attribute2",
"attribute3", "attribute4"]
```

### D.2.2 ENDOWMENT MODEL (ENDOWMENTMODEL)

System prompt:

```
You are an expert assistant trained to generate realistic,
diverse, and demographically plausible personas for social science
surveys.

Each persona should include:
 `eid`:  a short, lowercase, variable-safe identifier that encodes
key traits (e.g., urban_liberal_30s_female).  No punctuation or
spaces.
 `endow_text`:  a natural language description of the persona (1-2
sentences), written as if describing a survey respondent.

Instructions:
 Represent a wide range of age, gender, race, education, region,
and political ideology.
 Avoid repetition of phrasing or demographic combinations across
personas.
 Do not include explanations or formatting outside of the JSON
array.
```

To generate endowments from attributes:

```
Generate n diverse persona(s) that vary meaningfully along the
following attributes:  attr_string.

Each persona should reflect a distinct combination or value
expression of these traits.

Return a JSON array of dictionaries, each with:

- `eid`:  a short, lowercase, variable-safe identifier

- `endow_text`:  a brief natural-language description of the
persona
```

To generate endowments from survey topics (also used for Vanilla baseline):

```
Generate n diverse personas for a survey experiment.

Each persona must include:

- eid:  short, lowercase identifier

- endow_text:  a short natural-language description

topic_lineEnsure diversity across demographics.

Return the result as a JSON array.
```

### D.2.3  SURVEY CONDUCTOR (SURVEYCONDUCTOR)

To use an agent to answer a survey question:

```
You are completing a survey.

Your answer should reflect the person described in the profile
above, using their preferneces, beliefs and experiences.

Respond with only the final answer string, not the code or label
in brackets.

Do not include any reasoning, explanation, or commentary.

Do not preface your answer with phrases like 'I would choose'.

Just return the answer text exactly as it appears in the options.
```

## D.3 ACTIVE ENDOWMENT GENERATION

---

**Algorithm 1** Active Endowment Generation

---

1: Initialize attribute bank $\mathcal{B}$ with core, thematic, and theoretical templates
2: Derive `survey`-specific attributes from $\mathcal{Q}_{\text{train}}$ via `AttributeLearner`; append to $\mathcal{B}$
   **Initial Sampling Stage**
3: Sample initial endowments $\mathcal{E}_0$ via equal-mode sampling from $\mathcal{B}$
4: Instantiate initial agents $\mathcal{A}_0$ from $\mathcal{E}_0$ and elicit responses $\mathcal{R}_0$ on $\mathcal{Q}_{\text{train}}$
5: Initialize `ThemeVariabilityTracker` (henceforce, `Tracker`) with $\mathcal{A}_0$, $\mathcal{R}_0$, and $\mathcal{Q}_{\text{train}}$.
   **Expansion Stage**
6: **while** agent budget $N_A$ not yet reached **do**
7:     Compute variability scores and mode sampling probabilities via `Tracker`
8:     Allocate endowment budget across modes according to probabilities
9:     **if** question patching is enabled **then**
10:         Identify lowest-entropy questions $\mathcal{Q}_{\text{low}}$ via `Tracker`
11:         **for** each $q \in \mathcal{Q}_{\text{low}}$ **do**
12:             **if** $q$ is queried for the first time **then**
13:                 Extract attributes for $q$ via `AttributeLearner` and append to $\mathcal{B}$
14:             **else if** $q$ has appeared more than $n$ times **then**
15:                 Enable mixed-mode strategy using $q$ and top-performing mode
16:             **end if**
17:         **end for**
18:     **end if**
19:     Generate new endowments $\mathcal{E}_{\text{new}}$ using allocated budget over sampled modes, patched question modes (if any) and mixed modes (if any).
20:     Instantiate agents $\mathcal{A}_{\text{new}}$ and elicit responses $\mathcal{R}_{\text{new}}$ on $\mathcal{Q}_{\text{train}}$
21:     Update: $\mathcal{A} \leftarrow \mathcal{A} \cup \mathcal{A}_{\text{new}}$, $\mathcal{R}_{\mathcal{A}} \leftarrow \mathcal{R}_{\mathcal{A}} \cup \mathcal{R}_{\text{new}}$
22:     Update `Tracker` with $\mathcal{A}$, $\mathcal{R}_{\mathcal{A}}$, and $\mathcal{Q}_{\text{train}}$
23: **end while**
24: **Return:** Final agent pool $\mathcal{A}$ and full response set $\mathcal{R}_{\mathcal{A}}$

---

**Algorithm 2** Tracker Update Procedure

---

1: Input: Current agent pool $\mathcal{A}$, current agent responses $\mathcal{R}_{\mathcal{A}}$, and training questions $\mathcal{Q}_{\text{train}}$
2: **for** each question $i \in \mathcal{Q}_{\text{train}}$ **do**
3:     Compute question entropy $H_i(\mathcal{A})$
4: **end for**
5: **for** each mode **do**
6:     Compute variability score $V(\mathcal{A}_{\text{mode}})$
7: **end for**
8: Compute softmax probabilities from variability scores
9: Output: Variability scores, question entropies, and sampling probabilities.

---

## D.4 EXAMPLE ENDOWMENTS

**An example of an endowment for a progressive female in her 20s living in an urban area** "A 24-year-old urban female who identifies as bisexual and holds a master's degree in social work. She exhibits a high fairness bias and is often concerned about equality. Despite her awareness, she sometimes struggles with extreme response bias, influenced by her strong convictions. "

**An example of an endowment for a nonbinary in their 30s valuing liberty living in a suburban area** "They are a 32-year-old nonbinary software developer living in the suburbs. They highly value personal liberty and have a strong work ethic, often putting in extra hours for their projects. With a master's degree in computer science, they frequently engage with scientific research, feeling

that understanding complex topics fosters innovation. Their income is above average, and they embrace intelligence as a vital asset for personal growth. ”

**An example of an endowment for a white male progressive in his 60s living in a small town**
“Pat is a 67-year-old White male from a small town. He identifies as a progressive and believes strongly in the importance of intelligence in community discussions and personal growth. Pat, who identifies as heterosexual, is skeptical of the increasing influence of big corporations on local politics, and he often seeks media that highlights grassroots movements and community activism. He balances his media consumption between traditional outlets and local storytelling platforms, knowing the value of good journalism. ”

## E  EXTENDED LITERATURE REVIEW

### E.1  PLURALISTIC ALIGNMENT

AI alignment refers to the process of ensuring that an AI system operates in accordance with human intentions and values, whether at an individual or aggregate level (Ji et al., 2023; Leike et al., 2018). As AI systems are adopted by increasingly diverse users, they must be designed to recognize and address a wider range of needs. This necessitates for pluralistic systems capable of capturing and representing the diversity of human values and perspectives (Sorensen et al., 2024b). Pluralistic alignment therefore seeks to align models not with a single gold standard, but with a diverse range of user preferences across various attributes.

Most works on pluralistic alignment touch on three complementary ways, namely overton, steerable, and distributional pluralism, in which a single AI model or system can support diversity of views. In overton pluralism, the model aims to output a whole spectrum of reasonable responses (Sorensen et al., 2024a). Methods to achieve this include looking at alignment shifts to transform distributions toward overton coverage (Lake et al., 2024) or few shot prompting from community-specialized LLMs Feng et al. (2024). In steerable pluralism, models are typically steered to adopt or favor particular perspectives or value attributes, often through few-shot examples Feng et al. (2024); Adams et al. (2025). By contrast, our approach steers purely through attributes, with the goal of constructing diverse agents that embody different perspectives. In distributional pluralism, the distribution of the model over possible answers is intended to match that of some target population (Sorensen et al., 2024b), often achieved through model fine-tuning on human responses Cao et al. (2025).

Another way to categorize pluralistic alignment work is by the amount of data they rely on. While most approached in pluralistic alignment involve post-training, they range from resource-intensive approaches that fine-tune on large annotated corpora to lightweight methods that operate with minimal or no additional data. With sufficient data, specialized community models can be fine-tuned and subsequently combined using model merging techniques (Yuan et al., 2024), few-shot prompting from community model outputs (Feng et al., 2024), or federated averaging (Srewa et al., 2025) to scale with user diversity. If limited data are available, an alternative line of post-training approaches leverage in-context learning, where similarity-based retrieval (Adams et al., 2025) or group-informed retrieval (Chen et al., 2024) is performed to select few-shot examples that guide models toward pluralistic alignment. In the absence of detailed data from individual pluralistic users, a third post-training approach is inference-time conditioning. Inference-time conditioning uses structured prompts that encode sociodemographic traits or behavioral dispositions to condition an LLM to emulate the responses of a specific agent. These agents are also known as *endowments*, personas (Castricato et al., 2025) or Silicon Samples (Horton, 2023; Argyle et al., 2023) and are lightweight to deploy and data-efficient. As inference-time conditioning is cheap and fast, they enable mixture or vector-based formulations of pluralistic alignment, which combine agents via multi-objective rewards to avoid averaging out minority perspectives (Chen et al., 2025; Feng et al., 2025). Therefore, P2P follows this line of work for pluralistic alignment. In the first step, P2P performs inference-time conditioning to generate a sufficient set of agents that adequately span the preference space. Thereafter, in the second step, it combines agents to achieve a compact yet pluralistic set of preferences that represent the target survey population.

### E.1.1 ATTRIBUTES

Pluralistic alignment works that use inference-time conditioning often steer agents toward specific profiles using *attributes* that include moral and value dimensions (Adams et al., 2025), or broad morals, values, characteristics, and perspectives (Sorensen et al., 2024a). Attributes may also refer to demographic features, that are more tangible. Examples of these attributes include age, sex, education, income, and religion (Castricato et al., 2025). P2P uses the term *attributes* more broadly to encompass both moral and value dimensions and demographic features. Bundling these together allows P2P to define and condition LLM agents in a way that reflects both who they are and how they evaluate options.

Other related works on pluralistic alignment relate to the curation of datasets, which support research on pluralistic alignment (Sorensen et al., 2024a; Zhou et al., 2025).

### E.2 LARGE LANGUAGE MODELS TO EMULATE HUMAN PREFERENCES

Leveraging the ability of LLMs to emulate human or human-group responses with considerable fidelity, LLM alignment or personalization Tseng et al. (2024) has been largely focused on emulating human responses using various techniques. These include learning from feedback or reward models Bai et al. (2022); Christiano et al. (2017), under distribution shifts Leike et al. (2018); Krueger et al. (2021), finetuning Tan et al. (2024), retrieval augmented generation Prahlad et al. (2025), representation learning Ren et al. (2024), and using human-generated Horton (2023); Castricato et al. (2025); Argyle et al. (2023) prompts. However, when diverse responses need to be generated from many users or user groups instead of individuals, the data and time resources required for alignment grow linearly with the number of LLMs that need to be aligned. To reduce alignment costs, some works instead rely on AI-generated Schuller et al. (2024); Salminen et al. (2024); Ge et al. (2025); Simmons (2023) prompts, or alignment to multiple user groups Mondal et al. (2025); Zhao et al. (2024), even entire populations Cao et al. (2025). However, even the best models struggle with this task Cao et al. (2025). Furthermore, responses from LLMs have been found to have less variation than responses from real surveys Bisbee et al. (2024), and LLMs have a tendency to respond in the middle category Wang et al. (2024). Consequently, it is imperative to address these limitations and devise data- and time-efficient approaches that enable LLMs to more accurately and reliably serve as proxies for survey populations. Our work contributes to this goal by introducing an active learning–inspired alignment framework that dynamically constructs a diverse agent basis under resource constraints and reconstructs population-level preferences via regression-based aggregation, without relying on ground-truth demographic profiles.

### E.3 DATA SELECTION AND ACTIVE LEARNING

To increase preference coverage while operating under budget constraints, our active endowment generation pipeline draws inspiration from machine learning paradigms, particularly active learning.

Traditional data selection methods aim to improve the data efficiency of supervised deep learning models by identifying the most informative or important training examples for generalization Coleman et al. (2020); Paul et al. (2021). Selection via Proxy (SVP) Coleman et al. (2020) uses a lightweight proxy model to estimate sample utility at a fraction of the cost, while GraNd and EL2N scores Paul et al. (2021) identify important training examples early in training.

Similarly, deep active learning applies query strategies to iteratively select the most informative samples from a large pool of data to be added to the training dataset for retraining or fine-tuning Astorga et al. (2024); Hübotter et al. (2025); Li et al. (2025). Active learning strategies balance exploitation, focusing on uncertain or high-impact samples, with exploration, seeking diverse and novel samples to broaden coverage.

In our system, we apply this logic to guide both endowment generation and question patching. Specifically, low-entropy responses are treated as indicators of low coverage or model uncertainty. We refer to question patching as a boosting strategy—not in the ensemble learning sense, but to distinguish it from the main adaptive sampling process—since it targets underperforming questions by allocating additional sampling to enhance expressivity. Meanwhile, the exploitation–exploration mechanism used to sample across existing modes reflects the same active learning principle, but with a different emphasis: prioritizing high-utility regions while maintaining representational diver-

sity. Together, these components form an active learning–inspired loop that incrementally expands coverage of the preference space while minimizing redundant agent generation.

## F  LIMITATIONS AND FUTURE WORK

**Question weighting and regression loss**   By design, `P2P` converts all multiple choice questions into binary questions before fitting a regression model. For example, a question with five options will turn into five questions with binary options. While this design ensures that regression can run smoothly with heterogeneous question types, it steers the model to prioritize fitting questions with more answer options, e.g., a 10-point Likert scale. Put differently, information contained in questions with fewer answer options, e.g., Yes/No, gets diluted after the binarization. Whether or not this design is desirable depends on the user's assumption of the relationship between preference signal and choice granularity. The current design implicitly posits a positive relationship between the two—i.e., the finer the scale, the better a signal on the latent preference. For other assumptions, question reweighting or format transformation need to be done before the regression step.

**Survey data in, text out**   The current design of `P2P` limits its use case to labeling: Given question and options, the aligned agent ensemble expresses its preference by offering a probability distribution over the given options. A key future research direction is to extend the use case to text completion, e.g., drafting resolution, offering suggestion and answering freeform survey questions. How to conflate individual agent textual responses into a representative aggregate response remains an under-explored area of research. Auction design (cf. Dütting et al., 2024) can be a promising aggregation algorithm in this case.

**`P2P` as a benchmark for model steerability**   In this study our primary goal is understanding the preference reconstruction theory and its implementation through the two-stage alignment framework. While our focus is on the design aspects of the system, we acknowledge that the backend model's steerability is also a determinant of `P2P`'s empirical performance. The ablation study using different backends offers first insights on this aspect. It also indicates `P2P`'s potential of being transformed into a benchmark pipeline for the study of model steerability in pluralistic alignment.

# G   EXTRA RESULTS AND EXPERIMENTS

## G.1   MORE DISTRIBUTIONAL METRICS ON KEY RESULTS

To facilitate comparisons across works and corroborate the use of MSE as a valid distribution metric, we follow previous studies (Cao et al., 2025; Feng et al., 2024; Durmus et al., 2024) and additionally report Jensen-Shannon Divergence (1-JSD), majority-class prediction accuracy (MCPA) and Earth Mover Distance (EMD) for the key results in this paper (see Table 7 and Table 8).

Table 7: Extended comparison of baseline (Vanilla), PERSONA, and P2P (AEG) on ATP W42. For each model, we fix the endowment budget at 300. All values are mean $\pm$ std over 3 repeated runs.

| Metric | Vanilla | PERSONA | P2P (AEG) |
|---|---|---|---|
| Avg. Entropy ↑ | $0.3172 \pm 0.0069$ | $0.2871 \pm 0.0003$ | $\mathbf{0.4451} \pm 0.0256$ |
| Cost (USD) ↓ | $\mathbf{0.8649} \pm 0.0070$ | $1.7129 \pm 0.000016^{*}$ | $0.9708 \pm 0.0036$ |
| Test MSE (Lasso) ↓ | $0.0285 \pm 0.0490$ | $0.0362 \pm 0.0012$ | $\mathbf{0.0095} \pm 0.0024$ |
| Test 1-JSD (Lasso) ↑ | $0.7675 \pm 0.0103$ | $0.7533 \pm 0.0032$ | $\mathbf{0.8629} \pm 0.0188$ |
| Test MCPA (Lasso) ↑ | $0.7667 \pm 0.0764$ | $0.6500 \pm 0.0000$ | $\mathbf{0.8000} \pm 0.0707$ |
| Test EMD (Lasso) ↓ | $0.0814 \pm 0.0044$ | $0.0900 \pm 0.0011$ | $\mathbf{0.0480} \pm 0.0065$ |
| Test MSE (Enet) ↓ | $0.0286 \pm 0.0048$ | $0.0339 \pm 0.0012$ | $\mathbf{0.0079} \pm 0.0026$ |
| Test 1-JSD (Enet) ↑ | $0.7666 \pm 0.0096$ | $0.7746 \pm 0.0036$ | $\mathbf{0.8670} \pm 0.0205$ |
| Test MCPA (Enet) ↑ | $0.7833 \pm 0.0577$ | $0.7000 \pm 0.0000$ | $\mathbf{0.8250} \pm 0.1061$ |
| Test EMD (Enet) ↓ | $0.0819 \pm 0.0039$ | $0.0795 \pm 0.0015$ | $\mathbf{0.0443} \pm 0.0089$ |

$^{*}$ PERSONA cost only covers the response elicitation stage—higher if persona generation tokens are included.

## G.2   BASELINE COMPARISONS ACROSS ATP WAVES

To further prove robustness of baseline comparison results, we choose three representative waves in the 14-wave panel study, with weak, medium and high alignment performance: Wave 27, Wave 32 and Wave 45. We run Vanilla and PERSONA on these waves. Results are computed using 3 independent runs and reported in Tables 9, 10 and 11. P2P continues to outperform both baselines across waves, congruent with findings presented in Table 1.

## G.3   EXTRA BASELINES

As a validation of the soundness of the baseline designs (Vanilla and PERSONA) presented in the main paper, we conduct two extra baseline experiments.

### G.3.1   CLUSTERING

Based on Vanilla and PERSONA, we design two variants using K-means clustering as a pre-processing step. Specifically, we first expand the initial pool of personas for Vanilla and PERSONA to 1000, computing their embeddings (via OpenAI's text-embedding-3-large) and cluster them (k=300). We pick the personas (medoids) closest to the centroids to construct the candidate pool and then run our regression stage on the resulting pool. Table 12 shows results on ATP W42.

### G.3.2   MODEL BACKEND PERFORMANCE

To better contextualize the gain of the ensemble methods (Vanilla, PERSONA, and P2P), we provide the model backend performance as a reference baseline. To construct the baseline, we directly query the model backend (gemini-2.0-flash) for answers to the survey questions, using the following instruction:

```
You are completing a public opinion survey.  Answer
each question as an average adult respondent in the
US population, based only on the question text and
```

Table 8: Performance of `P2P` (5 update steps) on 14 ATP waves (additional metrics).

| Wave | W26 | W27 | W29 | W32 | W34 | W36 | W41 | |
|------|-----|-----|-----|-----|-----|-----|-----|---|
| **Lasso 1-JSD** | 0.8677 | 0.8234 | 0.8149 | 0.8325 | 0.8216 | 0.8184 | 0.8277 | |
| | (.0011) | (.0059) | (.0265) | (.0072) | (.0103) | (.0129) | (.0017) | |
| **Lasso EMD** | 0.0556 | 0.0715 | 0.0828 | 0.0655 | 0.0701 | 0.0626 | 0.0688 | |
| | (.0006) | (.0025) | (.0131) | (.0031) | (.0070) | (.0060) | (.0064) | |
| **Lasso MCPA** | 0.7436 | 0.5778 | 0.8205 | 0.5208 | 0.6364 | 0.6364 | 0.7111 | |
| | (.0444) | (.0385) | (.0444) | (.0361) | (.0909) | (.0455) | (.0385) | |
| **Enet 1-JSD** | 0.8677 | 0.8159 | 0.8149 | 0.8325 | 0.8216 | 0.8203 | 0.8277 | |
| | (.0011) | (.0107) | (.0265) | (.0072) | (.0103) | (.0098) | (.0017) | |
| **Enet EMD** | 0.0556 | 0.0729 | 0.0828 | 0.0655 | 0.0701 | 0.0614 | 0.0688 | |
| | (.0006) | (.0034) | (.0131) | (.0031) | (.0070) | (.0039) | (.0064) | |
| **Enet MCPA** | 0.7436 | 0.5556 | 0.8205 | 0.5208 | 0.6364 | 0.6212 | 0.7111 | |
| | (.0444) | (.0385) | (.0444) | (.0361) | (.0909) | (.0694) | (.0385) | |
| **Wave** | W42 | W45 | W49 | W50 | W54 | W82 | W92 | Avg |
| **Lasso 1-JSD** | 0.8529 | 0.8479 | 0.8137 | 0.8052 | 0.8012 | 0.8555 | 0.8100 | **0.8280** |
| | (.0027) | (.0134) | (.0166) | (.0146) | (.0193) | (.0123) | (.0045) | (.0205) |
| **Lasso EMD** | 0.0529 | 0.0567 | 0.0654 | 0.0755 | 0.0677 | 0.0520 | 0.0828 | **0.0664** |
| | (.0011) | (.0100) | (.0045) | (.0081) | (.0069) | (.0051) | (.0039) | (.0099) |
| **Lasso MCPA** | 0.7667 | 0.7778 | 0.6667 | 0.6333 | 0.5926 | 0.7451 | 0.6410 | **0.6764** |
| | (.0289) | (.0385) | (.0361) | (.1155) | (.0642) | (.0340) | (.0444) | (.0864) |
| **Enet 1-JSD** | 0.8566 | 0.8479 | 0.8137 | 0.8052 | 0.8004 | 0.8555 | 0.8085 | **0.8277** |
| | (.0052) | (.0134) | (.0166) | (.0146) | (.0193) | (.0123) | (.0014) | (.0212) |
| **Enet EMD** | 0.0503 | 0.0567 | 0.0654 | 0.0755 | 0.0672 | 0.0520 | 0.0843 | **0.0663** |
| | (.0019) | (.0100) | (.0045) | (.0081) | (.0071) | (.0051) | (.0018) | (.0105) |
| **Enet MCPA** | 0.7833 | 0.7778 | 0.6667 | 0.6333 | 0.5556 | 0.7451 | 0.6154 | **0.6704** |
| | (.0577) | (.0385) | (.0361) | (.1155) | (.0000) | (.0340) | (.0769) | (.0945) |

Table 9: Extended comparison of Vanilla, PERSONA, and `P2P` (AEG) on ATP W27 (weak alignment). For each model, we fix the endowment budget at 300. All values are mean $\pm$ std over 3 repeated runs. `P2P` results are taken from Panel Study (Table 8).

| Metric | Vanilla | PERSONA | P2P (Table 8) |
|--------|---------|---------|---------------|
| Avg. Entropy ↑ | $0.4623 \pm 0.0066$ | $0.4016 \pm 0.0001$ | $\mathbf{0.5363} \pm 0.0099$ |
| Cost (USD) ↓ | $\mathbf{0.6314} \pm 0.0037$ | $1.2562 \pm 0.00004^*$ | $0.7106 \pm 0.0039$ |
| Test MSE (Lasso) ↓ | $0.0322 \pm 0.0045$ | $0.0412 \pm 0.0006$ | $\mathbf{0.0218} \pm 0.0010$ |
| Test 1-JSD (Lasso) ↑ | $0.7951 \pm 0.0078$ | $0.7282 \pm 0.0039$ | $\mathbf{0.8234} \pm 0.0059$ |
| Test MCPA (Lasso) ↑ | $0.6000 \pm 0.0667$ | $\mathbf{0.6222} \pm 0.0385$ | $0.5778 \pm 0.0385$ |
| Test EMD (Lasso) ↓ | $0.4093 \pm 0.0187$ | $0.5566 \pm 0.0113$ | $\mathbf{0.0714} \pm 0.0025$ |
| Test MSE (Enet) ↓ | $0.0326 \pm 0.0042$ | $0.0403 \pm 0.0005$ | $\mathbf{0.0238} \pm 0.0027$ |
| Test 1-JSD (Enet) ↑ | $0.7925 \pm 0.0101$ | $0.7356 \pm 0.0059$ | $\mathbf{0.8159} \pm 0.0107$ |
| Test MCPA (Enet) ↑ | $\mathbf{0.6444} \pm 0.0385$ | $0.5777 \pm 0.0385$ | $0.5556 \pm 0.0385$ |
| Test EMD (Enet) ↓ | $0.4121 \pm 0.0291$ | $0.5482 \pm 0.0141$ | $\mathbf{0.0729} \pm 0.0034$ |

* PERSONA cost only covers the response elicitation stage—higher if persona generation tokens are included.

Table 10: Extended comparison of baseline (Vanilla), PERSONA, and P2P (AEG) on ATP W32 (medium alignment). For each model, we fix the endowment budget at 300. All values are mean $\pm$ std over 3 repeated runs. P2P results are taken from Panel Study (Table 8).

| Metric | Vanilla | PERSONA | P2P (Table 8) |
|---|---|---|---|
| Avg. Entropy ↑ | $0.5514 \pm 0.0055$ | $0.5569 \pm 0.0002$ | $\mathbf{0.6178} \pm 0.0153$ |
| Cost (USD) ↓ | $\mathbf{0.6363} \pm 0.0035$ | $1.2805 \pm 0.00002^{*}$ | $0.7241 \pm 0.0012$ |
| Test MSE (Lasso) ↓ | $0.0192 \pm 0.0005$ | $0.0252 \pm 0.0004$ | $\mathbf{0.0160} \pm 0.0019$ |
| Test 1-JSD (Lasso) ↑ | $0.8014 \pm 0.0029$ | $0.7721 \pm 0.0037$ | $\mathbf{0.8325} \pm 0.0072$ |
| Test MCPA (Lasso) ↑ | $0.4583 \pm 0.0361$ | $0.5000 \pm 0.0625$ | $\mathbf{0.5208} \pm 0.0361$ |
| Test EMD (Lasso) ↓ | $0.3500 \pm 0.0008$ | $0.4442 \pm 0.0082$ | $\mathbf{0.0655} \pm 0.0031$ |
| Test MSE (Enet) ↓ | $0.0207 \pm 0.0001$ | $0.0227 \pm 0.0002$ | $\mathbf{0.0160} \pm 0.0019$ |
| Test 1-JSD (Enet) ↑ | $0.7990 \pm 0.0034$ | $0.7835 \pm 0.0014$ | $\mathbf{0.8325} \pm 0.0072$ |
| Test MCPA (Enet) ↑ | $0.4583 \pm 0.0361$ | $0.5000 \pm 0.0625$ | $\mathbf{0.5208} \pm 0.0361$ |
| Test EMD (Enet) ↓ | $0.3679 \pm 0.0105$ | $0.4187 \pm 0.0025$ | $\mathbf{0.0655} \pm 0.0031$ |

$^{*}$ PERSONA cost only covers the response elicitation stage—higher if persona generation tokens are included.

Table 11: Extended comparison of baseline (Vanilla), PERSONA, and P2P (AEG) on ATP W45 (strong alignment). For each model, we fix the endowment budget at 300. All values are mean $\pm$ std over 3 repeated runs. P2P results are taken from Panel Study (Table 8).

| Metric | Vanilla | PERSONA | P2P (Table 8) |
|---|---|---|---|
| Avg. Entropy ↑ | $0.3888 \pm 0.0021$ | $0.3645 \pm 0.0011$ | $\mathbf{0.4450} \pm 0.0060$ |
| Cost (USD) ↓ | $\mathbf{0.6214} \pm 0.0036$ | $1.2397 \pm 0.00001^{*}$ | $0.6928 \pm 0.0019$ |
| Test MSE (Lasso) ↓ | $0.0153 \pm 0.0025$ | $0.0288 \pm 0.0014$ | $\mathbf{0.0079} \pm 0.0023$ |
| Test 1-JSD (Lasso) ↑ | $0.8046 \pm 0.0123$ | $0.7734 \pm 0.0057$ | $\mathbf{0.8479} \pm 0.0134$ |
| Test MCPA (Lasso) ↑ | $0.7555 \pm 0.1018$ | $0.7111 \pm 0.0385$ | $\mathbf{0.7778} \pm 0.0385$ |
| Test EMD (Lasso) ↓ | $0.1460 \pm 0.1246$ | $0.2192 \pm 0.2272$ | $\mathbf{0.0567} \pm 0.0100$ |
| Test MSE (Enet) ↓ | $0.0155 \pm 0.0023$ | $0.0278 \pm 0.0009$ | $\mathbf{0.0079} \pm 0.0023$ |
| Test 1-JSD (Enet) ↑ | $0.8063 \pm 0.0110$ | $0.7789 \pm 0.0040$ | $\mathbf{0.8479} \pm 0.0134$ |
| Test MCPA (Enet) ↑ | $\mathbf{0.8000} \pm 0.0667$ | $0.7333 \pm 0.0000$ | $0.7778 \pm 0.0385$ |
| Test EMD (Enet) ↓ | $0.1410 \pm 0.1163$ | $0.2062 \pm 0.2033$ | $\mathbf{0.0567} \pm 0.0100$ |

$^{*}$ PERSONA cost only covers the response elicitation stage—higher if persona generation tokens are included.

Table 12: Comparison of Vanilla, PERSONA and their clustering-based variants where 1000 agents are reduced to 300 via $k$-means clustering.

| Metric | Vanilla | Vanilla (1000→300) | PERSONA | PERSONA (1000→300) |
|---|---|---|---|---|
| Avg. Entropy ↑ | $0.3172 \pm 0.0069$ | $0.3162$ | $0.2871 \pm 0.00026$ | $0.2844$ |
| Test MSE (Lasso) ↓ | $0.0285 \pm 0.00490$ | $0.0266$ | $0.0362 \pm 0.00116$ | $0.0395$ |
| Test MSE (Enet) ↓ | $0.0286 \pm 0.00476$ | $0.0266$ | $0.0339 \pm 0.00124$ | $0.0387$ |
| Cost (USD) ↓ | $\mathbf{0.8649 \pm 0.0070}$ | $0.8834^{\dagger}$ | $1.7129 \pm 0.000016^{*}$ | $1.7203^{*\dagger}$ |

$^{*}$ Only considering survey elicitation cost on 300 personas—actual cost is higher.
$^{\dagger}$ Excluding embedding cost.

```
options.  Respond with only the final answer string,
not the code or label in brackets.  Do not include any
reasoning, explanation, or commentary.  Do not preface
your answer with phrases like 'I would choose'.  Just
return the answer text exactly as it appears in the
options.
```

We draw 300 Monte-Carlo samples per question with temperature set to 0.7. This results in an empirical answer distribution per question. On ATP W42, this baseline attains test MSE 0.0918 and average entropy 0.0161, conspicuously worse than the results for the ensemble baselines, confirming the gains via ensemble methods (Vanilla, PERSONA, and `P2P`).

## G.4   EXTRA SENSITIVITY ANALYSES

### G.4.1   ATTRIBUTE ABLATION

As discussed in Section 3.2, attributes play the role of control handles in `P2P`'s generation logic, and *freeform* templates allow `P2P` to derive attributes from a specific survey or question, complementing the preset attribute bank with data-driven insights on endowment generation. To evaluation the potency of the freeform templates, we conduct an ablation in study in which we sequentially turn off question patching, survey patching and both. We additionally test the impact of per-endowment attribute cap through a setting where the default max attribute=10 is changed to 20.

Table 13: Attribute Ablation. Reported are mean $\pm$ std over 3 repeated runs.

| Metric | Avg. Entropy ↑ | Test MSE ↓ (Lasso) | Test MSE ↓ (Enet) | Cost (USD) ↓ |
|---|---|---|---|---|
| Original | **0.4451** $\pm$ .0256 | **0.0095** $\pm$ .0024 | **0.0079** $\pm$ .0026 | 0.9708 $\pm$ .0036 |
| Survey patch off | 0.4378 $\pm$ .0134 | 0.0103 $\pm$ .0011 | 0.0102 $\pm$ .0002 | 0.9785 $\pm$ .0122 |
| Question patch off | 0.3989 $\pm$ .0203 | 0.0114 $\pm$ .0016 | 0.0109 $\pm$ .0015 | 0.9698 $\pm$ .0011 |
| Freeform off | 0.3953 $\pm$ .0097 | 0.0128 $\pm$ .0008 | 0.0123 $\pm$ .0012 | **0.9688** $\pm$ .0015 |
| max attributes = 20 | 0.4363 $\pm$ .0061 | 0.0124 $\pm$ .0046 | 0.0115 $\pm$ .0041 | 0.9880 $\pm$ .0167 |

Results in Table 13 suggest that performance of `P2P` is robust to attribute cap, whereas turning off freemode templates (especially question patching) degrades both average entropy and test MSE without significant cost gain. Therefore, from a performance vantage, it is recommended to keep the freeform templates for the extra benefits of data-driven insights.

### G.4.2   SENSITIVITY ANALYSIS OF WEIGHTING SCHEMES

In Appendix F, we have discussed the information assumption for the default question weighting scheme and its limitations. In `P2P`, we additionally the user the option of block weighting, with which each question contributes equally to the training loss. As a sensitivity check, we compare the performance of `P2P` under the two weighting schemes for ATP W42.

Table 14: Sensitivity check of weighting schemes. Reported are mean $\pm$ std over 3 repeated runs.

| Weighting Scheme | Train MSE ↓ (Lasso) | Test MSE ↓ (Lasso) | Train MSE ↓ (Enet) | Test MSE ↓ (Enet) |
|---|---|---|---|---|
| Default | 0.00526 $\pm$ .00141 | 0.00946 $\pm$ .00241 | **0.00353** $\pm$ .00161 | **0.00792** $\pm$ .00261 |
| Block | **0.00472** $\pm$ .00228 | **0.00884** $\pm$ .00281 | 0.00389 $\pm$ .00152 | 0.00797 $\pm$ .00263 |

As shown in Table 14, performance of `P2P` remains comparable under the two weighting schemes for ATP W42. We further spot-check the block weighting results on WVS GB data, which yields similar test MSE to the that under default weighting.

### G.5 STRESS TEST: COMPARISON WITH AN SFT-ALIGNED MODEL UNDER TOPIC SHIFT

`P2P` is designed as a plug-and-play preference alignment method that can be deployed on new populations with limited-data and limited-compute regimes. In the main paper, the baseline comparisons thus focus on prompting-based ensemble methods under the same constraints. As discussed in Appendix E, supervised finetuning (SFT) for a distribution-calibrated model offers an alternative approach under more lenient data and compute environments. An exemplar is Cao et al. (2025), where SFT is used to train a predictive model based on pooled World Values Survey (WVS) data from 46 countries, and conditioned on locale to directly predict answer distribution for a given survey question. They further test the SFT-aligned model on a distinct unseen set of WVS questions along with other cross-country and cross-dataset validation tasks.

In this regard, our work differs from Cao et al. (2025) not just in terms of training regimes, but also in key methodological premises. Specifically, we regard each survey panel as a benchmark for *a specific population preference on a given topic at a given moment in time*, where such preference is *gaugeable* through survey responses (cf. Figure 1, right panel). We do *not* assume that preferences on a topic domain can be reliably inferred if the training questions never probe that domain. For example, inferring political attitudes solely from non-political or religious items is a strong assumption for both humans and synthetic agents. Likewise, we do *not* assume preferences can carry over from one survey program to another, particularly when the surveys are conducted at different points in time and on different subpopulations of the same locale. Studying how aligned preferences generalize across topics, survey programs and over time is itself a key direction for future work, and will require both theoretical and empirical advances.

Nevertheless, Cao et al. (2025)'s setup constitutes an interesting stress test for `P2P` outside our studied scenarios. We present in this section a head-to-head comparison with their method on their WVS arena. We follow their protocol: Q1 and Q2 for train/validation and Q3 as test, which induces a topic shift from general attitudinal & religious/ethical items (Q1+Q2) to political interest and culture items (Q3). In our language, this asks: if we align to a population on broad attitudinal dimensions (Q1+Q2), how well does the aligned ensemble generalize to a new unseen dimension (Q3: political-cultural questions)? We choose three representative locales: US (overlaps ATP), Great Britain (Western Europe) and Hong Kong SAR (East Asia). For each, we filter Cao et al. (2025)'s data to that locale, train P2P (backend: gemini-2.0-flash; 300 endowments) on local Q1+Q2 (train+val), and evaluate on Q3. Table 15 shows the training schemes for `P2P` and Cao et al. (2025).

Table 15: Different training regimes for `P2P` and Cao et al. (2025).

| Setup | Train+Val Entries (Q1 + Q2) | Test Entries (Q3) | Model backend |
|---|---|---|---|
| `P2P` (US) | 185 (150 + 35) | 59 | gemini-2.0-flash |
| `P2P` (GB) | 172 (140 + 32) | 60 | gemini-2.0-flash |
| `P2P` (HK) | 185 (150 + 35) | 59 | gemini-2.0-flash |
| Cao et al. (C1: 46 countries) | 8427 (6841 + 1586) | 2719 | Llama3-8B-Instruct |

Table 16: `P2P` vs. Cao et al. (2025) on WVS (Q3). Reported are mean $\pm$ std over 3 repeated runs.

| Setup | Method | Test MSE $\downarrow$ | Test 1-JSD $\uparrow$ | Test MCPA $\uparrow$ | Test EMD $\downarrow$ |
|---|---|---|---|---|---|
| `P2P` (US) | Lasso | $0.0217 \pm .0014$ | $0.7492 \pm .0036$ | $0.6271 \pm .0143$ | $0.1188 \pm .0048$ |
| `P2P` (US) | Enet | $0.0218 \pm .0019$ | $0.7497 \pm .0051$ | $0.6384 \pm .0317$ | $0.1184 \pm .0066$ |
| `P2P` (GB) | Lasso | $0.0249 \pm .0009$ | $0.7416 \pm .0055$ | $0.4889 \pm .0347$ | $0.1296 \pm .0045$ |
| `P2P` (GB) | Enet | $0.0249 \pm .0009$ | $0.7463 \pm .0067$ | $0.5111 \pm .0096$ | $0.1294 \pm .0048$ |
| `P2P` (HK) | Lasso | $0.0284 \pm .0023$ | $0.7115 \pm .0125$ | $0.4802 \pm .0353$ | $0.1507 \pm .0122$ |
| `P2P` (HK) | Enet | $0.0290 \pm .0023$ | $0.7151 \pm .0133$ | $0.4689 \pm .0298$ | $0.1532 \pm .0108$ |
| Cao et al. | SFT | $-$ | $0.777$ | $0.43$ | $-$[*] |

[*] EMD not reported due to incompatible definition in Cao et al. (2025).

Table 16 presents the results on the performance comparison. In summary, P2P achieves 1-JSD on Q3 comparable to Cao et al. (2025)'s SFT model and matches or exceeds their MCPA, despite using far fewer training entries per locale and no fine-tuning (Table 15). Following our earlier discussion, we interpret performance on Q3 as evidence that some preference structure transfers across topics, rather than as a requirement that a preference-alignment method must perfectly extrapolate to arbitrary unseen domains.

## G.6 APPLYING P2P TO MORE LOCALES ON WVS

Our results in the main paper focus on ATP data on the US population. While it shows that the method can be applied to multiple topics across different cross sections, it remains intriguing how it performs on other locales and different survey programs. To address this question, we build upon Cao et al. (2025)'s WVS dataset and focus on the aforementioned three representative locales (US, GB and HK), but repartition the questions with the standard 7:1.5:1.5 random split so that the Q1+Q2 to Q3 topic shift resolves. Table 17 reports the performance results, with mean $\pm$ std computed using 3 independent runs.

Table 17: P2P performance on WVS Wave 7 (US, Great Britain, Hong Kong SAR) under a 7:1.5:1.5 question split, using gemini-2.0-flash as the backend. Reported are mean $\pm$ std over 3 repeated runs.

| Metric | WVS W7 US | WVS W7 GB | WVS W7 HK |
|---|---|---|---|
| Avg. Entropy $\uparrow$ | $0.5720 \pm 0.0149$ | $0.5710 \pm 0.0044$ | $0.5478 \pm 0.0211$ |
| Cost (USD) $\downarrow$ | $1.7551 \pm 0.0097$ | $1.7095 \pm 0.0031$ | $1.7365 \pm 0.0080$ |
| Test MSE (Lasso) $\downarrow$ | $0.0149 \pm 0.0005$ | $0.0141 \pm 0.0014$ | $0.0242 \pm 0.0013$ |
| Test 1-JSD (Lasso) $\uparrow$ | $0.8030 \pm 0.0055$ | $0.7895 \pm 0.0049$ | $0.7590 \pm 0.0046$ |
| Test MCPA (Lasso) $\uparrow$ | $0.6667 \pm 0.0152$ | $0.6481 \pm 0.0321$ | $0.5789 \pm 0.0263$ |
| Test EMD (Lasso) $\downarrow$ | $0.1075 \pm 0.0035$ | $0.1038 \pm 0.0081$ | $0.1424 \pm 0.0034$ |
| Test MSE (Elastic Net) $\downarrow$ | $0.0147 \pm 0.0005$ | $0.0137 \pm 0.0010$ | $0.0244 \pm 0.0005$ |
| Test 1-JSD (Elastic Net) $\uparrow$ | $0.8066 \pm 0.0067$ | $0.7950 \pm 0.0073$ | $0.7602 \pm 0.0032$ |
| Test MCPA (Elastic Net) $\uparrow$ | $0.6930 \pm 0.0152$ | $0.6296 \pm 0.0321$ | $0.5877 \pm 0.0304$ |
| Test EMD (Elastic Net) $\downarrow$ | $0.1066 \pm 0.0025$ | $0.1004 \pm 0.0054$ | $0.1430 \pm 0.0046$ |

As shown in Table 17, performance is higher for US and GB and somewhat lower for HK, which is consistent with broader findings that large language models trained primarily on English and Western-centric data tend to better match Western public opinion than non-Western populations. The performance range also accords with our 14-wave ATP study presented in the main paper (cf. Table 2). While backend selection warrants careful consideration when applying P2P to a specific locale, these results indicate that P2P remains broadly applicable across locales and survey programs.

## G.7 INTUITIVE EXAMPLES OF ENDOWMENTS

Figure 20 provides three examples of endowments generated and selected by P2P for a run on ATP W42, tracing back to their attributes, modes, and templates.

## G.8 EXTENDED LIMITATIONS & FUTURE WORK

**Temporal Alignment and Non-stationary Preference** Any method trained at a single moment risks becoming stale as attitudes and beliefs shift. This also affects P2P when considering using a fitted ensemble in future surveys. Our view is that synthetic agents should be periodically re-anchored using more recent, easier-to-collect data (e.g., shorter or cheaper surveys) and then tested on more demanding items. Architecturally, P2P is set up for this. The endowment generation and regression stages can be rerun on new waves, and one can track how learned weights and entropy patterns evolve, or when reconstruction quality begins to deteriorate. While P2P provides the machinery, systematic studies of preference shift and alignment across time and topics remain open.

**EID**: rural_workingclass_hispanic_extreme_response_bias_agree_397608

**Template**

Thematic

**Mode**

`core+cognition_psychology`

**Attributes**

decision fatigue; narcissistic tendency; sexual orientation; extreme response bias; emotion regulation capacity; urban/rural residency; income level; framing susceptibility; confirmation bias; race

**Endowment**

This respondent is a working-class Hispanic man residing in a rural community. He exhibits a strong acquiescence bias, tending to agree with most statements regardless of content. He might feel pressure to provide socially desirable answers or lack confidence in expressing dissenting opinions.

**Selected by Lasso?** ☑        **Weight:** 0.104142

---

**EID**: rural_elderly_limitedinfo_lowses_14507e

**Template**

Question (*Question Patching*)

**Mode**

`PQ3_F2Cb_W42`

**Attributes**

access to healthcare information; age; educational attainment; exposure to health sciences; field of study; interest in health and nutrition; socioeconomic status

**Endowment**

This respondent is a woman in her late 70s living in a rural area. She has limited access to reliable health information, primarily relying on local news and word-of-mouth. Her formal education ended after high school. She has very little interest in health and nutrition beyond basic daily needs and is of lower socioeconomic status, primarily concerned with managing on a fixed income.

**Selected by Lasso?** ☑        **Weight:** 0.011782

---

**EID**: urban_libertarian_iconoclast_highsci_5a5ac7

**Template**

Question (*Mixed Patching*)

**Mode**

`cognition_psychology+`
`core+Q9F2_W42`

**Attributes**

cynicism; political leaning/affiliation; locus of control; decision fatigue; iconoclastic thinking; prior beliefs about the industry group; cognitive style; awareness of conflicts of interest; level of science literacy; general trust in scientists

**Endowment**

This respondent is a young, urban man who identifies as libertarian. He thinks iconoclastically and questions established norms and authority. He has a high level of scientific literacy, but this does not necessarily translate to trust in scientists. He is aware of the potential for conflicts of interest and is deeply skeptical of centralized power structures, including scientific institutions.

**Selected by Lasso?** ☑        **Weight:** 0.041436

Figure 20: Intuitive examples of endowments generated by `P2P` for a run on ATP W42.

**Short-cut Learning and Generalization**  Overfitting to survey artifacts is a key concern for *any* survey-based evaluation. This concern is closely related to what the robustness literature calls *short-cut learning* or *Clever Hans* behavior, where models exploit superficial regularities of a dataset instead of learning the underlying concept (Geirhos et al., 2020; Lapuschkin et al., 2019). Our empirical evaluation across 14 ATP waves is designed to mitigate the risk that our findings hinge on idiosyncrasies of a single wave, and the additional WVS experiments further mitigate the risk of ATP-specific shortcutting. While these steps serve as initial evidence against program-specific short-cutting, it does not constitute a complete solution to the generalization problem. A full multi-survey study (e.g., across GlobalOpinionQA and other polling corpora) is beyond the scope of this paper, but `P2P`'s modular pipeline readily supports this kind of analysis. Fundamentally, we see this as partly a survey design challenge: if we want synthetic agents to learn stable preferences rather than survey-specific shortcuts, future work may need survey instruments that are explicitly constructed with learnability and cross-topic generalization in mind. In the last paragraph of Appendix , we suggest a semantic question module for `P2P` to quantify the expressiveness of survey questions, which could serve as a parallel line of work to mitigate shortcutting on the model-design side. Taken together, these directions highlight an important and promising interdisciplinary research frontier in our pursuit of alignment science.

