# OpenReview forum: "Prompts to Proxies: Emulating Human Preferences via a Compact LLM Ensemble"
_ICLR.cc/2026/Conference — Submitted to ICLR 2026_

### Official Review · Reviewer_qjZY · 2025-10-28

**Soundness:** 3
**Presentation:** 1
**Contribution:** 2
**Rating:** 4
**Confidence:** 3

**Summary:**

This paper introduces Prompts-to-Proxies (P2P), a framework to emulate population-level preferences through a two-stage process. Importantly, grounded in preference reconstruction theory, the authors argue that instead of learning a one-to-one mapping from human subgroups to model personas, it is sufficient to construct a functional basis of diverse proxy agents and learn sparse aggregation weights that reconstruct aggregate responses. Specifically, stage 1 employs generation of endowed agents using entropy-based adaptive sampling to create diverse agent personas from a structured attribute bank. Those agents are used to generate diverse basis. Stage 2 uses L1-regularized regression over agent responses to match population aggregates and select a parsimonious ensemble. The approach is validated on American Trends Panel (ATP), showing lower test MSE and higher response entropy than baselines with modest cost.

**Strengths:**

< Strength >

- The preference reconstruction theory provides an mathematical justification for the approach. The formalization that population preferences can be represented via multiple ensemble configurations without requiring one-to-one demographic mapping is both theoretically sound and practically impactful.
- The paper addresses a real social science challenge with a theoretically grounded solution, and tested on realistic dataset (ATP) compared to existing alignment method papers.
- The paper presents concrete ablation studies examining key components covering endowment budgets, regression methods, and model backends
- The two-stage architecture cleanly separates diversity generation from preference reconstruction, enabling independent improvement and modification of each component. The attribute bank structure is also extensible.

**Weaknesses:**

< Weakness >

- Main concern
    - Although the method is theoretically attractive, the proposed method doesn’t address the core motivation of avoiding averaging out minority perspectives. The paper didn’t demonstrate how the proposed method can address and model minor preference more accurately either theoretically or empirically.
    - Similarly, The motivation of declining public willingness to participate in surveys for specific regions/demographics (in line 69-72) raises a question that are not empirically proven. If certain groups are underrepresented in training data, how can the proposed method reliably emulate their preferences?
- About the metric and evaluation
    - One of the main statement, "captures both scope and distribution of preferences" in abstract, is not empirically supported. The paper motivates the work around preserving minority preferences, but the evaluation metric (MSE on averages) cannot verify this claim. The evaluation metrics used, MSE, cannot guarantee capturing the true distribution's variance or tail behaviors.
    - The proposed stage1 mostly focus on generating diverse persona, and entropy is also used for the evaluation metrics. Unlike simple data construction, higher entropy alone doesn't guarantee distributional fidelity as the paper is aiming of emulating human preferences. High standard deviation in generated samples may not reproduce the true population distribution. The method maximizes entropy without theoretical justification for why maximum entropy leads to better population representation.
- About design choice
    - While the functional perspective sidesteps one-to-one mapping requirements, the latent personas lose the practical benefit interpretability.
    - As state in Line 214, "attributes are typically viewed as reflections of latent preferences" in humans, yet P2P uses them generatively for models. This inversion needs stronger justification as the paper previously concretely define the decision-making process in Equation 1. What is the advantage of generating the basis with this inversion?
- About the experiment
    - Based on the concept of generating multiple persona, simple embedding clustering (e.g., k-means on persona embeddings) could serve as a stronger baseline than the current Vanilla approach
    - Table 1 shows Elastic Net performs better, contradicting the method section's focus on Lasso for selection and keeping essential basis. Elastic Net is not strictly a variable selection method.
- Concern on clarity
    - The paper introduces extensive new terminology (mode, variability score, tracker, attribute learner, question patching, mixed mode) without sufficient intuitive grounding. Table 6 only shows high-level categories. Additional material such as an intuitive example for a single scenario or unified flow diagram showing how all components interact, or table that summarizing the concept would greatly improve the clarity.
- Minor issues
    - I guess the sentence should be combined to be more fluent, “As the prompt space is far too vast to explore in an unstructured manner. To address this, we adopt a structured prompting strategy based on attributes” (Line 205)

**Questions:**

- Lines 473-474 claim the method can support "survey design, question testing, and nonresponse mitigation", which is valuably practical, but provide insufficient detail on implementation. How would practitioners actually deploy the proposed method for these applications?
- The motivation stresses avoiding “averaging out” minority perspectives, but the evaluation focuses on MSE of means. Could authors add metrics or analyses that directly test this recovery? Simple toy experiment with ground truth sub-groups would be sufficient.
- Table 1 shows that Elastic net performs better. Do authors have a explanation on that, as it contradicts to the role of Stage2? If the authors think Enet is better method than Lasso on their purpose, it would be better to modify abstract/intro, as they are focused on Lasso method.
- Does the author has a rationale of Stage1 compared to simple approach such as generating multiple prompts and combining them with simple embedding clustering? It would be even greater if author can add it as a baseline.

---

> ### Author Response · Authors · 2025-11-19
> **Response to Reviewer qjZY Part 1/4**
>
> Thank you for the thoughtful and detailed review. We are encouraged that you found the preference reconstruction theory sound, the separation between diversity generation and preference reconstruction meaningful, and the empirical results convincing, and we appreciate the concrete suggestions. We address the concerns and questions in order below.
>
> ---
>
> ## **Main Concern**
>
> ### **Minority perspective representation**
>
> We agree that avoiding the averaging-out of minority perspectives is central to pluralistic alignment. While ATP does not provide ground-truth subgroup labels at the level needed to *directly* verify minority recovery empirically, we do study this question in **Simulation Study 1 (Appendix B)**, where the ground truth is fully known.
>
> There, we construct synthetic populations from a small set of proxy agents with varied weights (representing the percentages of these agent types in the ground truth population)—agents with small weights play the role of minority perspectives. We then ask whether Stage 2 of P2P (Lasso) can recover these true profiles from the full candidate pool. The results (Figure 4) show that:
>
> - When the true profiles are present in the pool and there are enough observations, **Lasso reliably re-selects them**, including those with low weights;
> - Only when observations are extremely scarce, or when a true profile is deliberately removed from the pool, does the method resort to reconstructing it via mixtures of other agents.
>
> This simulation thus provides constructive evidence that the P2P pipeline does not simply “average out” minority types when they are represented in the candidate basis; instead, the sparsity prior actively helps preserve them.
>
> ### **On underrepresented groups and declining participation**
>
> We fully agree that if certain groups are systematically underrepresented in survey data, any purely data-driven method—including P2P—faces fundamental limits. How far P2P can help depends on assumptions about preference dynamics:
>
> - **Relatively stationary preferences over time (expedient approximation).** If a subgroup has reliable historical survey data, one can fit P2P on the *past* subgroup responses and use the learned ensemble to approximate their current responses on a new wave.
> - **Non-stationary preferences (cross-survey generalization).** If preferences drift, a more appropriate use is to fit P2P on an *easier* or higher-response survey for the subgroup and transfer the ensemble to predict on harder, low-response instruments.
>
> To rigorously assess reliability in either case, two ingredients are needed: (i) subgroup-specific panel data that track the same individuals or well-identified subpopulations over time, and (ii) a semantic question-partition module to structure cross-survey generalization by topic/domain. P2P is designed as a backbone that can plug into such richer data and semantic modules, but fully addressing nonresponse and subgroup undercoverage is an important direction for future work rather than a claim of the current paper. We will soften the phrasing in the revised paper to avoid confusion.
>
> ## **About the metric and evaluation**
>
> ### **On “scope and distribution of preferences” and evaluation metrics**
>
> We agree that a claim about capturing “scope and distribution” should not rest on a single scalar such as MSE. Two clarifications and additions address this concern:
>
> - **What our MSE actually measures.** In our setup, MSE is not computed on a single averaged score per question, but on the option-level probabilities after binarizing each multiple-choice item. Each question–option pair forms a separate dimension, so MSE penalizes misallocation of probability mass across the full categorical distribution, including minority options, rather than just matching a mean.
> - **Additional distributional metrics.** In the revised paper, we report Jensen-Shannon Divergence (1-JSD), majority-class prediction accuracy (MCPA) and Earth Mover Distance (EMD) alongside test MSE on ATP and WVS. Across settings, 1-JSD and EMD track the same performance ordering as MSE, indicating that the improvements are not restricted to means but reflect better distributional fit more broadly.
>
> (To be continued)

---

> ### Author Response · Authors · 2025-11-19
> **Response to Reviewer qjZY Part 2/4**
>
> (Continued from previous comment)
>
> Beyond these real-data metrics, **Simulation Studies 2 and 3 (Appendix B)** probe how well P2P (Stage 2) captures full response distributions when the true population types are *not* themselves in the proxy pool. In these simulations, the ground-truth population is constructed as a known mixture over a small set of synthetic agents, including components with small weights (conceptually, “minority” preference types), similar in spirit to Simulation Study 1 (see "Minority perspective representation"). The key difference is that here Lasso can only select and place weights on a separate set of proxy agents. The ground-truth types are never directly for selection. We evaluate fit at the distribution level, comparing the selected ensemble's categorical response distributions to those of the ground truth. We observe that when the number of observations is sufficient and the proxy pool is expressive (high entropy coverage), **P2P (Stage 2) achieves low distributional error, matching the ground-truth response distributions well, including their low-probability tails up to sampling noise**. When data are scarce and the proxy basis under-covers the ground-truth entropy (a learnability constraint), performance degrades gracefully rather than exhibiting a sharp collapse in distributional fit.
>
> Taken together—the option-level MSE, added 1-JSD/MCPA/EMD on ATP/WVS, and the synthetic experiments with known ground truth—these results support our use of the phrase “captures both scope and distribution of preferences,” while we agree that more targeted subgroup analyses on real panels remain an important direction for future work.
>
> ### **Entropy as a design signal (Stage 1)**
>
> We agree that “higher entropy alone doesn’t guarantee fidelity.” Our intent is not to maximize entropy unconditionally, but to use entropy as an operational proxy for response variability given the question set, in the absence of direct access to latent preferences. Here we clarify two points:
>
> - **Theoretical link**: In our revealed-preference framing, the observable object is the response distribution on a fixed question. When we cannot observe latent preferences directly, the variability of answers across agents is the main signal of how “rich” the induced preference basis is in that region of question space (cf. Fig. 1 right panel and Eq. (3)). Entropy is a natural scalar summary of this variability.
> - **Empirical justification**: Across the provided empirical results tables that contain average entropy (e.g., Table 5) and new experiments added during rebuttal, higher average entropy correlates negatively with test MSE (and 1-JSD & EMD). This empirical link is present across the evaluated ATP and WVS panels.
>
> Despite this relationship, in the conclusion of Appendix B.3, we explicitly caution against solely relying on entropy as a measure of preference diversity during the generation stage. Designing a diverse set of metrics for the endowment generation stage is a promising future work direction. We think adding a semantic module to P2P (Appendix C conclusion) is an important future development goal. We will highlight this more clearly in the revised version.
>
> ## **About design choice**
>
> ### **On the interpretability of the ensemble selected by P2P**
>
> We agree that interpretability is important, and we **deliberately avoid** treating proxy agents as literal human surrogates. As we note in the Ethics statement, P2P is a methodological tool for *preference reconstruction from aggregate data*, not a profiling system. This caution applies not only to our proxies but also to any demographically conditioned persona: even if a prompt mentions “age 18–29, high-school education,” the resulting agent is still a model construct, not an representative agent for a real individual or group.
>
> From an interpretive standpoint, P2P is closer to survey sampling than to one-to-one demographic mapping. In cross-sectional surveys, two carefully drawn samples (taken randomly from the target population) can yield nearly identical aggregate distributions on the same questionnaire, even though the underlying individuals differ. What matters for inference is the distributional fit, not the identity of each respondent. Our functional-basis view mirrors this: **the learned ensemble is best seen as a re-sampled survey population in function space**, constructed so that its aggregate responses match the observed distributions. Practically, each proxy is still tagged by its attribute configuration and mode, so practitioners can inspect which kinds of profiles (e.g., particular mode–attribute combinations) carry weight in the ensemble, giving interpretability at the level of which directions in attribute space matter, rather than at the level of named demographic “personas.”
>
> (To be continued)

---

> ### Author Response · Authors · 2025-11-19
> **Response to Reviewer qjZY Part 3/4**
>
> (Continued from previous comment)
>
> ### **Justification on using attributes as the control handles**
>
> We appreciate the reviewer’s concern about “inverting” attributes, from reflections of latent human preferences to generative handles for LLMs (Sec. 3.2). In our setting, this inversion is deliberate and grounded in how LLMs work: they are **conditional probability models**, and prompts (including attributes) act as control variables that modify the output distribution. Here, we want to highlight two practical reasons why our prompt strategy is based on these attributes:
>
> - A structured attribute bank gives us a systematic, interpretable coordinate system on prompts, instead of ad-hoc prompt hacking. It constitutes **a principled way of doing prompt engineering**.
> - The active sampling logic (by mode / template) lets us track which parts of this attribute space the backend is actually sensitive to, **turning attributes into a measurable handle on steerability**.
>
> ## **About the experiment**
>
> ### **Clustering as a baseline**
>
> The reviewer’s suggestion of a clustering-based variant is reasonable, and we implement it as an additional baseline. Concretely, we first expand the initial pool of personas for Vanilla and PERSONA to 1000, computing their embeddings (via OpenAI’s text-embedding-3-large), cluster them (k=300), pick the personas (medoids) closest to the centroids to form the candidate pool, then run our regression stage on the resulting pool. Table R4 (below) shows results on ATP W42 for a representative run:
>
> **Table R4. Clustering does not improve baseline quality.**
>
> | **Metric** | **Vanilla** | **Vanilla (clustering)** | **PERSONA** | **PERSONA (clustering)** | **P2P**  |
> | --- | --- | --- | --- | --- | --- |
> | **Avg. Entropy ↑** | 0.3172 ± 0.0069 | 0.3162 | 0.2871 ± 0.00026 | 0.2844 | **0.4451 ± 0.0256** |
> | **Test MSE (Lasso) ↓** | 0.0285 ± 0.00490 | 0.0266 | 0.0362 ± 0.00116 | 0.0395 | **0.00946 ± 0.00241** |
> | **Test MSE (Elastic Net) ↓** | 0.0286 ± 0.00476 | 0.0266 | 0.0339 ± 0.00124 | 0.0387 | **0.00792 ± 0.00261** |
> | **Cost (USD) ↓** | **0.8649 ± 0.0070** | 0.883391^ | 1.7129 ± 0.000016* | 1.7203*^ | 0.9708 ± 0.0036 |
>
> *Only considering survey elicitation cost on 300 personas—actual cost is higher.
>
> ^Excluding embedding cost.
>
> Clustering yields a test MSE for Vanilla (0.0266) that lies well within the variability band of the original Vanilla baseline (0.0285 ± 0.0049) and slightly worse MSE for PERSONA, with no conspicuous change on entropy and a slight increase in cost (for Vanilla due to extra generation). Hence, **clustering does not yield a stronger baseline**.
>
> This matches our intuition for two reasons. First, when we apply clustering *on top of* the existing baselines (starting from the same 300 endowments/personas), it can only **discard information that the regression step could have used**. In other words, it weakens the candidate pool without giving L1 regression any new capability. Second, even in the *more lenient* setting where we give clustering a head start by generating 1000 candidates and then compressing them to 300, the mechanism still does not produce a stronger baseline. As **clustering operates in embedding space and tends to group personas by textual or stylistic similarity**, it does not necessarily track **preference-relevant variation** in survey responses. As a result, minority or low-density preference types are prone to being averaged away in the embedding clustering, whereas our regression layer selects proxies directly based on their contribution to matching the observed response distributions.
>
> Viewed jointly, this suggests that our original baselines are not weak, and in fact are more cost-effective, avoiding an extra clustering step that adds computation without improving alignment quality.
>
> ### **Elastic Net clarification**
>
> **Elastic Net is still a variable selection method**: it combines an L1 penalty (as in Lasso) with an L2 shrinkage term. Lasso is thus the special case where all regularization weight is on the L1 term. In our setting, the choice between Lasso and Elastic Net reflects a trade-off between **parsimony** and small **performance gains**. Elastic Net can be advantageous when candidate proxies are highly correlated, but it typically selects a larger ensemble. In our experiments, Elastic Net achieves good test MSEs but with more nonzero agents, whereas Lasso attains comparable performance with a sparser basis. For this reason, we present Lasso the default method and use Elastic Net as a robustness check, rather than as a conceptual change to the framework.
>
> (To be continued)

---

> ### Author Response · Authors · 2025-11-19
> **Response to Reviewer qjZY Part 4/4**
>
> ## **Concern on clarity & minor issues**
>
> We agree that the terminology load can make the first read heavy, and we appreciate the suggestion. Our current submission includes Figure 2 as a unified flow diagram, and Appendix D.3 provides step-by-step algorithms for Active Endowment Generation and the tracker update, together with example endowments (D.4) and Table 6 summarizing the attribute bank. In the revision, we will add additional illustrative examples that walks through endowments generated under different attributes, tracing back to their modes and templates to aid intuition.
>
> ## **Q1: How would practitioners actually deploy the proposed method for applications?**
>
> Our primary goal in this paper is to provide a simple, data-efficient framework for population-level preference reconstruction from existing survey data, not to claim a fully productized tool for survey operations. We agree that large-scale deployment would require caution, additional work and careful validation, and we will clarify the phrasing in the conclusion to present these applications as promising directions rather than current capabilities.
>
> At the current stage P2P is best viewed as a *low-stakes design aid* for researchers. Given an existing survey (e.g., on trust in science), a practitioner can: (i) fit a P2P ensemble on past responses; (ii) use the fitted proxies to answer candidate new questions or variants; and (iii) inspect response variability and entropy to flag redundant or uninformative items before fielding them with humans. We see on-campus studies and behavioral-lab experiments as natural testbeds for this kind of usage and, in line with our Ethics Statement, we emphasize that such applications should be treated as **exploratory aids for survey design rather than as substitutes for real human respondents**.
>
> ## **Q2: Experiment testing the recovery of minority perspectives.**
>
> See the ‘On underrepresented groups and declining participation’ under ‘Main Concern.’
>
> ## **Q3: Clustering Experiment.**
>
> See ‘Clustering as a baseline’ under ‘About the experiment.’
>
> ---
>
> We thank the reviewer again for engaging carefully with both the theory and the pipeline design. We hope the additional experiments, clarified metric choices, and strengthened discussion of entropy, baselines, and deployment adequately address the concerns raised and improve your overall assessment of the paper.

---

### Official Review · Reviewer_S997 · 2025-10-30

**Soundness:** 3
**Presentation:** 3
**Contribution:** 3
**Rating:** 4
**Confidence:** 3

**Summary:**

Pluralistic alignment is a relatively new research area within LLM alignments. These methodologies attempt to alignLLMs with a diverse set of human preferences in comparison to standard LLM alignment methodologies which attempts to align LLMs with a single gold-standard set of human preferences. In this paper, the authors formulate a two staged process for pluralistic alignment - (1) an endowment generation process that attempts to form a functional basis for the human responses (2) a regression based aggregation which attempts to ensure that the net aggregation models the population level preferences. The paper examines this methodology using the revealed preference theory.

Through empirical experiments, the authors show that the two staged process shows better results when measured by MSE in comparison to baselines - (1) a single agent and (2) 300 agents from the PERSONA dataset. The paper further shows ablation studies over various design choices and several LLM backend

**Strengths:**

1. The novelty in this work is clear. The two staged process - using an endowment generation process to form a functional basis over the human preferences followed by a regression step to aggregate the agents responses is a novel idea.
2. The empirical results shows that this method shows better alignment in comparison to the baselines selected in the paper

**Weaknesses:**

1. A strong limitation of this method is that this is limited to survey style problems with multiple choice questions. It is not clear how this method could be applied to problems which require free text generation
2. As discussed in the related works, there are existing papers tackling pluralistic alignment problems. It is not clear how this solution performs relative to existing works. I would encourage the authors to add empirical results comparing this performance to Chen et al 2025, Feng et al 2024, Sorensen et al 2024b.
3. There is room for improvement in presentation of this paper. It is particularly difficult to parse through Sections 3.2 and 3.3. The authors could formalize the end-to-end algorithm which would make it easier to understand the proposed solution.

**Questions:**

1. Why have LLMs led to a shift away from SFT and RLHF? [Lines 033-035]
2. How can this method be applied to LLMs where free text generation is required?
3. Are the modes in the experiments consistent across all three models - the vanilla baseline, the PERSONA dataset and P2P model?

---

> ### Author Response · Authors · 2025-11-19
> **Response to Reviewer S997 Part 1/2**
>
> Thank you for your review and for highlighting the novelty of our two-stage P2P framework and its empirical gains over baselines. Below, we address the concerns and clarify a few points that may have been misunderstood.
>
> ---
>
> ## **Clarifying the baselines**
>
> We note that the reviewer’s description of our baselines as “(1) a single agent and (2) 300 agents from the PERSONA dataset” slightly mischaracterizes the setup, and is worth clarification. As noted in Sec. 4.1,
>
> - **Vanilla**: uses **300 agents** generated by a simple endowment-generation prompt that encourages diversity but does not use entropy-guided sampling or question patching.
> - **PERSONA**: uses **300 agents** drawn from an externally constructed, demographically conditioned persona set.
>
> In both cases, we apply the same regression-based aggregation layer as in P2P, so the comparison isolates the effect of P2P’s active endowment generation rather than differences in the downstream aggregator.
>
> ## **W1 & Q2: Free-text generation**
>
> Our design deliberately starts from **categorical survey responses** because they provide a clean and standard alignment yardstick for population preferences (cf. lines 116-117 of Sec. 2). We agree with the reviewer that extending the output format to free-form text can greatly broaden the framework’s empirical applicability. The main challenge there is to **aggregate** agent-level text outputs in a way that reflects the learned selection weights. As discussed in the Appendix F ("Survey data in, text out"), a promising direction is to draw on mechanism-design tools, e.g., auction-style aggregation schemes (Dütting et al., 2024), to effectively synthesize weighted textual responses. Fully developing such text-level aggregation is beyond the scope of this paper, and we therefore focus here on the categorical setting as a controlled testbed, and a natural first step.
>
> ## **W2: Relation to Chen et al. 2025, Feng et al. 2024, Sørensen et al. 2024b**
>
> We thank the reviewer for pointing to these works. As noted in the paper’s introduction and discussed more explicitly in **Appendix E**, prior work on pluralistic alignment varies substantially both in **how pluralism is formalized** and in **what data each method requires**. Our target problem—recovering *population preference from anonymous survey responses*—differs in both data and objective from the cited papers, which prevents a meaningful empirical head-to-head. Here we briefly clarify these mismatches for each paper:
>
> - **Chen et al. 2025** requires per-user interaction histories and trains a personalized embedding for each user. Our data contain no user identifiers and only provide aggregate choice distributions, making the method incompatible without major modification.
> - **Feng et al. 2024** (distributional pluralism) is closest conceptually, but their method assumes community-specific LMs trained on labeled community corpora and known population priors. Our setting lacks community corpora with community corpora, so the method cannot be instantiated as a baseline.
> - **Sorensen et al. 2024b** distills value structures from a powerful closed-source LLM (e.g., GPT-4). Our framework can already leverage strong models when needed (for instance we also use GPT-4 model in 4.4), and their contribution centers on a new value-dataset plus model distillation, while not on learning from sparse, aggregate human preference data. Thus it is not parallel to our task.
>
> For these reasons, the referenced works cannot serve as direct empirical baselines in our setting. Instead, we add a head-to-head comparison with Cao et al. (2025), which does operate on aggregate survey distributions, as a closer reference point (**see our response to Reviewer XZ1y and the new results in Table R1**).
>
> ## **W3: End-to-End Algorithm**
>
> We agree with the reviewer that a clear view of how the components fit together is important. Our current submission includes:
>
> - Figure 2, which provides a high-level overview of the two-stage P2P pipeline and evaluation framework.
> - Appendix D.3, which offers formalized step-by-step algorithms for Active Endowment Generation and the tracker update procedure, together with Appendix D.1 affording a structured view of the attribute bank.
>
> We will make the references more prominent in the paper. To further improve clarity, In the revision we will add additional illustrative examples of generated endowments, highlighting their underlying attributes, modes and templates.
>
> (To be continued)

---

> ### Author Response · Authors · 2025-11-19
> **Response to Reviewer S997 Part 2/2**
>
> (Continued from previous comment)
>
> ## **Q1: “Why have LLMs led to a shift away from SFT and RLHF?”**
>
> Our intent here was not to claim that SFT/RLHF are obsolete, but to describe a practical trend: many applications now favor prompting-based adaptation (ICL, lightweight instruction tuning) because full SFT/RLHF require substantial supervised data, infrastructure, and access to model internals, whereas prompting is cheap, fast, and works with closed-source APIs. In Appendix E.1, we contextualize this trend within pluralistic alignment. We will soften the phrasing to avoid confusion.
>
> ## **Q3:  “Are the modes in the experiments consistent across all three models?”**
>
> **Modes** (core/thematic/theoretical etc.) are specific to P2P’s Active Endowment Generator in Stage 1 and control how we sample and expand attributes. The Vanilla and PERSONA baselines do **not** use these modes; they differ only in how endowments are constructed (see “Clarifying the baselines”), while sharing the same regression-based aggregation and evaluation protocol.
>
> ---
>
> We thank the reviewer again for highlighting both the novelty and empirical promise of the two-stage P2P framework, and we hope these clarifications address your concerns and improve your opinion on the paper.

---

### Official Review · Reviewer_1dZo · 2025-10-31

**Soundness:** 1
**Presentation:** 1
**Contribution:** 1
**Rating:** 2
**Confidence:** 4

**Summary:**

In order to predict population-level human averages to survey questions, the paper learns an ensemble of agents (defined by prompts steering to diverse attributes). It then fits a regression on top of the agents' responses to approximate the true population average. The authors evaluate their method on data from OpinionQA.

**Strengths:**

The paper attempts to recover the population-level average survey responses through a weighted basis of "agents", which is a potentially interesting idea, though there are several weaknesses in the execution, listed below.

**Weaknesses:**

# Minimal theoretical contribution
In the introduction, the paper states as one of its contributions that it provides a "theoretical foundation" for their approach. However, the theoretical content presented in Section 3 is superficial.

For example, the authors state "A direct consequence is that if human population preference is learnable in the revealed sense, so is individual human preference—we defer the proof to Appendix A."  However, it does not make sense that they "defer the proof" because the deferred proof is one line and follows directly from the Defn 2.2. There is no need to create a proposition for this; the authors can simply state this observation.

Similarly, Thm 1 and Thm 2 also follow directly from Defn 2.2.

Additionally, the theorems are not written in a formal manner. For example, the statement “If human population preference is learnable in the revealed sense, the ensemble of LLM agents that can learn this is not unique” lacks preciseness. In general, clarity could be significantly improved.

Overall, the theoretical contributions are minimal and do not add substantial value to the paper.

# Greater contrast to distributional alignment
The authors focus on developing a method that matches the population mean of survey responses. However, this appears to be strictly less powerful than constructing a model whose responses are calibrated to the full distribution of the population, as one could simply sample from such a model and compute the average. Maybe there is a computational efficiency argument to be made here about requiring fewer samples from the model, but I think the authors need to do more to justify why they take their approach and better situate their work in the literature on distributional alignment.

# Experiments

The authors compare their method to two baselines, both of which fit within the overall framework of their approach: (1) constructing a set of diverse agents, and (2) running a regression on the agents' responses to match the observed population mean. The two baselines differ only in how the agents are constructed in step (1), and both use less comprehensive methods than the authors’ full approach. The evaluation would be more compelling if the authors compared their method to entirely different approaches, such as the one proposed in Cao et al. (2025), which is already cited in the paper. It would also be helpful to report the performance of the original model backend to better contextualize the potential improvement.

It is strange to me that the authors evaluate their method across 14 waves, but only evaluate against the baselines for a much smaller subset---just one of the waves. This raises the question of whether the method actually improves compared to the baseline across the larger dataset. I recommend that the authors provide baseline results for all waves to allow for a more comprehensive comparison.

Additionally, the evaluation is limited to OpinionQA, which focuses solely on the American population. This raises questions about the generalizability of the method to other populations. The evaluation could be strengthened by including results on GlobalOpinionQA and assessing the method’s ability to approximate population responses across different countries. If the experiments are not possible, this should at least be acknowledged as a potential limitation.

**Questions:**

- What model is Table 1 using?

Btw since the paper is focused on human attributes, this paper might also be a relevant reference:
"PrefPalette: Personalized Preference Modeling with Latent Attributes" Li et al (2025)

---

> ### Author Response · Authors · 2025-11-13
> **Response to Reviewer 1dZo: Clarifying Theoretical Contributions and Ensemble Expressiveness**
>
> Thank you for taking the time to read our paper and provide detailed feedback. We are currently reviewing all experimental suggestions raised across the reviews and are running additional analyses and baseline comparisons. We will update the discussion with results as they become available during the rebuttal period. In the meantime, we would like to clarify several theoretical misunderstandings that appear to have shaped the reviewer's concerns.
>
> ---
>
> ## **On the observation following Definition 2.2**
> Thank you for pointing this out. We agree that the observation follows directly from Def. 2.2. In the main paper, we present it informally as a one-line consequence of the definition, and restate it as Proposition 1 in the appendix for completeness and parallel structure. The intent was not to elevate it as a technical contribution, but to make explicit why our analysis focuses on *population preference*: if an alignment method works at the population level, it also works at the individual level as a special case. This is the premise on which the subsequent theorems build.
>
> ## **On the style and substance of Section 2**
> Our goal in Section 2 is to provide a clear conceptual framework for pluralistic alignment rather than mathematically heavy results. Theorem statements are written in natural language to remain accessible across ML, social-science, and HCI audiences, while the proofs are fully formal and appear in Appendix A.
>
> Although Proposition 1 is immediate, the remaining results do not follow directly from Def. 2.2:
>
> - Theorem 1 relies on the convexity property established in Theorem 0 (Appendix A);
> - Theorem 2 requires convex decompositions of aggregate responses;
> - Theorem 3 shows non-uniqueness of ensembles.
>
> None of these are derivable from Def. 2.2 alone. These theorems are included in the main paper because they *unify representative-agent models (trained via SFT/RLHF), demographic conditioning, and functional basis reconstruction under one revealed-preference framework*. Theorem 3 also motivates the optimization objective used in P2P.
>
> We can add formal versions of the theorem statements in the appendix if preferred, though we believe the current presentation offers the clearest exposition for a broad audience.
>
> ## **On ensemble expressiveness**
> The reviewer's concerns about the method's "powerfulness" relative to a distribution-calibrated model relate to the distinction between individual and population preference learnability. The observation following Def. 2.2 (Proposition 1) highlights this distinction, and we clarify its consequence below.
>
> **Crucially, the reverse of Proposition 1 need not hold:** individual preferences may lie in a model's learnable space $r_i \in f(\widehat{P})$, while the population preference does not $r_{\mathrm{pop}} \notin  f(\widehat{P})$ (e.g., when $ f(\widehat{P}) $ is non-convex). In such cases, no single representative model can recover $ r_{\mathrm{pop}} $. In contrast, because ensembles represent $ \mathrm{conv}(f(\widehat{P}))$, the ensemble can recover $ r_{\mathrm{pop}} $ exactly via convex combinations of model-learnable individuals. This shows that the ensemble approach is strictly more expressive than single-model distributional alignment, even before considering its benefits for interpretability and sparsity.
>
> ## **On the "mean vs. distribution" misunderstanding**
> The reviewer's characterization that P2P "matches the population mean of survey responses" reflects a misunderstanding of how the method behaves. In the ensemble, each proxy agent specifies a categorical distribution over response options, and the learned weights induce a **mixture distribution** over these agents. In practice, we set the agent temperature to 0 for stability and reproducibility, so each proxy behaves as a (near-)deterministic policy from questions to options, and the resulting mixture over many such agents yields a full distribution over response options for each question. At inference time, the ensemble defines a generative distribution from which answers are sampled, as with a distribution-calibrated model.
>
> Our regression objective likewise does not operate on means: after binarizing each multiple-choice question, the loss is computed over option-level probabilities, ensuring distributional matching. This is also reflected in our evaluation metric (MSE). Following the reviewers' suggestions, we will additionally report metrics such as 1-JSD and majority-class prediction accuracy to facilitate comparison across works.
>
> Put simply, a distribution-calibrated model is a **degenerate one-agent ensemble**, whereas mixtures are **strictly more expressive** unless the model's learnable space is convex (Theorem 1).
>
> ---
>
> We hope these clarifications resolve the theoretical concerns, and we appreciate the reviewer's patience as we continue to work on the expanded experimental evaluation.

---

> ### Author Response · Authors · 2025-11-19
> **Response to Reviewer 1dZo: Addressing Experimental Concerns and Remaining Questions | Part 1/2**
>
> Beyond the theoretical clarifications above, we address the reviewer’s remaining concerns on baselines, coverage across waves, and generalizability.
>
> ---
>
> ## **On baselines and coverage across ATP waves**
>
> P2P is explicitly designed as a plug-and-play preference alignment method for low-data, low-compute settings, so our baselines intentionally focus on prompting-based ensemble methods under the **same constraints** (Sec 4.1), rather than SFT-style approaches that require large training runs. Vanilla uses a single topic-conditioned endowment prompt that encourages diversity but no adaptive sampling or question patching; PERSONA draws endowments from an externally constructed, **demographically conditioned** persona set. In both cases, we apply the same regression-based selection as in P2P, which **strengthens the baselines and isolates the effect of active, entropy-guided endowment generation**.
>
> We agree that showing baselines on only one ATP wave is not ideal. The original choice was driven by budget constraints and the need for a single “anchor” wave for ablations, not ex post cherry-picking.  In the revision, we extend Vanilla and PERSONA to three representative ATP waves chosen to span the range of difficulty observed across all 14 waves where we ran P2P (one relatively weakly aligned, one medium, one strongly aligned). For these three representative waves, we report mean ± std over 3 independent runs (Table R2). **In all cases, P2P attains higher entropy and lower test MSE than both baselines at comparable or lower cost, consistent with our finding on ATP W42**.
>
> **Table R2. Extended baseline comparisons on three representative ATP waves (weak/medium/strong alignment).**
>
> | Wave | Method | Avg. Entropy ↑ | Test MSE (Lasso) ↓ | Cost (USD) ↓ |
> | --- | --- | --- | --- | --- |
> | 27 | Vanilla | 0.4623 ± 0.0066 | 0.0322 ± 0.0045 | **0.6314** ± 0.0037 |
> |  | PERSONA | 0.4016 ± 0.0001 | 0.0412 ± 0.0006 | 1.2562 ± 0.00004 |
> |  | P2P | **0.5363** ± 0.0099 | **0.0218** ± 0.0010 | 0.7106 ± 0.0039 |
> | 32 | Vanilla | 0.5514 ± 0.0055 | 0.0192 ± 0.0005 | **0.6363** ± 0.0035 |
> |  | PERSONA | 0.5569 ± 0.0002 | 0.0252 ± 0.0004 | 1.2805 ± 0.00002 |
> |  | P2P | **0.6178** ± 0.0153 | **0.0160** ± 0.0019 | 0.7241 ± 0.0012 |
> | 45 | Vanilla | 0.3888 ± 0.0021 | 0.0153 ± 0.0025 | **0.6214** ± 0.0036 |
> |  | PERSONA | 0.3645 ± 0.0011 | 0.0288 ± 0.0014 | 1.2397 ± 0.00001 |
> |  | P2P | **0.4450** ± 0.0060 | **0.0079** ± 0.0023 | 0.6928 ± 0.0019 |
>
> Full metrics (including Elastic Net, 1-JSD, MCPA, and EMD) for these three waves are included in Appendix G of the revised paper.
>
> ## **On comparison to “entirely different approaches” (Cao et al.) and backend performance**
>
> ### **Cao et al. comparison**
>
> We agree that it is valuable to compare against methods outside our ensemble framework. Following the reviewer’s suggestion, we now evaluate P2P in the WVS Wave 7 setting of Cao et al. (2025) and compare directly against their supervised model on its own arena (**see our reply to Reviewer 1dZo**).
>
> Cao et al. train a single multi-country SFT model on 6841 training and 1586 validation entries pooled across 46 countries, conditioned on locale, to predict answer distributions directly. In contrast, P2P learns a sparse mixture of proxy agents separately for each locale, using only the local entries and no fine-tuning. On the three representative locales (US, Great Britain, Hong Kong SAR), **P2P achieves comparable performance despite using far fewer training entries per locale and no fine-tuning**.
>
> ### **Backend performance**
>
> Following the reviewer’s suggestion, we report backend performance as a baseline by directly querying the model backend (gemini-2.0-flash) for answers to the survey questions. The instruction prompt we used is
> > *“You are completing a public opinion survey. Answer each question as an average adult respondent in the US population…”*
> >
> We draw 300 Monte-Carlo samples per question with temperature set to 0.7. This results in an empirical answer distribution per question. On ATP W42, this baseline attains test MSE 0.0918 and average entropy 0.0161, conspicuously worse than the results for the ensemble baselines, **confirming the gains via ensemble methods** (Vanilla, PERSONA, and P2P).
>
> (To be continued)

---

> ### Author Response · Authors · 2025-11-19
> **Response to Reviewer 1dZo: Addressing Experimental Concerns and Remaining Questions | Part 2/2**
>
> ## **On generalizability beyond US-only Opinion QA / GlobalOpinionQA**
>
> We share the reviewer’s concern about US-centrism. While running P2P on the full multi-country GlobalOpinionQA dataset is infeasible within the rebuttal window, we follow the suggestion in spirit by evaluating P2P on the **WVS Wave 7 data** released by Cao et al., which are derived from the same source as the WVS portion of GlobalOpinionQA. This provides a cross-national, non-ATP benchmark built from GlobalOpinionQA’s underlying survey family.
>
> We repartition the questions using a standard 7:1.5:1.5 random split over questions (rather than their original Q1/Q2→Q3 topic split; cf. W2(b) in our reply to Reviewer **XZ1y** for reasons on this design choice) and run P2P on three representative locales (US, GB, HK), reporting mean ± std over three independent runs (Table R3). **P2P achieves low test MSE (≈ 0.014–0.024) and high 1-JSD (≈ 0.76–0.81), in line with our findings on ATP**.
>
> **Table R3. P2P performance on WVS Wave 7 (US, Great Britain, Hong Kong SAR) under a 7 : 1.5 : 1.5 question split and using gemini-2.0-flash as the backend.**
>
> | **Metric** | WVS W7 US | WVS W7 GB | WVS W7 HK |
> | --- | --- | --- | --- |
> | **Avg. Entropy ↑** | 0.5720 ± 0.0149 | 0.5710 ± 0.0044 |  0.5478 ± 0.0211 |
> | **Test MSE (Lasso) ↓** | 0.0149 ± 0.0005 | 0.0141 ± 0.0014 | 0.0242 ± 0.0013 |
> | **Test 1-JSD (Lasso) ↑** | 0.8030 ± 0.0055 | 0.7895 ± 0.0049 | 0.7590 ± 0.0046 |
> | **Test MCPA (Lasso)  ↑** | 0.6667 ± 0.0152 | 0.6481 ± 0.0321 | 0.5789 ± 0.0263 |
> | **Test EMD (Lasso)  ↓** | 0.1075 ± 0.0035 | 0.1038 ± 0.0081 | 0.1424 ± 0.0034 |
> | **Test MSE (Elastic Net) ↓** | 0.0147 ± 0.0005 | 0.0137 ± 0.0010 | 0.0244 ± 0.0005 |
> | **Test 1-JSD (Elastic Net)  ↑** | 0.8066 ± 0.0067 | 0.7950 ± 0.0073 | 0.7602 ± 0.0032 |
> | **Test MCPA (Elastic Net)   ↑** | 0.6930 ± 0.0152 | 0.6296 ± 0.0321 | 0.5877 ± 0.0304 |
> | **Test EMD (Elastic Net)   ↓** | 0.1066 ± 0.0025 | 0.1004 ± 0.0054 | 0.1430 ± 0.0046 |
> | **Cost (USD) ↓** | 1.7551 ± 0.0097 | 1.7095 ± 0.0031 | 1.7365 ± 0.0080 |
>
> Performance is higher for US and GB and somewhat lower for HK, which is consistent with broader findings that large language models trained primarily on English and Western-centric data tend to better match Western public opinion than non-Western populations. Nonetheless, these results, together with our 14-wave ATP evaluation, indicate that the framework is not tied to a single country or survey program.
>
> A full GlobalOpinionQA-wide, multi-backend study is beyond the scope of this paper and is a natural direction for future work, e.g., using P2P as a steerability benchmark (Appendix F).
>
> ## **Q1: “Original model backend” for Table 1**
>
> We thank the reviewer for flagging this. Table 1 uses **gemini-2.0-flash** as the common LLM backend for all three methods (Vanilla, PERSONA, P2P), chosen for its cost advantage (Table 5). All other experiments also use this as the default, except the backend ablation. We will state this explicitly in Sec. 4.2 in the revised version.
>
>
> ## **On PrefPalette (Li et al., 2025)**
>
> We thank the reviewer for pointing out PrefPalette. However, it addresses a different problem: PrefPalette predicts *instance-level content preferences* on Reddit, using latent attributes of the *text* (stylistic and affective properties) and attention-based weighting to decide which reply a community upvotes. In contrast, our work targets **population preference alignment based on survey data**. We do this via sparse mixtures over **endowed agents** that encode synthetic socio-demographic and value profiles of *people*, not properties of candidate texts. Because the data, supervision signal, prediction target, and even the meaning of “attributes” (content vs. agents) differ, PrefPalette is orthogonal to our setting and not a natural baseline or core related work for our task of survey-based preference alignment. We hope this further clarifies the distinction of our work.
>
> ---
>
> Thank you again for your reading and suggestions. We hope these clarifications and expanded results address your remaining concerns and help improve your assessment of our paper.

---

> > ### Comment · Reviewer_1dZo · 2025-11-26
> > **Unconvinced by theoretical contribution**
> >
> > I thank the authors for their detailed response and appreciate their added experimental results.
> >
> > However, I continue to disagree with the theoretical points. In my view, the proofs are unnecessarily complicated. Both Theorem 1 and Theorem 2 appear to follow directly from Definition 2.2 and, similar to Proposition 1, could be stated in approximately one line.
> >
> > Theorem 1:
> > - I do not believe “Thm 0” is needed.
> > - Def 2.2 says that \conv(h(P)) \subset f(P^)). Then, by Def 2.2, since r_pop \in f(P^), there must exist a p^ \in P^ such that f(p^) = r_pop. This is simply based on the definition of the range of a function.
> >
> > Theorem 2:
> > - Def 2.2 immediately implies that if the population is learnable, then individuals are learnable (what the authors call Prop 1). By associating each agent with each individual, Thm 2 follows.
> >
> > Please let me know if I've misunderstood. However, I remain unconvinced by the authors’ assertion that “the current presentation offers the clearest exposition for a broad audience.” There is a much simpler and more direct way to present these results, which the current manuscript does not employ.  I am concerned that these results are being presented as more substantial than they actually are.

---

> > > ### Author Response · Authors · 2025-11-26
> > > **Follow-up response to Reviewer 1dZo**
> > >
> > > We thank the reviewer for the positive feedback on the added experimental results, and for clearly articulating their reasoning about Theorems 1 and 2, which lets us zoom in for further discussion.
> > >
> > > First, we agree with the reviewer that Theorem 1 and Theorem 2 are **mathematically straightforward** once we assume the learnability of population preference (Def. 2.2), unpack the definition of the image (range) of $f$, and use the population decomposition in Eq. (2).
> > >
> > > In cleaned-up form, our proof for Theorem 1 is:
> > >
> > > - $\mathrm{conv}(h(P)) \subset f(\widehat P)$ (by Def 2.2; learnability assumption)
> > > - $r_\mathrm{pop} \in \mathrm{conv}(h(P))$  (by definition of the population response)
> > > - $\exists  \hat{p}_\mathrm{pop}  \in f^{-1}(\mathrm{conv}(h(P)))$ such that $r_\mathrm{pop} = f(\hat{p}_\mathrm{pop})$. (by Theorem 0)
> > >
> > > The step that the reviewer summarizes as “simply based on the definition of the range of a function” corresponds, in our appendix proof, to the application of Theorem 0 to $f$ restricted to $f^{-1}(\mathrm{conv}(h(P)))$. In other words, for Theorem 1 the logical core of this step is the same inclusion–range argument the reviewer sketches.
> > >
> > > We isolate Theorem 0 (Appendix A) as a separate result to emphasize the connection to representative-agent theory in economics (cf. implication 1 of Lemma 1 in Appendix A) and make explicit the structural link between properties of the preference-response mapping and the existence of representative agents. This also affords an interpretability angle on model preferences in a social-science context.
> > >
> > > In cleaned-up form, our proof for Theorem 2 is:
> > >
> > > - $\mathrm{conv}(h(P)) \subset f(\widehat P)$ (by Def 2.2; learnability assumption)
> > > - $r_\mathrm{pop} = \sum_{i} w_i r_i \in \mathrm{conv}(h(P))$  (by definition and convex decomposition)
> > > - $\forall r_i \in h(P) \subset \mathrm{conv}(h(P)) \subset f(\hat P), \exists \hat{p}_i \in \widehat{P}$ with $f(\hat{p}_i) = r_i$ (by Prop. 1 & range definition)
> > > - Hence, $r_\mathrm{pop} = \sum_i w_i f(\hat{p}_i)$,
> > >
> > > exactly as in the reviewer’s sketch.
> > >
> > > The proofs we include in Appendix A write out a few intermediate steps and natural language explanations to keep the exposition accessible to readers beyond pure theory. We can further tighten the proofs, following the reviewer’s advice.
> > >
> > > Second, as noted in the introduction of our main paper (Lines 76-79), **our theoretical contribution is the preference reconstruction theory and framework:**
> > >
> > > > **Theoretical foundation.** We formalize population preference alignment as a problem of representation learning, and establish the preference reconstruction theory, which states that learnable population preference can be represented using a functional basis of proxy agents and with appropriately determined aggregation weights.
> > > >
> > >
> > > Theorem 0, Proposition 1, and Theorems 1–2 are auxiliary observations within this framework, not standalone contributions, which is why Theorem 0 and Proposition 1 are already stated and proved in Appendix A. As stated in our earlier reply, our goal in Section 2 is to provide a clear conceptual framework for pluralistic alignment rather than mathematically heavy results. Theorems 1 and 2 are formalized and stated alongside Theorem 3 in the main paper because they *unify representative-agent models (trained via SFT/RLHF), demographic conditioning, and functional basis reconstruction under one revealed-preference framework*.
> > >
> > > Crucially, the preference reconstruction theory motivates Equation 3, the optimization objective that inspires P2P’s design logic.
> > >
> > > We hope this further clarifies our stance.

---

### Official Review · Reviewer_XZ1y · 2025-11-01

**Soundness:** 3
**Presentation:** 4
**Contribution:** 3
**Rating:** 6
**Confidence:** 3

**Summary:**

The authors propose Prompts-to-Proxies (P2P) as a way to reconstruct population preferences for LLM alignment. P2P is a two-stage framework for pluralistic alignment that 1) generates a diverse basis of LLM agents via attribute-driven prompts with entropy-guided sampling and 2) reconstructs a target population’s response distribution by learning weights over the agents outputs. The authors motivate this via preference reconstruction theory where they argue that population preferences are learnable by non-unique ensembles that span the revealed preference space. They also share empirical results comparing aggregated responses with the American Trends Panel survey showing better performance in MSE and avg entropy vs baselines.

**Strengths:**

S1. The two-stage framework is clearly defined and has a principled diversity objective. The active endowment generation is guided by normalized question entropy and a mode-level variability score over training questions, with adaptive sampling and question patching for targeting low-entropy questions. This approach seems simple and effective in relation to the pluralistic alignment goals and is well-backed up by the theoretical development of the preference reconstruction theory the authors present.

S2. The authors present useful ablations of the technique across endowment budget, the regression-based aggregation module, and different model backends. The finding that model performance is highly variable is not surprising and may contribute to challenges in replicability over the long-term but it seems the authors have tried to mitigate this well wherever possible.

S3. The research shows broad empirical results and the cost considerations help with adoption and impact assessment. Cost comparisons against PERSONA baseline support the claims of the authors on the cost-effectiveness of the approach.

S4. The authors present a useful analysis against a well-validated survey in the form of the American Trends Panel work from Pew. Although this work is restricted in applicability to the US, which already shows higher alignment from current model training than other areas in the world, the setup allows for easy scaling to other locales.

**Weaknesses:**

W1. Despite the strengths there are a number of weaknesses to the synthetic data generation approach and validation through survey methods. In particular: limited evaluation metrics or good diagnostics. There is not any analysis of the calibration of the responses against human reviewers or any other spot checking of the endowments to validate the metric-driven results.

W2. In section 4 the comparison set omits key survey response distribution simulation baselines. While PERSONA and a vanilla generator are useful references, recent methods closer in spirit like Cao et al (2025) “Specializing Large Language Models to Simulate Survey Response Distributions for Global Populations” are not empirically compared. A head-to-head on shared survey tasks would strengthen claims about efficiency and fidelity and inclusion of standard metrics like Jenson-Shannon divergence and Earth Mover Distance on the distribution accuracy would help validate.

W3. In the current paper there is some risk of overfitting to the survey artifacts of the American Trends Panel. Because training and selection operate over distributions of responses to specific items from the same panel, the agents could be learning survey-specific heuristics rather than stable latent preferences. Cross-dataset generalization (e.g., OpinionsQA -> a distinct polling corpus) is not shown and understanding if this method could scale to multiple survey types and structures would be very beneficial in validating the impact of this work.

W4. These evaluations are also all tied to multiple-choice labeling. The current pipeline only handles categorical options and binarizes heterogeneous scales to fit the regression system, implicitly up-weighting multi-point Likert items. This design choice can bias what the model learns as “preference,” and the authors themselves note reweighting/format transformation might be needed but no sensitivity analysis is conducted.

W5. The connection between model steerability and a given models’ ability to represent the response distribution for a given population mixture of human responses is not fully explored. Although ablations on different models are conducted. As shown in Kirk et al (2024) “The PRISM Alignment Dataset” there are quite a range of group dimensions beyond the core/thematictheoretical template groups which can be used as attributes for group representation. Different models have different default steerability across these dimensions. Developing a measure of model steerability for group attributes would help ground the model selection for this use-case and for the general problem of pluralistic alignment.

**Questions:**

Q1. How sensitive are results to the attributes selected for the attribute bank? Reporting a variance analysis where you 1) shuffle or replace entire modes 2) vary attribute counts per endowment or 3) ablate the free-form survey/question-derived attributes would help with understanding the impact of this selection. Measuring the change in entropy, sparsity, and test MSE would be useful?

Q2. The authors mention that response rates to surveys are dropping across regions and demographics (i.e. human behavior is continuing to change) and synthetic agents can offer a way to simulate public opinion. How can we continue to validate that the synthetic agents maintain their distributional representation over time if they are developed for a single moment-in-time?

Q3. Since a key part of pluralistic alignment is the ability to avoid excluding minority perspectives in larger groups, it would be helpful too understand if the P2P method has any benefits or regressions with respect to subgroups (e.g. age / education-level) in the case where that subgroup may not be represented directly in the ensemble.

---

> ### Comment · Reviewer_XZ1y · 2025-11-13
>
> After reading the other reviews I agree with a few additional points that have been raised:
> 1. 1dZo mentions the benefit of extending evaluations to GlobalOpinionQA instead of just the US-centric OpinionQA as well as discrepancy between which set of waves are used for evaluation of baselines vs the proposed method which calls into question some of the findings.
> 2. qjZY concerns about the framework design choice and the justification for using personas for generation
> 3. S997 and I agree on the need to add better comparisons against recent reference methods and related work
>
> Based on this I would not mind if the paper is rejected though I am open to hearing rebuttal from the authors.

---

> ### Author Response · Authors · 2025-11-19
> **Response to Reviewer XZ1y Part 1/4**
>
> Thank you for the incisive review and valuable suggestions, and for delineating the strengths and novelty of our paper (S1-S4). We address W1-W5 and Q1-Q3 in turn below.
>
> ---
>
> ## **W1: Limited evaluation metrics and diagnostics**
> We share the reviewer's view that richer diagnostics are important. In the revised paper, we **report Jensen-Shannon Divergence (1-JSD)**, **majority-class prediction accuracy (MCPA)** and **Earth Mover Distance (EMD)** (cf. Cao et al., 2025) in addition to test MSE. Across our experiments, the distributional metrics (1-JSD and EMD) track the **same performance trends** as MSE and lead to consistent conclusions. To extend Appendix D.4 ("Example Endowments"), we provide additional intuitive spot checks of generated endowments from different modes (cf. Figure 20 in Appendix G.7) in the revised paper, so readers can inspect the diversity and plausibility of the generated endowments.
>
> Important as it is, human evaluation is less natural in our setting. In our theoretical framework (Sec. 2), we explicitly assume that human's and model's preference spaces differ. Any human judgment on the "consistency" of agent output implicitly requires annotators to impersonate the model in the *model*'s preference space. For this reason, we view human evaluation more germane once P2P is extended to the "Survey data in, text out" setting (Appendix F), where quality of outputs can be evaluated not via consistency but through other established text-based protocols.
>
> ## **W2: Comparison with Cao et al. and additional metrics (1-JSD, EMD)**
> ### **(a) Why we did not include an SFT baseline on ATP**
> Our main empirical studies focus on prompting-based baselines rather than SFT-style models for both data and compute reasons. Cao et al.'s SFT model is trained on thousands of World Values Survey (WVS) aggregate response entries pooled across 46 countries (6841 train, 1586 val). In contrast, a single ATP wave has only ~60-150 questions (cf. Sec. 4 & Figure 18). With a 7:1.5:1.5 split, this yields roughly 42-105 training questions for the target population, which makes a very weak regime for a robust, population-specific SFT baseline. SFT also adds a separate finetuning phase and associated compute overhead. P2P is designed as a **plug-and-play preference alignment method** that can be deployed on new populations with **limited-data and limited-compute regimes**, so for the main results we restricted baselines to prompting-based methods that operate under the same constraints.
>
> ### **(b) Head-to-head comparison with Cao et al. on their arena**
> Although the different training regimes (Table R0) makes it hard for a fair comparison, we challenge P2P directly in Cao et al.’s own WVS setting, as it presents an interesting scenario outside our designed studies. We follow their protocol: Q1 and Q2 for train/validation and Q3 as test, which induces a **topic shift** from general attitudinal & religious/ethical items (Q1+Q2) to political interest and culture items (Q3). In our language, this asks: *if we align to a population on broad attitudinal dimensions (Q1+Q2), how well does the aligned ensemble generalize to a new unseen dimension (Q3: political-cultural questions)?*
>
> Due to time and budget limits during rebuttal, we could not run P2P separately on all 46 countries with multiple repeats. Instead, we choose three representative locales: US (overlaps ATP), Great Britain (Western Europe) and Hong Kong SAR (East Asia). For each, we filter Cao et al.’s data to that locale, train P2P (backend: gemini-2.0-flash; 300 endowments) on local Q1+Q2 (train+val), and evaluate on Q3. We report mean ± std over 3 runs in Table R1 below.
>
> **Table R0. Different training regimes for P2P vs Cao et al.**
>
> | Setup | **Train+Val Entries (Q1 + Q2)** | **Test Entries (Q3)** | **Model backend** |
> | --- | --- | --- | --- |
> | P2P (US) | 185 (150 + 35) | 59 | gemini-2.0-flash |
> | P2P (GB) | 172 (140 + 32) | 60 | gemini-2.0-flash |
> | P2P (HK) | 185 (150 + 35) | 59 | gemini-2.0-flash |
> | Cao et al. (C1: 46 countries) | 8427 (6841 +1586) | 2719 | Llama3-8B-Instruct |
>
> **Table R1. P2P vs Cao et al. on WVS Q3.**
>
> | Setup | **Method** | **Test MSE ↓** | **Test 1-JSD ↑** | **Test MCPA ↑** | **Test EMD ↓** |
> | --- | --- | --- | --- | --- | --- |
> | P2P – US | AEG + Lasso | 0.0217 ± .0014 | 0.7492 ± .0036 | 0.6271 ± .0143 | 0.1188 ± .0048 |
> | P2P – US | AEG + Enet | 0.0218 ± .0019 | 0.7497 ± .0051 | 0.6384 ± .0317 | 0.1184 ± .0066 |
> | P2P – GB | AEG + Lasso | 0.0249 ± .0009 | 0.7416 ± .0055 | 0.4889 ± .0347 | 0.1296 ± .0045 |
> | P2P – GB | AEG + Enet | 0.0249 ± .0009 | 0.7463 ± .0067 | 0.5111 ± .0096 | 0.1294 ± .0048 |
> | P2P – HK | AEG + Lasso | 0.0284 ± .0023 | 0.7115 ± .0125 | 0.4802 ± .0353 | 0.1507 ± .0122 |
> | P2P – HK | AEG + Enet | 0.0290 ± .0023 | 0.7151 ± .0133 | 0.4689 ± .0298 | 0.1532 ± .0108 |
> | Cao et al. - C1 | SFT on Llama3-8B-Instruct | – | 0.777 | 0.43 | –* |
>
> *Excluded for the reasons listed in (c).
>
> (To be continued)

---

> ### Author Response · Authors · 2025-11-19
> **Response to Reviewer XZ1y Part 2/4**
>
> (Continued from previous comment)
>
> In summary, **P2P achieves 1-JSD on Q3 comparable to Cao et al.’s SFT model and matches or exceeds their MCPA, despite using far fewer training entries per locale and no fine-tuning** (Table R0).
>
> Conceptually, Cao et al. cast the problem as a supervised prediction task, with a single SFT model trained on data from 46 countries and conditioned on locale to directly predict answer distributions. P2P instead treats each survey panel as a benchmark for *a specific population preference on a given topic domain at a given moment in time*, where such preference is *gaugeable* through survey responses (cf. Figure 1, right panel in our paper). Our experiments on the Cao et al. arena are therefore be best viewed as a **cross-topic stress test of aligned preferences**, not as the primary regime for which P2P is designed. Importantly, we do *not* assume that preferences on a domain can be reliably inferred if the training questions never touch that domain. For example, inferring political attitudes solely from non-political or religious items is a strong assumption for both humans and synthetic agents. We therefore interpret performance on Q3 as evidence that some preference structure transfers across topics, rather than as a requirement that a preference-alignment method must perfectly extrapolate to arbitrary unseen domains. Studying how aligned preferences generalize across topics is itself a key direction for future work, and will require both theoretical and empirical advances.
>
> ### **(c) JSD, EMD, and metric choices**
>
> We adopt the reviewer’s suggestion and report 1-JSD, MCPA, and EMD in addition to test MSE for P2P on Q3 in the Cao et al setting. However, we do not quote Cao et al.’s EMD values. In their released code, “EMD” is computed as:
>
> ```python
> emd_score = wasserstein_distance(torch.tensor(golden_dist)/100, torch.softmax(torch.tensor(pred_dist), dim=0).cpu())
> ```
> i.e., the Wasserstein distance is applied to the *probability values* themselves, not to an *ordered categorical support* with those *probabilities as masses*. This difference makes their EMD values not directly comparable to a standard formulation. Even under the standard definition, EMD is (i) not scale-free (it depends on how labels/categories are numerically encoded) and (ii) only well-justified when labels form an approximately linear ordinal scale. Many WVS/ATP survey items are nominal or ordinal with unknown spacing. For instance,
>
> ATP W45:
>
> > Which of the following would you say you prefer for getting news?
> >
> >
> > {1.0: 'A print newspaper', 2.0: 'Radio', 3.0: 'Television', 4.0: 'A social media site (such as Facebook, Twitter or Snapchat)', 5.0: 'A news website or app', 99.0: 'Refused'}
> >
>
> WVS:
>
> > If you had to choose, which one of the things on this card would you say is most important?
> {"A": "Maintaining order in the nation", "B": "Giving people more say in important government decisions", "C": "Fighting rising prices", "D": "Protecting freedom of speech"}
> >
>
> In these cases, standard EMD inevitably bakes in unverified cardinality assumptions.
>
> For these reasons, we emphasize **scale-free metrics such as 1-JSD**, **MSE**, together with MCPA, as our primary criteria, and report EMD for P2P only with explicit caveats.
>
> ## **W3: Overfitting concern and cross-dataset generalization**
>
> We agree that overfitting to survey artifacts is a key concern for *any* survey-based evaluation. This concern is closely related to what the robustness literature calls *shortcut learning* or *Clever Hans behavior*, where models exploit superficial regularities of a dataset instead of learning the underlying concept (e.g., Geirhos et al., 2020 [1]; Lapuschkin et al., 2019 [2]).
>
> Our empirical evaluation across 14 ATP waves (Sec. 4 & Appendix C.3) is designed to **mitigate the risk that our findings hinge on idiosyncrasies of a single wave**. In addition, we now **step outside ATP in two ways**. First, our above new head-to-head comparison with Cao et al. on WVS (see W2) shows that P2P performs competitively on a different questionnaire, populations, and a topic shift. Second, on WVS for the three representative locales (US, GB, HK) with a standard 7:1.5:1.5 random split over questions (resolving the Q1+Q2 to Q3 topic shift), P2P attains test MSE ≈ 0.014-0.024 and 1-JSD ≈ 0.76-0.81, which is in line with our ATP results (see **Table R3** in our reply to Reviewer **1dZo**). We view these results as initial evidence against severe ATP-specific shortcutting, but not as a complete solution to the generalization problem.
>
> (To be continued)

---

> ### Author Response · Authors · 2025-11-19
> **Response to Reviewer XZ1y Part 3/4**
>
> (Continued from previous comment)
>
> A full multi-survey generalization study (e.g., across GlobalOpinionQA and other polling corpora) is beyond the scope of our paper, but **P2P’s modular pipeline readily supports this kind of analysis**. Fundamentally, we see this as partly a survey design challenge: if we want synthetic agents to learn stable preferences rather than survey-specific shortcuts, future work may need survey instruments that are explicitly constructed with learnability and cross-topic generalization in mind. In the last paragraph of Appendix C.3, we suggest a semantic question module for P2P to quantify the expressiveness of survey questions, and we will highlight this direction more explicitly in the revised Future Work section.
>
> [1] Geirhos et al., 2020. “Shortcut Learning in Deep Neural Networks.”
>
> [2] Lapuschkin et al., 2019. “Unmasking Clever Hans Predictors and Assessing Their Vulnerability in Explainable AI.”
>
> ## **W4: Reweighting of heterogeneous scales**
>
> We thank the reviewer for foregrounding our discussion on question weighting (Appendix F “Question weighting and regression loss”) and agree that the current weighting scheme gives more weights to questions with more answer options. As noted in Appendix F, the default weighting is built upon the information-theoretic assumption that a finer scale carries higher information for the latent preference. That said, we recognize this is not the only defensible choice, especially when scales are not psychometrically validated. In P2P, we additionally provide the user with the option of **block weighting**, where each question contributes equally to the regression loss.
>
> As a robustness check, we report block weighting results here for ATP Wave 42, with means calculated 3 independent runs. **The results are comparable under the two weighting schemes.**
>
> | **Metric** | Train MSE (Lasso) | Test MSE (Lasso) | Train MSE (Enet) | Test MSE (Enet) |
> | --- | --- | --- | --- | --- |
> | Original | 0.00526 ± 0.00141 | 0.00946 ± 0.00241 | 0.00353 ± 0.00161 | 0.00792 ± 0.00261 |
> | With block weighting | 0.00472 ± 0.00228 | 0.00884 ± 0.00281 | 0.00389 ± 0.00152 | 0.00797 ± 0.00263 |
>
> We also spot-checked the block weighting results on WVS GB data (**cf. Table R3 in our reply to Reviewer 1dZo** for the original experiment results). In this case, block weighting yields a similar test MSE (0.0144 for Lasso and 0.0141 for Elastic Net).
>
> ## **W5: Steerability and Pluralism**
>
> We appreciate the link to PRISM (Kirk et al., 2024) and the broader issue of how steerable a given model is along group dimensions. In our view, steerability depends on (i) whether the target population preference lie within the model’s learnable space, and (ii) how well the preset and expanded attributes span the relevant group axes.
>
> **In P2P, we incorporate steerability indirectly via entropy-guided sampling and question patching**: modes that induce more diverse, high-entropy responses on training questions are prioritized for endowment construction (Sec. 3.3), effectively concentrating on attribute combinations the model is most sensitive to. This is an expedient way to exploit a fixed model’s steerability without explicitly estimating it.
>
> We agree that measuring steerability itself is an important problem. By making the attribute bank user-definable and expandable, **P2P is intended as a testbed for steerability studies** (cf. Appendix F “P2P as a benchmark for model steerability”).
>
> We also thank the reviewer for suggesting model selection based on group attributes steerability, and consider an ensemble with mixed model backends an exciting future extension of the P2P pipeline.
>
> ## **Q1: Sensitivity to attribute selection**
>
> As the reviewer noted, attributes are crucial in our framework. In the revised version, we add a small sensitive analysis on ATP where we
>
> - ablate survey-level and question-level patching separately;
> - turn off freeform templates (survey and question patching) completely;
> - increase the per-endowment attribute cap from 10 to 20.
>
> For each setting, we track average entropy, test MSE, and cost. Results are shown in the table below, with means computed using 3 independent runs.
>
> | **Metric** | Avg. Entropy | Test MSE (Lasso) | Test MSE (Enet) | Cost (USD) |
> | --- | --- | --- | --- | --- |
> | Original | **0.4451** ± 0.0256 | **0.0095** ± 0.0024 | **0.0079** ± 0.0026 | 0.9708 ± 0.0036 |
> | Survey patch off | 0.4378 ± 0.0134 | 0.0103 ± 0.0011 | 0.0102 ± 0.0002 | 0.9785 ± 0.0122 |
> | Question patch off | 0.3989 ± 0.0203 | 0.0114 ± 0.0016 | 0.0109 ± 0.0015 | 0.9698 ± 0.0011 |
> | Freeform off | 0.3953 ± 0.0097 | 0.0128 ± 0.0008 | 0.0123 ± 0.0012 | **0.9688** ± 0.0015 |
> | max attributes = 20 | 0.4363 ± 0.0061 | 0.0124 ± 0.0046 | 0.0115 ± 0.0041 | 0.9880 ± 0.0167 |
>
> (To be continued)

---

> ### Author Response · Authors · 2025-11-19
> **Response to Reviewer XZ1y Part 4/4**
>
> (Continued from previous comment)
>
> We find that **the full attribute bank with both survey and question patching yields the strongest performance**. Turning off survey patching has only a mild effect, while disabling question patching and especially ablating the freeform templates completely reduces entropy by roughly 10%, and worsens test MSE by a wider range of 20-35% (Lasso). Increasing the maximum number of attributes per endowment to 20 does not help—entropy stays similar but test MSE degrades and cost increases slightly.
>
> Overall these results suggest that **P2P is reasonably robust to moderate changes in attribute count, but the templates of the attribute bank, especially question-patching, plays an important role in achieving high-entropy, low-error reconstruction**.
>
> ## **Q2: Non-stationarity of preferences**
>
> This question goes to the heart of non-stationary preferences in social science. Any method trained at a single moment risks becoming stale as attitudes and beliefs shift. Our view is that **synthetic agents should be periodically re-anchored** using more recent, easier-to-collect data (e.g., shorter or cheaper surveys), and then tested on more demanding items. We will consolidate this as an application note in the Appendix.
>
> Architecturally, P2P is set up for this. The endowment generation and regression stages can be rerun on new waves, and one can track how learned weights and entropy patterns evolve, or when reconstruction quality begins to deteriorate. We will explicitly flag this temporal alignment problem as an important direction for future work in the revised paper. While P2P provides the machinery, systematic studies of preference shift and alignment across time and topics remain open.
>
> ## **Q3: Minority subgroups and coverage**
>
> By design, P2P’s endowment generation is **demographic-agnostic** and **data-agnostic**. It explores attribute combinations and questions without matching empirical subgroup frequencies. Abstracting away from learnability concerns (Def. 2.2), this design is intended to *over-cover* the attitudinal space, so that a wide range of (including minority) perspectives appear in the candidate angel pool. The regression stage then selects a sparse mixture to match aggregate survey distributions. Moreover, in our **Simulation Studies (Appendix B)** where learnability is known, we showed that **Lasso conducts principled selection**, **prioritizing ground-truth profiles when they are in the candidate pool (Figure 4)**, and only resorts to reconstruction when such profiles are absent or the number of observations falls below a very low threshold. (We provide a more detailed reply on how the Simulation Studies address the minority subgroup representation question in our reply to Reviewer **qjZY**.)
>
> If the training data under-represent a subgroup, any purely data-driven method can underweight it, including P2P. However, P2P offers two handles:
>
> - users can fix certain demographic attributes (e.g., “18-29, high-school education”) when generating endowments to guarantee coverage of specific subgroups; and
> - the attribute bank can be enriched with subgroup-specific descriptors inspired by resources like PRISM, then the same alignment machinery applies.
>
> We believe these design features give users more control over subgroup coverage when needed.
>
> ---
>
> We also thank the reviewer for their follow-up comment synthesizing concerns from other reviews. In response, we (i) extend our prompting baselines to additional ATP waves to address concerns about wave-specific effects (see our reply to Reviewer **1dZo**); (ii) clarify our design choice and the interpretability of the generated ensemble (see our reply to Reviewer **qjZY**); (iii) add a head-to-head comparison with Cao et al. (2025), introduce a clustering baseline (see our reply to Reviewer **qjZY**), and explain why the other proposed works do not constitute directly comparable baselines in our setting (see our reply to Reviewer **S997**).
>
> We thank the reviewer again for the constructive feedback and hope that our responses improve your opinion on our paper.

---

### Author Response · Authors · 2025-11-19
**Global Response to All Reviewers | Part 1/2**

We thank all reviewers for their valuable feedback. In this global comment, we synthesize the common concerns that appeared across the reviews and summarize the steps we took during the rebuttal period to address each of them. Full point-to-point responses are offered in the individual reviewer threads. All newly added experiments and results during rebuttal are provided in Appendix G of the revised paper for easy reference while keeping the main text within the 9-page limit.

## **[I] Strengthened empirical evaluation, ablation and new baselines**

A primary suggestion across reviewers (esp. XZ1y, 1dZo, and qjZY) was to broaden empirical validation. In the rebuttal, we addressed this systematically through:

- **New evaluation metrics (1-JSD, EMD, MCPA).**

    The added metrics are consistent with the trends observed earlier with MSE, providing further evidence on the key findings of the paper. We thank reviewer XZ1y for explicitly flagging the metrics (1-JSD & EMD).

- **Extra baseline comparisons across multiple ATP waves.**

    To address Reviewer 1dZo’s concern about evaluation breadth and potential wave-specific effects, we extended the Vanilla and PERSONA baseline comparisons to three representative ATP waves (weak/medium/strong alignment), each run with multiple repeats. P2P continues to outperform both baselines across waves, congruent with findings presented in Table 1.

- **Direct comparison with Cao et al. (2025) & cross-dataset generalization.**

    Following suggestions from Reviewers XZ1y and 1dZo, we completed a head-to-head evaluation on Cao et al.’s WVS setting using their topic-shift (Q1/Q2→Q3) protocol. We ran P2P on the US, Great Britain, and Hong Kong SAR subsets. Despite far fewer training entries per locale and no finetuning, P2P achieves competitive 1-JSD and MCPA compared with their SFT method.

    We also evaluated P2P under a standard 7:1.5:1.5 random split across the same locales (US/GB/HK), showing performance in line with our ATP findings and demonstrating that P2P’s alignment mechanism transfers across survey programs.

- **New clustering baseline.**

    Inspired by Reviewer qjZY’s suggestion, we added k-means medoid variants for our Vanilla and PERSONA baselines, starting with pools of 1000 personas. Clustering does not improve either entropy or MSE but slightly increases cost, indicating that embedding-space clustering does not constitute stronger baselines to evaluate the performance of P2P. This further validates our original baselines as reasonable and competitive points of comparison.

- **Backend performance baseline.**

    Per Reviewer 1dZo’s suggestion, we provided the backend performance for gemini-2.0-flash for reference, where it is directly queried as an “average adult” in the population. This baseline yields significantly worse MSE and entropy than all existing baselines, confirming the gains of the ensemble methods.

- **More sensitivity analyses.**

    We appreciate Reviewer XZ1y’s questions on sensitivity checks, and included two additional ablation studies: (i) question weighting, where questions contribute equally to the training loss; (ii) attribute ablation, where we test the impact of freeform templates and more attributes. Results show that performance is robust to question weighting and more attributes per endowment, and that turning off the freeform templates (especially question patching) degrades both entropy and MSE.


All results from these new experiments are contained in Appendix G of the revised paper.

## **[II] Theoretical clarity and consistency**

Three reviewers (S997, qjZY, XZ1y) found our theoretical framing clear and sound, while Reviewer 1dZo raised concerns regarding derivations. To avoid misinterpretation, we

- clarified the links among the paper’s Definitions, Propositions, and Theorems;
- clarified our styling choice for the theorem statements and the formal proofs in the Appendix;
- highlighted why the ensemble approach is not less powerful than a single distribution-calibrated model using our propounded theory.

No substantial theoretical changes were introduced, in keeping with the overall reviewer consensus that the theoretical content was appropriate and to the point for the scope of the paper.

---

### Author Response · Authors · 2025-11-19
**Global Response to All Reviewers | Part 2/2**

## **[III] Minority representation and method interpretability**

Reviewer qjZY and XZ1y raised questions on minority perspectives and subgroup representation. In the response, we emphasized

- Simulation Study 1 (Appendix B) evaluates recovery of minority profiles under known ground truth. When true profiles are present in the candidate pool, Lasso correctly prioritizes these files and assign accurate weights, only resorting to reconstruction when data are extremely scarce or ground truth profiles are absent.
- Conditioned on learnability being sufficiently satisfied (Sec 2), Stage 1 (AEG) of P2P is intentionally designed to over-cover the preference spectrum by focusing on optimizing response entropy. In practice, users can also enforce demographic coverage through the extensible attribute bank and modular framework.
- Interpretability can be approached through a survey sampling vantage and consistently framed with our preference reconstruction theory.

These clarifications are foregrounded and explained in more detail in the reviewer-specific responses.

## **[IV] Preprocessing & typo correction**

While running the new baselines and computing the distribution metrics, we identified and redressed a small preprocessing inconsistency in the ATP W42 repeat run 1 (which used an outdated, overly strict missing-value rule). The update does not alter any algorithmic components nor affect other waves and experiment runs in the paper. As a sole consequence, it slightly improves P2P’s MSE in Table 1 (e.g., 0.0104 → 0.00946 for Lasso). All new results in the rebuttal used the modern, consistent pipeline.

We also fixed a few typographical errors in the revised paper to improve clarity.

## **[V] Structure of revised submission**

Given the 9-page limit for the main paper, all new experiments, expanded tables, and supplementary discussions appear in the new **Appendix G**, intentionally placed after the Limitations and Future Work sections for ease of navigation. We will further consolidate them in the main paper when more space permits.

In sum, the aforementioned revisions directly address the concerns raised across the reviews, strengthen the empirical validation of P2P, and clarify the theoretical framework, design choice and the interpretability of P2P. We are deeply grateful for the reviewers’ careful reading and constructive feedback during the rebuttal period, which has helped enormously to improve the clarity and robustness of our submission.

---

### Meta-Review · Area_Chair_kCMK · 2026-01-10

**Summary:**

The paper proposes Prompts-to-Proxies (P2P), a two-stage framework for pluralistic alignment that aims to reconstruct population-level preferences. The method involves generating a diverse basis of LLM agents via entropy-guided sampling (Stage 1) and aggregating their responses using a regression model to match target population statistics (Stage 2). The authors frame this approach within a "preference reconstruction theory," arguing that population preferences can be represented by a functional basis of proxy agents. The method is evaluated on the American Trends Panel (ATP) survey data.

**Reviewer Concerns:**

Initially, reviewers raised several significant concerns regarding both the theoretical foundation and the empirical evaluation. Reviewer 1dZo strongly criticized the theoretical contribution, arguing that the proofs are trivial consequences of the definitions and that the presentation is unnecessarily complicated. Reviewer XZ1y and Reviewer S997 pointed out missing baselines, specifically requesting comparisons to recent works like Cao et al. (2025) and questioning the evaluation's limitation to US-centric data (OpinionQA). Reviewer qjZY raised concerns about whether the method truly preserves minority perspectives or simply optimizes for MSE, and suggested that simple clustering baselines might outperform the proposed method. There were also concerns regarding the limitation of the framework to multiple-choice survey formats (Reviewer S997) and the potential for overfitting to specific survey waves (Reviewer XZ1y).

**Reviewer Scores:**

The initial scores were (6,2,4,4). During the rebuttal, the authors made a substantial effort to address the empirical concerns. They added new metrics (JSD, EMD), extended baselines to more waves, included a direct comparison with Cao et al. (2025), and added a clustering baseline as requested. However, despite these extensive empirical additions, the fundamental concerns regarding the paper's theoretical significance and novelty remain unresolved. Reviewer 1dZo maintained their low score (2), emphasizing that the theoretical results (Theorems 1 & 2) follow directly from the definitions and do not constitute a substantial contribution, despite the authors' clarifications. Furthermore, Reviewers S997 and qjZY did not respond to the rebuttal. Given this silence, it remains unclear whether the additional experiments and clarifications were sufficient to alleviate their reservations. Consequently, there is no strong basis to assume that the initial concerns, regarding the method's limitation to survey formats and the validation of minority perspectives, have been resolved to a level that leads to a score increase. Given the strong disagreement on the significance of the theoretical contribution and the borderline empirical assessment, the paper is not recommended for acceptance at this time.

---

### Decision · Program_Chairs · 2026-01-26

Reject